# How Transformers Learn Causal Structures In-Context: Explainable Mechanism Meets Theoretical Guarantee

**Jianzhe Wei**[1,2]   **Siyu Chen**[2]   **Jianliang He**[2]   **Zhuoran Yang**[2,*]
[1]Georgia Institute of Technology   [2]Yale University
jwei345@gatech.edu, {siyu.chen.sc3226,jianliang.he,zhuoran.yang}@yale.edu

## Abstract

Transformers have demonstrated remarkable in-context learning abilities, adapting to new tasks from just a few examples without parameter updates. However, theoretical understanding of this phenomenon typically assumes fixed dependency structures, while real-world sequences exhibit flexible, context-dependent relationships. We address this gap by investigating whether transformers can learn causal structures – the underlying dependencies between sequence elements – directly from in-context examples. We propose a novel framework using Markov chains with randomly sampled causal dependencies, where transformers must infer which tokens depend on which predecessors to make accurate predictions. Our key contributions are threefold: (1) We prove that a two-layer transformer with relative positional embeddings can implement Bayesian Model Averaging (BMA), the optimal statistical algorithm for causal structure inference; (2) Through extensive experiments and parameter-level analysis, we demonstrate that transformers trained on this task approximate BMA, with attention patterns directly reflecting the inferred causal structures; (3) We provide information-theoretic guarantees showing how transformers recover causal dependencies and extend our analysis to continuous dynamical systems, revealing fundamental differences in representational requirements. Our findings bridge the gap between empirical observations of in-context learning and theoretical understanding, showing that transformers can perform sophisticated statistical inference over structural uncertainty.

## 1 Introduction

Modern transformers exhibit a remarkable capability: they can adapt to entirely new tasks using only a handful of examples, without any parameter updates. This phenomenon, known as in-context learning (ICL) Brown et al. (2020), has revolutionized our understanding of what neural networks can achieve. A model trained on diverse text can suddenly perform arithmetic, translate languages, or write code – all by simply observing a few demonstrations. Yet despite extensive empirical success Wei et al. (2022); Garg et al. (2022) and theoretical investigations Von Oswald et al. (2023); Akyürek et al. (2023); Goel & Bartlett (2024), a fundamental question remains: how do transformers adapt to the varying dependency structures present in real-world sequences? (Allen-Zhu & Li, 2025; Bietti et al., 2023; Zhao et al., 2023; Wibisono & Wang, 2024)

**The Theory-Practice Gap.** Current theoretical understanding of ICL rests on a critical simplification: most analyses assume that dependencies between sequence elements follow a fixed, predetermined structure. For instance, theoretical works typically study settings where tokens are independent $[[x_1, f(x_1)], [x_2, f(x_2)], ...]$ or follow rigid patterns like $[x_1, f(x_1), x_2, f(x_2)]$ (Bai et al., 2023; Chen et al., 2024a; Wang et al., 2025). However, natural language and real-world sequences exhibit far richer structure – words depend on previous words in complex, context-dependent ways that vary across sentences and domains. Recent work by Nichani et al. (2024) began addressing this by showing transformers can encode fixed causal structures during training. Specifically, they assume

---

*Corresponding author.

an $n$-gram causal model (e.g., bigrams where each token depends only on the previous one) (Rajaraman et al., 2024; Edelman et al., 2024), and prove that transformers can embed this structure in their attention weights to perform inference. However, in real-world scenarios, the dependency graph itself is not fixed but varies across different sequences. For example, in language, the syntactic structure can change dramatically between different documents, and in stock price prediction, the relationships between assets can shift over time. Thus, a key challenge is

*Can transformers infer and adapt to causal structure in-context?* $\quad\quad\quad (\star)$

**Our Approach.** We introduce a novel framework where sequences are generated from Markov chains with randomly sampled causal dependencies. In our setting, each token depends on exactly one predecessor, or its "parent", but crucially, these parent relationships are not fixed and must be inferred from context examples, which is a collection of sequences sharing the same underlying causal structure. This setup captures the essence of $(\star)$ by requiring the model to adapt to different latent structures across contexts. The transformer must infer these latent dependencies from context examples to accurately predict new sequences – mirroring how language models must adapt to different syntactic structures or reasoning patterns.

**Main Contributions.** We consider two types of Markov chains: discrete chains over a finite vocabulary and continuous linear dynamical systems. Our work makes the following contributions:

**(1) Theoretical Construction:** For discrete Markov chains, we prove that a two-layer transformer with relative position embeddings can implement Bayesian Model Averaging (BMA), the statistically optimal algorithm for inferring causal structures from observations. Our construction shows how attention mechanisms can perform sophisticated probabilistic inference over structural uncertainty. **(2) Empirical Verification:** Through extensive experiments on Markov chains, we demonstrate that transformers trained via gradient descent converge to solutions remarkably similar to our theoretical construction. Parameter-level analysis reveals that learned attention patterns directly encode posterior probabilities over causal structures, providing mechanistic insight into how transformers perform statistical inference. **(3) Information-Theoretic Analysis:** We establish conditions under which causal structures can be recovered in-context, using mutual information and data processing inequalities. Additionally, we show that gradient-based learning naturally discovers these structures early in training through $\chi^2$-mutual information maximization. **(4) Extensions to Continuous Systems:** We extend our framework to linear dynamical systems in continuous space, revealing fundamental differences in how transformers handle discrete versus continuous causal inference. While transformers show strong empirical performance, we identify representational limitations that prevent exact BMA implementation in continuous settings.

## 2 PRELIMINARY

### 2.1 TASK SETUP

To investigate the question $(\star)$, we consider data generated from distributions with a latent causal structure. Each sample is a sequence of tokens $\boldsymbol{x}_{1:H} = [\boldsymbol{x}_1, \ldots, \boldsymbol{x}_H]$, where the $h$-th token $\boldsymbol{x}_h$ depends on one of its predecessors, called the parent token $\boldsymbol{x}_{\mathrm{pa}(h)}$. This dependency relation is represented as a directed tree graph $\mathcal{G} = \{\mathrm{pa}(h)\}_{h \in [H]}$, where $\mathrm{pa}(h) \sim \mathrm{Unif}(1, \ldots, h-1), \forall h \in \{2, \ldots, H\}$. Given the causal structure defined above, the data generation process can be written as $\boldsymbol{x}_h = G(\boldsymbol{x}_{\mathrm{pa}(h)})$, where $G(\cdot)$ denotes either stochastic sampling from the transition kernel $\pi(\cdot | \boldsymbol{x}_{\mathrm{pa}(h)})$ of Markov chains, or the autoregressive process for dynamical systems. $G(\cdot)$ is fixed during the sampling of the whole dataset.

For the in-context learning task, suppose we have $L+1$ samples $\{\boldsymbol{x}_{1:H}^{(l)}\}_{l \in [L+1]}$ from the same causal graph $\mathcal{G}$. The first $L$ samples are provided as in-context demonstrations from which the model infers the latent graph structure, while the last sample is the target for prediction. Except for the first token $\boldsymbol{x}_1^{L+1}$, every token $\boldsymbol{x}_h^{L+1}$ in this trajectory is required to be predicted via next-token prediction conditioned on $\boldsymbol{x}_{1:H}^{1:L}$ and its past observations $\boldsymbol{x}_{1:h-1}^{L+1}$.

**Markov Chain.** Following the Markovian assumption adopted in Edelman et al. (2024); Nichani et al. (2024); Chen et al. (2024b), we assume that sequences are sampled from a Markov chain with random dependencies. In this setting, tokens $\{\boldsymbol{x}_h\}$ are drawn from a finite vocabulary $\mathcal{V} =$

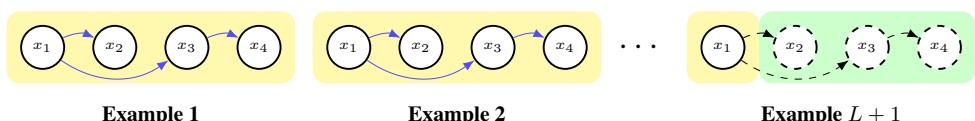

**Figure 1: Task overview of in-context causal structure learning.** Each training sequence consists of $L$ examples with observed variables and hidden parent relations, followed by a new example $L+1$ where the model must infer the underlying parent indices in context from previous demonstrations.

$\{e_1, \ldots, e_d\}$, where $|\mathcal{V}| = d$ and $\{e_i\}$ are one-hot vectors. The random dependencies are specified by a latent causal graph $\{\mathrm{pa}(h)\}_{h \in [H]}$. Let $\pi : \mathcal{V} \to \Delta(\mathcal{V})$ denote the Markov transition kernel, where $\Delta(\mathcal{V})$ is the probability simplex over $\mathcal{V}$. Then each token is generated as $\boldsymbol{x}_h \sim \pi(\,\cdot\,|\boldsymbol{x}_{\mathrm{pa}(h)}) \in \Delta(\mathcal{V})$, $\forall h \in [H]$, where, by slight abuse of notation, we also regard $\pi$ as the stochastic matrix $\pi \in \mathbb{R}^{|\mathcal{V}| \times |\mathcal{V}|}$ with $\pi[i, j] = \pi(j|i)$, $\sum_j \pi[i, j] = 1$.

**Dynamical System.** Beyond the discrete Markov chain case, we also consider a more challenging setting with a continuous sampling space. Here tokens $\{\boldsymbol{x}_h\}$ are dense vectors in $\mathbb{R}^d$. The link function $g(\cdot)$ replaces the discrete transition kernel, and we instantiate it as a *linear dynamical system with additive Gaussian noise*: $\boldsymbol{x}_h = g(\boldsymbol{x}_{\mathrm{pa}(h)}) = \rho A^\top \boldsymbol{x}_{\mathrm{pa}(h)} + \sqrt{1 - \rho^2}\,\boldsymbol{\eta}_h$, where $A \in \mathcal{O}(\mathbb{R}^d)$ is orthogonal, $\boldsymbol{x}_1 \sim \mathcal{N}(0, I_d)$, $\boldsymbol{\eta}_h \sim \mathcal{N}(0, I_d)$ and $\rho$ ensures the stability of the sequence.

These settings evaluate the extent to which transformers can perform in-context causality learning.

**Goal: Inferring the Causal Structure.** The task formulation naturally raises the following question: *Given $L$ in-context examples, how can the model infer the underlying graph structure $\mathcal{G}$?* A classical approach to this problem is *Bayesian Model Averaging* (BMA), which leverages Bayes' rule to compute the posterior distribution over the possible parameter space. Treating the parent structure $\mathrm{pa}(h)$ as the parameter to be estimated, the distribution of having parent $h'$ will be predicted as its posterior probability given $L$ observations:

$$\mathbb{P}\big(\mathrm{pa}(h) = h' \,\big|\, \boldsymbol{x}_{1:H}^{1:L}\big) = \frac{\mathbb{P}\big(\boldsymbol{x}_{1:H}^{1:L} \,\big|\, \mathrm{pa}(h) = h'\big)\, \mathbb{P}(\mathrm{pa}(h) = h')}{\sum_{h'' \in [H]} \mathbb{P}\big(\boldsymbol{x}_{1:H}^{1:L} \,\big|\, \mathrm{pa}(h) = h''\big)\, \mathbb{P}(\mathrm{pa}(h) = h'')}. \tag{1}$$

By Eq. (1) and our task assumption, we have the following lemma of the formulation of BMA.

**Lemma 1.** *Suppose $L$ samples are observed from the Markov chain (or dynamical system) $\mathrm{x}_{1:H}$ with latent causal structure $\mathcal{G}$. Bayesian Model Averaging makes prediction of $pa(h) \in [h - 1]$:*

$$\mathbb{P}(pa(h) = h' \mid \boldsymbol{x}_{1:H}^{1:L}) = \frac{\exp(\sum_{l \in [L]} \log \pi(\boldsymbol{x}_h^l | \boldsymbol{x}_{h'}^l))}{\sum_{h'' \in [h-1]} \exp(\sum_{l \in [L]} \log \pi(\boldsymbol{x}_h^l | \boldsymbol{x}_{h''}^l))} = \sigma\big(\hat{\boldsymbol{p}}^{h,L}(\log \pi)\big)_{h'}, \tag{2}$$

*where $\hat{\boldsymbol{p}}^{h,L}(\log \pi) \in \mathbb{R}^{h-1}$, $\hat{\boldsymbol{p}}_{h'}^{h,L} = \sum_{l \in [L]} \log \pi(\boldsymbol{x}_h^l | \boldsymbol{x}_{h'}^l)$. See Appendix C.2 for detailed proof.*

This Bayesian formulation provides a principled baseline for inferring causal structure, and serves as a point of comparison for the in-context learning behavior of transformers.

## 2.2 MODEL ARCHITECTURE

### 2.2.1 STANDARD TRANSFORMER

Decoder-only Transformer is a neural network structure for handling sequential data. Given a sequence of tokens $(\boldsymbol{w}_1, \ldots, \boldsymbol{w}_T)$, transformers first embed these tokens and add a positional encoding: $\boldsymbol{h}_t^{(0)} = E(\boldsymbol{w}_t) + P(t) \in \mathbb{R}^d, \forall t \in [T]$. In matrix form, the mapped tokens of the input are represented as $\boldsymbol{H}^{(0)} = \boldsymbol{h}_{1:T}^{(0)\top} \in \mathbb{R}^{T \times d}$. Subsequent layers consist of multi-headed attention layers (MHA) followed by multilayer perceptron layers (MLP). At layer $l$, the hidden features $\boldsymbol{H}^{(l-1)}$ are updated as follows. First, the causal-mask self-attention layer computes the ouput by:

$$\mathrm{Attn}(\boldsymbol{H}; \boldsymbol{W}_Q, \boldsymbol{W}_K, \boldsymbol{W}_V) = \sigma\bigg(\mathcal{M}\Big(\frac{(\boldsymbol{H}\boldsymbol{W}_Q)(\boldsymbol{H}\boldsymbol{W}_K)^\top}{\sqrt{d_k}}\Big)\bigg)\boldsymbol{H}\boldsymbol{W}_V,$$

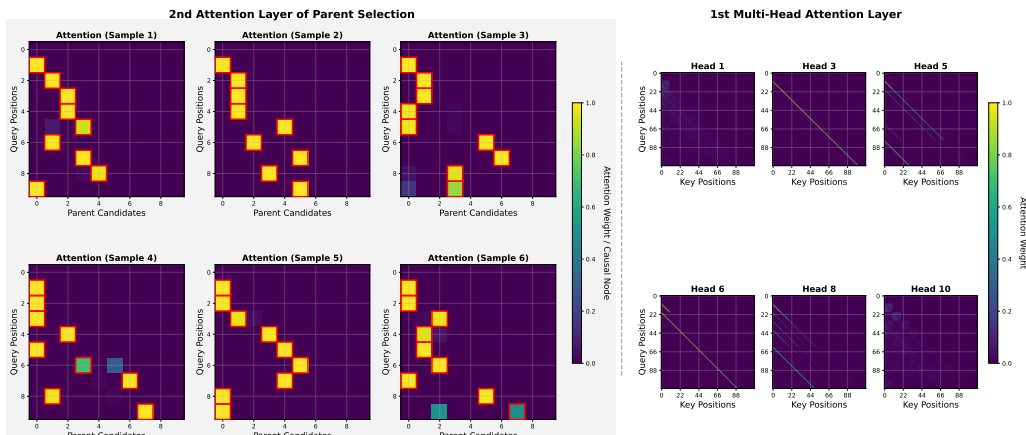

Figure 2: Visualization of Attention Weights $\mathcal{A}^{(1)}, \mathcal{A}^{(2)}$. Left: $\mathcal{A}^{(2)}$ on 6 examples. Transformer shows the capability to select true causal tokens (red rectangle). Right: Six representative heads (out of 10) from $\mathcal{A}^{(1)}$ (Head 1 and 10 degenerate). Trained with $L = 10, H = 10, d = 5$, 1024 steps.

where $\boldsymbol{W}_Q, \boldsymbol{W}_K, \boldsymbol{W}_V \in \mathbb{R}^{d \times d_k}$, $\sigma(\boldsymbol{v})_i = \frac{\exp(v_i)}{\sum_j \exp(v_j)}$ applied to matrix row-wisely, and $\mathcal{M}$ is the causal mask where $\mathcal{M}(\boldsymbol{X})_{ij}$ is $-\infty$ if $i > j$ else $\boldsymbol{X}_{ij}$. Then multi-headed attention gives the output:

$$\text{MHA}(\boldsymbol{H}) = \bigg( \bigoplus_{m=1}^{M} \text{Attn}(\boldsymbol{H}; \boldsymbol{W}_Q^m, \boldsymbol{W}_K^m, \boldsymbol{W}_V^m) \bigg) \boldsymbol{W}_O,$$

where $\bigoplus$ denotes the concatenation of vectors and $\boldsymbol{W}_O \in \mathbb{R}^{Md_k \times d}$. After obtaining the intermediate features $\text{MHA}_l(\boldsymbol{H}^{(l-1)})$ from the attention layer, this feature will be added to the **residual stream**, which aggregates the previous output: $\hat{\boldsymbol{H}}^l = \boldsymbol{H}^{(l-1)} + \text{MHA}_l(\boldsymbol{H}^{(l-1)})$. The FFN layer adopts this as input and updates this stream as:

$$\text{FFN}(\hat{\boldsymbol{H}}) = \sigma\big(\hat{\boldsymbol{H}} \boldsymbol{W}_1\big) \boldsymbol{W}_2, \quad \boldsymbol{H}^{(l)} = \hat{\boldsymbol{H}}^{(l)} + \text{FFN}_l(\hat{\boldsymbol{H}}^{(l)}),$$

where $\boldsymbol{W}_1 \in \mathbb{R}^{d \times d_m}, \boldsymbol{W}_2 \in \mathbb{R}^{d_m \times d}$, and $\sigma(\cdot)$ is the activation function. Finally, the output of the $L$-layer Transformer is $\sigma(\boldsymbol{H}^{(L)} \boldsymbol{W}_U)$, projected to vocabulary logits by $\boldsymbol{W}_U \in \mathbb{R}^{d \times V}$.

### 2.2.2 DISENTANGLED TRANSFORMERS

To better analyze the role that each part of transformers plays in learning a task, prior works Friedman et al. (2023) propose the disentangled transformer, which decouples the intertwined features in the residual stream. Instead of adding each layer's output, the disentangled transformer concentenates it with the residual stream. Suppose in transformers, the hidden states are $\boldsymbol{H}^{(l-1)} \in \mathbb{R}^{T \times d_{l-1}}$:

$$\boldsymbol{H}^{(l)} = [\boldsymbol{H}^{(l-1)}, \text{Attn}_1(\boldsymbol{H}^{(l-1)}), \dots, \text{Attn}_M(\boldsymbol{H}^{(l-1)})] \in \mathbb{R}^{T \times (1+M)d_{l-1}}. \tag{3}$$

Here we consider the decoder-based attention-only transformers. And following Nichani et al. (2024), in each attention head, $\boldsymbol{W}_K \boldsymbol{W}_Q^\top$ is reparameterized by $\boldsymbol{W}_{KQ}$, $\boldsymbol{W}_O \boldsymbol{W}_V$ by $\boldsymbol{W}_{OV}$ and the initial input $\boldsymbol{H}^{(0)}$ is given by $\boldsymbol{h}_t^{(0)} = [E(\boldsymbol{w}_t), P(\boldsymbol{w}_t)] = [\boldsymbol{e}_{\boldsymbol{w}_t}, \boldsymbol{e}_t] \in \mathbb{R}^{d+T}$. In our task, the input sequence consists of $L + 1$ examples of length-$H$ chains, leading to the positional embedding size $T$ equal to $(L+1)H$. If we set $d = H = L = 10$, then $d = 10 \ll T = 110$ in the input embedding and $\boldsymbol{W}_{KQ}$ in the first layer will have $\Theta(H^2 L^2) = \Theta(10^4)$ parameters. Instead, considering $\boldsymbol{w}_t$ as the $h$-th token in example $l$, we use two types of embeddings to represent this positional information: $\text{Pos}_L(\boldsymbol{w}_t) = \boldsymbol{e}_l \in \mathbb{R}^L$, $\text{Pos}_H(\boldsymbol{w}_t) = \boldsymbol{e}_h \in \mathbb{R}^H$. This reduces the required parameters to $\Theta(H^2 + L^2)$ for training. The formulation of this transformer is based on Eq. (60) and (69).

**Relative Positional Embedding.** While the original transformer employs the absolute positional embeddings, subsequent research has demonstrated the advantages of relative positional embeddings (RPE) (Shaw et al., 2018; Su et al., 2024). To enable tractable parameter-level analysis, we adopt a simplified structure based on RPE that reduces the parameter space. Crucially, empirical results presented in Appendices G, H and I demonstrate this simplification didn't prevent mechanistic

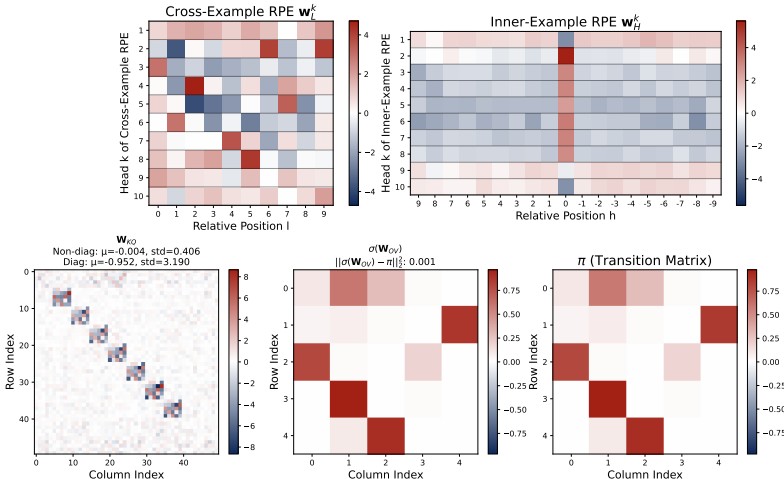

Figure 3: Parameter visualization of 2-layer transformer with RPE. Heads of inner-example RPE $\mathbf{w}_H^k$ uniformly show the largest value at position $h = 0$ except Head 1, 9, 10. Correspondingly, $\boldsymbol{W}_{KQ}$ shows similar blocks on diagonal except block 1, 9, 10. Besides, $\sigma(\boldsymbol{W}_{OV})$ approximates $\pi$.

interpretation on more standard architectures like standard transformers with FFNs and disentangled ones. Instead, the following simplified model helps to understand the mechanism of transformers. For RPE, it is commonly parameterized by a vector $\mathbf{w} \in \mathbb{R}^T$ which assigns attention score $\mathbf{w}(i, j)$ only via relative distance $i - j$ between positions of query $i$ and key $j$. Recall the sequence length is $T = H(L + 1)$. Similar to the absolute positional ones, we adopt two types of RPE: $\mathbf{w}_L \in \mathbb{R}^L$ representing the order $l$ from $L+1$ examples and $\mathbf{w}_H \in \mathbb{R}^{2H-1}$ denoting the order $h$ from $H$ tokens.

$$\mathbf{w}_H(h, h') = \mathbf{w}_H[h - h'], \ \forall (h, h') \in [H]^2, \quad \mathbf{w}_L(l, l') = \begin{cases} \mathbf{w}_L[l - l'], & l > l', \\ -\infty, & \text{else. (for causal mask)} \end{cases}$$

In the transformers we consider, the first layer adopts this RPE. Its output $\boldsymbol{u}_t$ for token $\boldsymbol{x}_t = \boldsymbol{x}_h^l$ is:

$$\begin{aligned} \boldsymbol{u}_t = \text{Attn}_{\boldsymbol{x}_t \to \boldsymbol{x}_{1:T}} &= \sum_{t' \in [T]} \sigma_{t'}(\mathbf{w}_H(h, \cdot) + \mathbf{w}_L(l, \cdot)) \boldsymbol{x}_{t'} \\ &= \sum_{t' \leftrightarrow (h', l')} \frac{\exp(\mathbf{w}_H(h, h') + \mathbf{w}_L(l, l'))}{\sum_{t''} \exp(\mathbf{w}_H(h, h'') + \mathbf{w}_L(l, l''))} \boldsymbol{x}_{t'}. \end{aligned} \tag{4}$$

Suppose we have $K$ heads in the first layer. The outputs $\{\boldsymbol{u}_t^k\}_{k \in [K]}$ will be concatenated by disentangled residual as the input $\boldsymbol{z}_t$ for the next layer. Then the next single-headed **self-attention** layer takes features of last example $\boldsymbol{z}_{1:H}^{L+1}$, as query, key and value tokens and gives the final prediction. Recall the input is $\boldsymbol{x}_{1:T} = \boldsymbol{x}_{1:H}^{1:L+1}$. This transformer architecture is formulated as follows:

**1st RPE Attention (K-head):** $\quad \boldsymbol{u}_h^k = \text{Attn}_{\boldsymbol{x}_h^{L+1} \to \boldsymbol{x}_{1:T}}^k = \sigma(\mathbf{w}_H^k(h, \cdot) + \mathbf{w}_L^k(L+1, \cdot)) \boldsymbol{x}_{1:T}^\top \quad \in \mathbb{R}^d$

**Disentangled Residual:** $\quad \boldsymbol{v}_h = [\boldsymbol{u}_h^1, \dots, \boldsymbol{u}_h^K], \ \boldsymbol{z}_h = [\boldsymbol{x}_h^{L+1}, \boldsymbol{v}_h] \quad \in \mathbb{R}^{d_1}$

**2nd Attention (1-head):** $\quad \boldsymbol{f}_{\text{tf}}(\cdot \,|\, \mathcal{H}_h^L) = \sigma(\boldsymbol{z}_{1:h-1}^\top \boldsymbol{W}_{KQ} \boldsymbol{z}_h)^\top \boldsymbol{z}_{1:h-1}^\top \boldsymbol{W}_{OV}$

$\qquad\qquad\qquad\qquad\qquad = \sigma(\boldsymbol{v}_{1:h-1}^\top \boldsymbol{W}'_{KQ} \boldsymbol{v}_h)^\top \boldsymbol{x}_{1:h-1}^{L+1}{}^\top \boldsymbol{W}'_{OV} \quad \in \mathbb{R}^d$

$$\tag{5}$$

where $\boldsymbol{f}_{\text{tf}}(\cdot \,|\, \mathcal{H}_h^L) \in \mathbb{R}^d$ denotes the output of the transformer based on context $\mathcal{H}_h^L = [\boldsymbol{x}_{1:H}^{1:L}, \boldsymbol{x}_{1:h-1}^{L+1}]$ (or the context denoted by $\mathcal{H}$ for brevity), and we assume some blocks in $\boldsymbol{W}_{KQ}, \boldsymbol{W}_{OV}$ are 0:

$$\boldsymbol{W}_{KQ} = \begin{bmatrix} 0_{d \times d} & 0_{d \times Kd} \\ 0_{Kd \times d} & \boldsymbol{W}'_{KQ} \end{bmatrix}, \boldsymbol{W}_{OV} = \begin{bmatrix} \boldsymbol{W}'_{OV} & 0_{d \times Kd} \\ 0_{Kd \times d} & 0_{Kd \times Kd} \end{bmatrix}, \tag{6}$$

where $\boldsymbol{W}'_{KQ} \in \mathbb{R}^{Kd \times Kd}$, $\boldsymbol{W}'_{OV} \in \mathbb{R}^{d \times d}$ are trainable and are represented by $\boldsymbol{W}_{KQ}$ and $\boldsymbol{W}_{OV}$ for brevity in the rest of the paper. To train transformers, cross-entropy loss is used for the Markov chain (MC) and MSE loss for the dynamical system (DS) shown in Appendix Eq. (13).

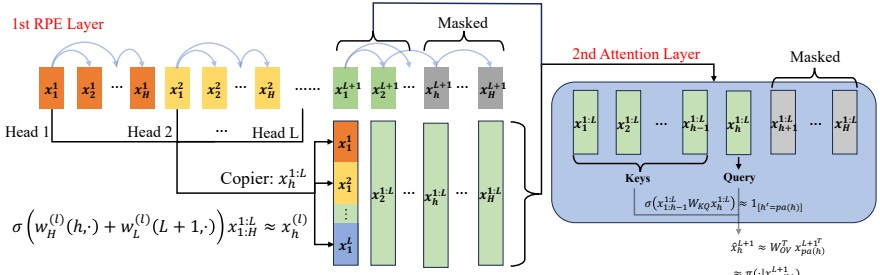

Figure 4: Overview of the constructed mechanism in transformers: the 1st layer works as copier, and the attention patterns in 2nd layer follows BMA, which approximately select correct parent token.

# 3 CAN TRANSFORMERS IN-CONTEXT LEARN CAUSAL STRUCTURES?

## 3.1 2-LAYER TRANSFORMER LEARNED TO SELECT CAUSAL STRUCTURE IN-CONTEXT

To investigate the question ($\star$), we first train 2-layer transformers with RPE introduced above on the Markov chain setting. Each input has $L+1$ samples $\{\boldsymbol{x}^l_{1:H}\}$ of Length-$H$ Markov chain with causal structure $\mathcal{G}$. The input sequence length is $T = H(L+1)$. We set the transformer to have $K$ heads in the first RPE layer: $\{(\mathbf{w}^k_H, \mathbf{w}^k_L)\}_{k\in[K]}$ and 1 head for the 2nd attention layer $(\boldsymbol{W}_{KQ}, \boldsymbol{W}_{OV})$. All RPE parameters are initialized randomly from Gaussian distribution and $(\boldsymbol{W}_{KQ}, \boldsymbol{W}_{OV})$ from zero.

For an attention layer, the attention weights $\mathcal{A}$ normalized by the $\sigma$ reveal which tokens a query primarily attends to, enabling mechanistic interpretability analyses such as circuit discovery Olsson et al. (2022). We first investigate the attention patterns $\mathcal{A}^{(1)}, \mathcal{A}^{(2)}$ from the first and second transformer layers. Mathematically, they are matrices where the $i$-th row denotes the attention weights of token $\boldsymbol{x}_i$ to the whole sequence and $\mathcal{A}^{(*)}_{ij} = \mathcal{A}^{(*)}_{i \to j}$ is formulated by the following for $i, j \in [T]$ and $\mathcal{A}^{(1),k} \in \mathbb{R}^{T\times T}, \mathcal{A}^{(2)} \in \mathbb{R}^{H\times H}$:

$$\mathcal{A}^{(*)} = \sigma(\mathcal{M}(\tilde{\mathcal{A}}^{(*)})), \quad \tilde{\mathcal{A}}^{(1),k}_{i \to j} = \mathbf{w}^k_H(h_i, h_j) + \mathbf{w}^k_L(l_i, l_j), \quad \tilde{\mathcal{A}}^{(2)}_{h \to h'} = \boldsymbol{v}^\top_{h'} \boldsymbol{W}_{KQ} \boldsymbol{v}_h,$$

where the index $i$ is attributed to $\boldsymbol{x}_i$ from Eq. (5) which is the $h_i$-th token of $l_i$-th example (similarly for index $j$), and $\boldsymbol{v}_{h(')}$ are the hidden features $\boldsymbol{v}^{L+1}_{1:H}$ of the $L+1$-th example from Layer 1. Empirically, we observe the trained attention patterns of $\mathcal{A}^{(2)}$ match the groundtruth causal structure in Fig. 2. For the first layer, shown by Fig. 2, some heads of attention weights show specific attendance among tokens while some heads degenerate (e.g., Head 1, 9, 10). Further, we visualize the trainable parameters of 2-layer transformer $\mathbf{w}^k_H, \mathbf{w}^k_L, \boldsymbol{W}_{KQ}, \boldsymbol{W}_{OV}$. Positional pattern in $\mathbf{w}^k_H$, diagonal pattern in $\boldsymbol{W}_{KQ}$ and the similarity between $\boldsymbol{W}_{OV}$ and $\log \pi$ can be observed in Fig. 3. To fully understand why the transformer can select causal structure, we analyze it theoretically.

**Takeaway 1**. Transformer formulated by Eq. (5) effectively identifies latent causal parents in-context (Fig. 2) and learns highly structural parameters aligned with the task (Fig. 3).

## 3.2 CONSTRUCTED TRANSFORMERS IMPLEMENT STATISTICAL ALGORITHM

Based on the patterns observed in experiments (Fig. 3), we make the following assumptions for the transformer defined by Eq. (5):

$$\mathbf{w}^k_H[h] = \beta \begin{cases} +1, & h = 0, \\ -1, & h \in [\pm H] \setminus 0, \end{cases} \quad \exists\, k' \in [L] \text{ s.t. } \mathbf{w}^k_L[l] = \beta \begin{cases} +1, & l = k', \\ -1, & l \in [L] \setminus k', \end{cases}$$

$$\boldsymbol{W}_{KQ} = \begin{bmatrix} \boldsymbol{W} & 0_{d\times d} & \cdots & 0_{d\times d} \\ 0_{d\times d} & \boldsymbol{W} & \cdots & 0_{d\times d} \\ \vdots & \vdots & \ddots & \vdots \\ 0_{d\times d} & 0_{d\times d} & \cdots & \boldsymbol{W} \end{bmatrix}, \quad \sigma(\boldsymbol{W}_{OV}) = \pi, \tag{7}$$

where $\boldsymbol{W}$ is unknown parameter and for RPE, we assume that 0-th entry dominates $\mathbf{w}_H^k$ and one element $k'$ of $\mathbf{w}_L^k$ dominates it: $\mathbf{w}_H^k[0] \gg \mathbf{w}_H^k[-0], \mathbf{w}_L^k[k'] \gg \mathbf{w}_L^k[-k']$. Since the $K$ heads are identical up to their indices, we assume without loss of generality that the dominant entry of $\mathbf{w}_L^k$ occurs at position $k$, i.e., $\mathbf{w}_L^k[k] = \beta$ and we set $K = L$. The theorem below shows that, aligned with the above restriction, a constructed transformer can implement statistical algorithm for inferring the causal structure $\{pa(h)\}$ hidden behind $\boldsymbol{x}_{1:H}^{1:L}$ and predicting $\boldsymbol{x}_h^{L+1} \sim \pi(\cdot \mid \boldsymbol{x}_{pa(h)}^{L+1})$:

**Theorem 1.** *Under the restriction by Eq. (7), the transformer $\boldsymbol{f}_\theta$ is parameterized by $\theta \in \{(\beta, \boldsymbol{W})\}$. Then for $\boldsymbol{f}_\theta$ with $\boldsymbol{W} = \log \pi$ in Eq. (7), its second attention layer $\mathcal{A}^{(2)}(\mathcal{H}; \theta)$ approximates Bayesian Model Averaging (see Lemma 1):*

$$\lim_{\beta \to \infty} \mathcal{A}_{h \to \cdot}^{(2)}(\mathcal{H}; \theta) = \lim_{\beta \to \infty} \sigma(\mathcal{M}(\tilde{\mathcal{A}}_{h \to \cdot}^{(2)}(\mathcal{H}; \theta))) = \sigma(\hat{\boldsymbol{p}}_{\text{BMA}}^{h,L}). \tag{8}$$

*Further, the transformer's prediction of the last example $\boldsymbol{x}_{1:H}$ with $L$ context examples converges to the true conditional distribution given the causal parent guaranteed by Theorem 2:*

$$\lim_{\beta, L \to \infty} \boldsymbol{f}_\theta(\cdot \mid \mathcal{H}_h^L) = \pi(\cdot \mid \boldsymbol{x}_{pa(h)}), \quad \forall h \in [H]. \tag{9}$$

*Proof Sketch.* Figure 4 gives an overview of the construction: in the first RPE attention layer, each head from Eq. (7) is assigned to retrieve one historical copy of the same token $\boldsymbol{x}_h$, so that concatenating $L$ heads recovers past $L$ observations $\boldsymbol{x}_h^{1:L}$. In the second layer, with the condition in Eq. (7), the attention score between tokens $(h, h')$ reduces to a bilinear form $\hat{\boldsymbol{p}}_{h'}^h(\boldsymbol{W}) = \sum_l \boldsymbol{x}_{h'}^{l\top} \boldsymbol{W} \boldsymbol{x}_h^l$, which by $\boldsymbol{W} = \log \pi$ coincides with the BMA score $\hat{\boldsymbol{p}}_{\text{BMA}}^{h,L} = \sum_{l \in [L]} \log \pi(\boldsymbol{x}_h^l | \boldsymbol{x}_{h'}^l)$. With the causal mask, the softmax attention exactly matches the parent-selection distribution in BMA. By the theoretical guarantee of causal token selection (Theorem 2), OV matrix $\boldsymbol{W}_{OV}$ receives the correct parent $\boldsymbol{x}_{pa(h)}^{L+1}$ and makes prediction for $\pi(\cdot \mid \boldsymbol{x}_{pa(h)}^{L+1})$. The full technical proof is deferred to Appendix C.1. $\square$

D'Angelo et al. (2025) also considers an in-context causal learning task. With minor modification of the above construction, we can show that transformers can implement BMA for that easier task.[1]

> **Takeaway 2.** Two-layer transformers can explicitly implement BMA for causal token selection.

## 3.3 WHAT ALGORITHM DOES THE TRANSFORMER LEARN?

Although we have constructed a transformer implementing the BMA algorithm, what do transformers actually learn after training? Since the core lies in the attention weight $\mathcal{A}^{(2)}$ with $\boldsymbol{W}_{KQ}$ which recovers graph structures, we next analyze its characteristics in detail. In the following, we use $\boldsymbol{W}_{\text{tf}}$ to denote the trainable submatrix in Eq. (7). We first define the parent selection metric from cross-entropy loss, which quantitively shows the accuracy of models to predict parent indices:

$$\mathcal{L}_{pa}(\mathcal{A}^{(2)}(\boldsymbol{x}_{1:H}^{1:L+1}; \mathcal{G}), \mathcal{G}) = -\frac{1}{H} \sum_{h \in [H]} \boldsymbol{e}_{pa(h)}^\top \log \mathcal{A}_{h\cdot}^{(2)} = -\frac{1}{H} \sum_{h \in [H]} \log \mathcal{A}_{h \to pa(h)}^{(2)}, \tag{10}$$

where $\mathcal{A}^{(2)}$ is considered as an algorithm for predicting parent $\boldsymbol{e}_{pa(h)}$ given the input $\boldsymbol{x}_{1:H}^{1:L+1}$ and we have $\mathcal{L}_{pa}(\mathcal{A}_{\text{BMA}}, \mathcal{G}) = -\frac{1}{H} \sum_{h \in [H]} \boldsymbol{e}_{pa(h)}^\top \sigma(\hat{\boldsymbol{p}}_h(\log \pi))$ by Eq. (2) where $\hat{\boldsymbol{p}}_h(\boldsymbol{W}) = \sum_l \boldsymbol{x}_{1:h-1}^{l\top} \boldsymbol{W} \boldsymbol{x}_h^l$. We visualize this metric $\mathcal{L}_{pa}$ during the transformer training process in Fig. 8 and compare it with BMA's. We observe that the transformer's parent selection loss decreases during training while remaining above the loss of BMA, gradually approaching it.

**Generalized parent selection with varying size $L'$.** We further test how well the transformer and BMA generalize in parent selection under different sample sizes $L'$: since $\mathcal{A}^{(2)}$ and $\mathcal{A}_{\text{BMA}}$ are formulated via $\hat{\boldsymbol{p}}^L = \sum_{l \in [L]} \boldsymbol{x}_{1:h-1}^{l\top} \boldsymbol{W} \boldsymbol{x}_h^l$, we vary the number of demonstrations as a set of $L'$, and finally compute $\hat{\boldsymbol{p}}^{L'}, \mathcal{A}_{h\cdot}^{L'}$, and the parent selection loss $\mathcal{L}_{\text{pa}}^{L'}(\boldsymbol{W})$ with $\boldsymbol{W} \in \{\boldsymbol{W}_{\text{tf}}^{(L)}, \log \pi\}$.

---

[1] A comprehensive comparison with D'Angelo et al. (2025), highlighting both similarities and distinctions, is included in Appendix A (Related Work).

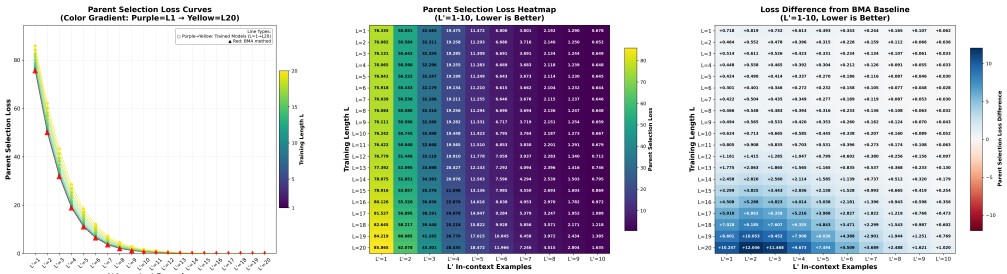

Figure 5: Generalization of parent selection loss $\{\mathcal{L}_{\mathrm{pa}}^{L'}\}$ for transformers trained with $L \in \{1, \ldots, 20\}, d = 10$, and $H = 15$ with first layer fixed as constructed.

From Fig. 5, we observe that: 1) across different test sizes $L'$, the trained transformers achieve performance close to BMA (loss differences mostly within a small margin); 2) models with smaller training length $L$ generalizes better, with loss curves approaching BMA more closely; 3) for a model with fixed training size $L$, the parent loss decreases rapidly as $L'$ increases, converging toward zero. The above results show that the trained transformers have comparable performance to BMA.

**Parameter Verification.** Beyond behavioral agreement, a crucial question is: *can we interpret the parameters of transformers with the aligned BMA inference?* We evaluate the similarity between the trained weight $\boldsymbol{W}_{\mathrm{tf}}^{(L)}$ and the theoretical BMA parameter $\boldsymbol{W} = \log \pi$. As row sum of $\pi$ equals 1, we first check whether $\sigma(\boldsymbol{W}_{\mathrm{tf}}) = \pi$. Note that, $\sigma(\boldsymbol{W}_{\mathrm{tf}}) = \pi \iff \boldsymbol{W}_{\mathrm{tf}} = \log \pi + \boldsymbol{b}\mathbf{1}^\top$, where $\boldsymbol{b}\mathbf{1}^\top$ denotes a row-wise shift canceled out by the row-softmax $\sigma$. This softmax of $\sigma(\boldsymbol{W}_{\mathrm{tf}})$ provides a reasonable way to normalize KQ matrix making its scale comparable to $\pi$ with scale $[0, 1]$. However, the empirical results in Fig. 6 (first three subfigures) show that misalignments exist between $\sigma(\boldsymbol{W}_{\mathrm{tf}})$ and $\pi$. From the view of the output prediction, we can see the attention mechanism $\sigma(\boldsymbol{v}_{1:h-1}^\top \boldsymbol{W} \boldsymbol{v}_h)$ introduces a degree of freedom on the **column** of $\log \pi$ rather than the **row**:

**Proposition 1** (Invariance). *If the columns of $\boldsymbol{W}_{\mathrm{tf}}$ differ from those of $\log \pi$ by a column-wise shift, i.e., $\boldsymbol{W}_{\mathrm{tf}} = \log \pi + \mathbf{1}\boldsymbol{a}^\top, \forall \boldsymbol{a} \in \mathbb{R}^d$, then we have the same output of attention module and BMA:*

$$\sigma\left(\sum_l \boldsymbol{x}_{1:h-1}^{l\top} \boldsymbol{W}_{\mathrm{tf}} \boldsymbol{x}_h^l\right) = \sigma\left(\sum_l \boldsymbol{x}_{1:h-1}^{l\top} \log \pi \boldsymbol{x}_h^l\right).$$

*Further, if Markov chain $\boldsymbol{x}_{1:H}$ is stationary and $\boldsymbol{W}_{\mathrm{tf}} = \log \pi + \mathbf{1}\boldsymbol{a}^\top + \boldsymbol{b}\mathbf{1}^\top, \boldsymbol{a}, \boldsymbol{b} \in \mathbb{R}^d$, the above conclusion also holds asymptotically as $L \to \infty$. See the detailed proof in Appendix C.3.*

This proposition indicates that instead of checking whether $\sigma(\boldsymbol{W}_{\mathrm{tf}}) = \sigma(\log \pi)$, we should check whether $\sigma_{\mathrm{col}}(\boldsymbol{W}_{\mathrm{tf}}) = \sigma_{\mathrm{col}}(\log \pi)$ as it verifies if $\boldsymbol{W}_{\mathrm{tf}} = \log \pi + \mathbf{1}\boldsymbol{a}^\top$. We follow this to evaluate the discrepancy between $\sigma_{\mathrm{col}}(\boldsymbol{W}_{\mathrm{tf}})$ and $\sigma_{\mathrm{col}}(\log \pi)$. As illustrated in Fig. 6, the deviation remains small, with *column softmax error*

$$\frac{1}{d}\|\sigma_{\mathrm{col}}(\boldsymbol{W}_{\mathrm{tf}}) - \sigma_{\mathrm{col}}(\log \pi))\|_1 < 0.05. \tag{11}$$

This alignment also holds across various training model sizes $d \in \{10, 30, 50\}$, as shown in Fig. 10. It confirms that trained transformers did approximate $\log \pi + \mathbf{1}\boldsymbol{a}^\top$, *which is equivalent to implement BMA method via Proposition 1.* Taken together with theoretical construction and empirical results, we conclude that transformers can and do implement BMA for in-context causal structure learning.

**Takeaway 3.** Transformers with trainable $\boldsymbol{W}_{\mathrm{tf}}$ closely approximate BMA in causal token selection (Fig. 5) and learn parameters which explicitly implement BMA method (Fig. 6).

**Robustness to Model Architecture.** A natural question might arise: *is the specific RPE-based construction essential for this mechanism?* We show in Appendix G that standard disentangled transformers with absolute positional embeddings are also theoretically capable of implementing BMA. Empirically, we trained such disentangled models and standard transformers with FFNs (details in Appendices G–I). Despite the increased parameter complexity, *these models converge to the same attention patterns and achieve parent selection performance comparable to BMA* (Fig. 19). This confirms that the interpreted parent selection mechanism is a general solution found by the optimization process, independent of the simplified RPE parameterization.

Figure 6: Parameter-level comparison between transformer $W_{\tt tf}$ and BMA $\log \pi$. Here, $W_{\tt tf}$ denotes the diagonal block being trained. Trained with $d = 20$, $H = 50$, $L = 3$, and 2048 training steps while $W_{OV}$ remained fixed. (See results with $W_{OV}$ and $W_{\tt tf}$ both trainable in Fig. 9.)

### 3.4 THEORETICAL GUARANTEE OF LEARNED ALGORITHM

Beyond identifying what algorithm a trainable transformer adopts, we further establish the theoretical understanding of why transformers can select the correct causal token via information-theoretic principles. Our approach follows Nichani et al. (2024), which leverages *Mutual Information* together with the data's inherent property of the *Data Processing Inequality* (DPI). In contrast to their gradient-based proof, we show that transformers can exploit this property **directly in context**. Moreover, our analysis generalizes the $\chi^2$–mutual information framework of Nichani et al. (2024) to the setting reducible to classical mutual information, by exploiting the information-theoretic structure characterized in Lemma 3 and 4. Finally, our proof of Theorem 2 applies to finite-horizon Markov chains whose marginal distributions are not assumed to be stationary. [2]

We leverage the data processing inequality regarding classical mutual information $I$ and $\chi^2$-mutual information $I_{\chi^2}$ (see Definition 1 in Appendix C.10) to establish the identifiability of causal structures. $I, I_{\chi^2}$ can be uniformly derived from $f$-divergence which helps to prove DPI for generalized $f$-mutual information $I_f$. These information metrics reveal an essential property in data:

**Lemma 2** (DPI. Theorem 3.9 and 7.16 in Polyanskiy & Wu (2023)). *If random variables* $x \to y \to z$, *i.e., satisfy the Markov property* $p(x, y, z) = p(x)p(y|x)p(z|y)$, *then we have* $I_f(y; z) \geq I_f(x; z)$. *Further, for classical mutual information,* $I(x; z) = I(y; z)$ *iff* $I(x; y|z) = 0$ *iff* $x \to z \to y$.

In our Markov chain setting with random causal structures, $x_{h'} \to x_{pa(h)} \to x_h$ holds, which guarantees the DPI. Further, following Nichani et al. (2024), we develop a stricter version of DPI for classical mutual information in our setting, guaranteeing the selection of unique parents.

**Lemma 3.** *Let* $x_{1:H}$ *be a Markov chain with transition kernel* $\pi$ *and causal structure* $\mathcal{G}$. *Suppose there exist* $\gamma, \delta \in (0, 1)$ *such that* $\min_{s,s' \in \mathcal{V}} \pi(s'|s) \geq \gamma/|\mathcal{V}|$ *and the marginal distribution* $\min_s \mathbb{P}(x_h = s) > \delta$. *Then there exists* $\alpha \leq 1 - \delta\gamma < 1$ *such that:*

$$I(x_h; x_{h'}) \leq \alpha \cdot I(x_h; x_{pa(h)}), \quad \forall h' < h, \ h' \neq pa(h).$$

The aforementioned assumption is the transition lower bound condition in Nichani et al. (2024), which ensures a uniform minorization of the transition kernel and implies the strong DPI; see Appendix C.4 for details. Building on the above lemma, we establish the following result.

**Lemma 4.** *For a Markov chain* $x_{1:H}$ *with causal structure* $\mathcal{G}$ *satisfying the conditions of Lemma 3, suppose further that* $I(x_h; x_{pa(h)}) > 0$. *Then*

$$\mathbb{E}_{\boldsymbol{X}}[\log \pi(x_h|x_{pa(h)})] > \mathbb{E}_{\boldsymbol{X}}[\log \pi(x_h|x_{h'})], \quad \forall h' \neq pa(h).$$

This establishes the DPI in the expected log-likelihood (MLE) form. See Appendix C.5 for details.

Then the following theorem shows that the attention weights of transformers leverage in-context examples to instantiate the above criterion for causal structure. The proof is deferred to Appendix C.6.

**Theorem 2.** *Under the conditions of Lemma 4, consider the transformer constructed in Theorem 1, which implements the BMA method. The attention weights* $\mathcal{A}_{h\cdot}^L = \mathcal{A}_{h\cdot}^{(2)}(\boldsymbol{x}_{1:H}^{1:L})$ *satisfy:*

$$\lim_{L \to \infty} \mathcal{A}_{h\cdot}^L = \lim_{L \to \infty} \sigma(\mathcal{M}(\hat{\boldsymbol{p}}^{h,L})) = \boldsymbol{e}_{pa(h)} \in \mathbb{R}^H, \quad \text{where } \hat{\boldsymbol{p}}_{h'}^{h,L} = \sum_{l=1}^L \log \pi(\boldsymbol{x}_h^l|\boldsymbol{x}_{h'}^l).$$

**Takeaway 4.** Information-theoretic analysis reveals that the parent selection exploits the conditional entropy, where the strong DPI guarantees the identifiability of the true causal parent.

---

[2] In other words, the guarantee does not require samples to be drawn from a stationary distribution and depends only on the finite-horizon transition structure.

### 3.5 Causal Structure in Training Dynamics

We further investigate the training dynamics of the transformer. We show that random causal structures embedded in the inputs will be recovered in the gradients of loss w.r.t. the core $W_{KQ}$ matrix:

**Theorem 3** (Informal). *Consider the transformer $f_\theta$ constructed as in Theorem 1 with trainable diagonal block $W$ of $W_{KQ}$ specified in Eq. (7) and trained with cross-entropy loss*

$$\mathcal{L}(\theta) = -\sum_{h=1}^{H} \mathbb{E}_{X,\mathcal{G}}[\log(f_\theta(x_h^{L+1}|\mathcal{H}) + \epsilon)] = -\sum_{h=1}^{H} \mathbb{E}_\mathcal{G}[\ell(\theta; h, \mathcal{G})], \qquad (12)$$

*where $\ell(\theta; h, \mathcal{G}) = \mathbb{E}_X[\log f_\theta(x_h^{L+1}|\mathcal{H})]$. Let $\hat{p}$ denote the intermediate parameter $\hat{p}(W) = \sum_l x_{1:h-1}^{l\top} W x_h^l$ from attention scores. If the Markov chain is stationary, i.e., $x_h \sim \mu^\pi, \forall h \in [H]$ and the initial $f_{\theta_0}$ with $W = 0$ outputs $\mu^\pi$ for any input, then the gradient at initialization statisfies:*

$$\frac{\partial \ell(\theta_0; h, \mathcal{G})}{\partial \hat{p}}\bigg|_{pa(h)} \geq \frac{\partial \ell(\theta_0; h, \mathcal{G})}{\partial \hat{p}}\bigg|_{h'}, \quad \forall h' \neq pa(h).$$

In derivation of the gradients, we show that these terms are highly related to $\chi^2$-mutual information, which reveals the numerical relation in gradients via data processing inequality. See Appendix C.7 for the detailed proof. This result explains how transformers extract meaningful structural information from data via intermediate parameters, rather than directly encoding this information to its parameters. Empirically, Fig. 11 shows the gradient of $\frac{\partial \ell(\theta_0)}{\partial \hat{p}}$ matches the latent causal structure.

> **Takeaway 5.** Gradient at initialization is able to recover the latent causal structure, driven by $\chi^2$-mutual information, which facilitates structural discovery in early training (Thm. 3 and Fig. 11).

## 4 Dynamical System Extension: From Discrete to Continuous

Further, we investigate the Markov chain in continuous space, where we examine the linear dynamical system with latent causal structures: $x_h = \rho A^\top x_{pa(h)} + \sqrt{1-\rho^2}\,\eta_h \in \mathbb{R}^d, \eta_h \sim \mathcal{N}(0, I_d)$. We initially train a transformer with RPE introduced in Eq. (5) on data generated from the dynamical system. Similar experiment results on attention weights $\mathcal{A}^{(1)}, \mathcal{A}^{(2)}$, and parameter visualizations can be found in Appendix Fig. 12, 13, 14, and 15. Visualization of RPE parameters is consistent with the construction in Eq. (7). Moreover, the attention weights $\mathcal{A}^{(2)}$ of the transformer yield accurate predictions of parent indices across various examples. Similar to the discrete case, we can define the transition $p(\cdot|\cdot)$ by $x_h|x_{pa(h)} \sim \mathcal{N}(\rho A^\top x_{pa(h)}, (1-\rho^2)I_d)$. Consequently, Eq. (2) similarly specifies the BMA formulation under the dynamical system setting. In this context, Lemma 4 remains valid and guarantees the asymptotic correctness of *BMA's* parent selection. To investigate *transformers'* mechanism of parent selection, we test the parent selection loss $\mathcal{L}_{pa}^{L'}$ of the transformer and BMA in the dynamical setting, where we set varying $L'$ in-context samples as introduced in Sec. 3.3. Fig. 17 demonstrates that the transformer with trainable $(W, W_{OV})$ achieves performance comparable to the BMA method when $L'$ approaches 20, while the loss $\mathcal{L}_{pa}^{L'}$ remains with a noticeable gap as $L'$ is small. We conjecture that the proposition below explains this discrepancy:

**Proposition 2** (Representation Limitation of Transformers). *Under the assumption Eq. (7), both the transformer and BMA take the unified form $\mathcal{A}_{h\to h'} = \sigma(\hat{p}^h)_{h'}$. In the DS setting, transformer logits are bilinear, $\hat{p}_{\mathtt{tf},h'}^h = \sum_l x_{h'}^{l\top} W_{\mathtt{tf}} x_h^l$, whereas BMA logits are $\hat{p}_{\mathtt{BMA},h'}^h = c_1 \sum_l x_{h'}^{l\top} A x_h^l + d\sum_l \|x_{h'}^l\|^2$ with $d \neq 0$. Then there exists no $W_{\mathtt{tf}}$ such that $\sigma(\hat{p}_{\mathtt{tf}}^h) = \sigma(\hat{p}_{\mathtt{BMA}}^h)$ holds for all DS samples $(x_{1:H}^{1:L+1}, \mathcal{G}) \sim P_\pi$. Hence, transformers under Eq. (7) cannot represent BMA in the DS setting. However, in the MC setting, $W_{\mathtt{tf}} = \log \pi$ matches the BMA formulation.*

See Appendix C.8 for detailed proof. For BMA, $(\hat{p}_{\mathtt{BMA}}^h)_{h'} = \sum_l \log p(x_h^l|x_{h'}^l)$. In dynamical systems, transition $p(\cdot|\cdot)$ involves not only cross but also quadratic terms. When the representation equation holds, substituting $x_h$ by $\rho A^\top x_{pa(h)} + \sqrt{1-\rho^2}\,\eta_h$ and using the independence of $\eta_h$, coefficients of $\eta_h$ must vanish, which requires $W_{\mathtt{tf}} = c_1 A$. But this $W_{\mathtt{tf}}$ fails to represent BMA's quadratic term. So no $W_{\mathtt{tf}}$ can yield $\mathcal{A}_{h\cdot}^{(2)} = \sigma(\mathcal{M}(\hat{p}^h(W_{\mathtt{tf}})))$ as BMA's.

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

## A    NOTATION AND RELATED WORK

**Notation.** We use $[h]$ to denote the set $\{1, 2, \ldots, h\}$. For causal structure, we use $pa(h)$ to represent the parent index of node $h$. If the stationary distribution of Markov chain $\boldsymbol{x}_h \sim \pi(\cdot | \boldsymbol{x}_{pa(h)})$ exists, then it is denoted by $\mu^\pi \in \Delta^d$. For the transformer, the input of a sequence of vectors is given by $\boldsymbol{x}_{1:T} \coloneqq [\boldsymbol{x}_1, \boldsymbol{x}_2, \ldots, \boldsymbol{x}_T] \in \mathbb{R}^{d \times T}$. Given the input, we denote the attention scores of a standard self-attention layer as $\boldsymbol{p}^t \coloneqq \boldsymbol{x}_{1:T}^\top \boldsymbol{W}_K^\top \boldsymbol{W}_Q \boldsymbol{x}_t \in \mathbb{R}^T$. However the causal mask $\mathcal{M}$ in the attention layer will lead to $\hat{\boldsymbol{p}}^t \coloneqq \boldsymbol{x}_{1:t-1}^\top \boldsymbol{W}_K^\top \boldsymbol{W}_Q \boldsymbol{x}_t \in \mathbb{R}^{t-1}, \sigma(\hat{\boldsymbol{p}}^t)_{t'} = \sigma(\mathcal{M}(\boldsymbol{p}^t))_{t'}, \forall t' \in [t-1]$. We do not distinguish between them in the proofs. For the matrix form of the attention, we use $\tilde{\mathcal{A}}$ and $\mathcal{A}$ to denote attention weights and scores, correspondingly, where we have $\hat{\boldsymbol{p}}_{t'}^t = \tilde{\mathcal{A}}_{t \to t'}$ and $\sigma(\mathcal{M}(\tilde{\mathcal{A}})) = \mathcal{A}$. In training, we use cross-entropy loss and MSE loss for Markov chain and dynamical system settings, respectively:

$$\mathcal{L}^{MC}(\theta) = -\frac{1}{H} \sum_{h=2}^{H} \boldsymbol{x}_h^{L+1\top} \log\left(\sigma(\boldsymbol{f}_{\mathtt{tf}_\theta}(\cdot \mid \mathcal{H}_h)) + \epsilon\right),$$

$$\mathcal{L}^{DS}(\theta) = -\frac{1}{H} \sum_{h=2}^{H} ||\boldsymbol{x}_h^{L+1} - \boldsymbol{f}_{\mathtt{tf}_\theta}(\cdot \mid \mathcal{H}_h)||_2^2, \tag{13}$$

where $\theta$ represents all trainable parameters and $\epsilon$ is a small value to avoid numerical issues with $\log$.

**Related Work.** A growing body of work studies the in-context learning (ICL) ability of transformers from different perspectives. One line of work understands ICL as a form of Bayesian inference, showing how the latent concept can be approximately inferred under restrictive theoretical assumptions (Xie et al., 2022; Zhang et al., 2025; Panwar et al., 2024). Another direction of research investigates how transformers can simulate standard algorithms, such as gradient descent on linear regression (Von Oswald et al., 2023; Ahn et al., 2023; Guo et al., 2024). While these works demonstrate the ICL power of transformers, they commonly assume i.i.d or uncorrelated input tokens. To move beyond i.i.d. assumptions, recent works investigate ICL with *correlated data*, particularly Markovian sequences (Edelman et al., 2024; Chen et al., 2024b; Makkuva et al., 2025). These settings provide insight into how transformers handle in-context learning with sequential dependencies, but typically focus on fixed dependency structures. In contrast, our work addresses *variable causal structures* that differ across prompts. Pioneering this direction, Nichani et al. (2024) demonstrated that transformers can encode fixed parent-child dependencies (e.g., bigrams) in Markov chains. D'Angelo et al. (2025) introduced *selective induction heads*, enabling transformers to identify the underlying Markovian order (or "lag") from a candidate set to learn this structure in-context. Our work generalizes this setting. While D'Angelo et al. (2025) focus on inferring a single structural parameter (the lag $k$) shared across the sequence, we tackle flexible structure inference where dependencies can vary arbitrarily for each position, effectively modeling latent trees rather than fixed-lag chains. D'Angelo et al. (2025) construct a three-layer transformer that asymptotically implements maximum likelihood estimation for their task, where the construction is verified via attention pattern visualization as well as KL divergence validation of next-token prediction targets. In our work, we theoretically derive a two-layer architecture that explicitly implements Bayesian Model Averaging (BMA) in-context. Empirically, we go beyond behavioral metrics and provide *parameter-level* verification, demonstrating that the trained weights directly encode the transition kernel. Furthermore, we provide theoretical understanding of in-context causal structure learning based on the Data Processing Inequality (DPI) and extend our analysis to continuous dynamical systems, revealing representational gaps not occuring in the discrete setting.

## B    CONCLUSION

In this work, we investigated the capability of transformers to infer and adapt to latent causal structures in-context, moving beyond the fixed dependency assumptions common in prior theoretical analyses. We proposed a novel framework based on Markov chains with randomly sampled causal dependencies, requiring the model to identify position-specific predecessor-successor relationships from context examples. First, we provided a constructive proof that a two-layer transformer with relative positional embeddings (RPE) can explicitly implement Bayesian Model Averaging (BMA). This demonstrates that the attention mechanism is theoretically capable of performing statistical

inference over structural uncertainty. Second, through extensive experiments and parameter-level analysis, we showed that trained transformers implement BMA method, which converge to the theoretical construction: the learned attention patterns recover the posterior probabilities of causal parents, and the weights explicitly recover the log-transition kernel of the underlying generative process. Third, we established information-theoretic guarantees using the Data Processing Inequality (DPI), which helps understand how the selection mechanism identifies causal structures in context, and showed that gradients at initialization recover these dependencies via $\chi^2$-mutual information. Finally, we extended our framework to continuous linear dynamical systems. While transformers continue to exhibit strong empirical performance in this setting, we identified the representational difference that prevents the exact implementation of BMA, unlike in the discrete case. Collectively, our findings offer a mechanistic explanation of how transformers perform in-context causal learning, highlighting their ability to act as statistical inference engines for both discrete and continuous data.

**Broader Implications.** Our findings support theoretical frameworks that model in-context learning as a statistical inference task (Xie et al., 2022). Distinct from "Induction Heads" which typically focus on copying fixed positional dependencies (Olsson et al., 2022; Nichani et al., 2024), we demonstrate a probabilistic setting where the model must infer a latent dependency structure that varies per example. This provides a mechanistic grounding for how LLMs adapt to flexible, context-dependent rules rather than relying solely on fixed n-gram statistics (Allen-Zhu & Li, 2025). Furthermore, this helps understand why LLMs demonstrate ICL capabilities on empirical tasks with "unstructured" language data (Wibisono & Wang, 2024), mirroring our setting where the transition mappings between words are fixed while the structural positions of a couple of words vary from input to input.

**Limitations and Future Work.** We acknowledge that real-world sequences often involve complex non-linear dynamics or hierarchical dependencies (e.g., context-free grammar) beyond the Markovian and dynamical systems studied here. However, our primary objective in this work was to prioritize mechanistic interpretability for Markov chain or dynamical system: explicitly characterizing how transformers infer latent structures in-context on these tasks. By focusing on these tractable settings, we were able to derive exact theoretical guarantees and provide parameter-level verification that the model implements Bayesian Model Averaging. We believe this explainable framework serves as a necessary foundation, and we leave the extension to more complex non-linear and hierarchical data-generating processes for future exploration.

## C  DEFINITIONS AND PROOFS

### C.1  PROOF OF THEOREM 1

*Proof.* By the condition of $(\mathbf{w}_H, \mathbf{w}_L)$ in Eq. (7), the attention score $\tilde{\mathcal{A}}^{(1)}_{t\to\cdot}$ of the query $\boldsymbol{x}_t = \boldsymbol{x}^{L+1}_h$ in the first layer is:

$$\tilde{\mathcal{A}}^{(1),k}_{t\to t'} = \mathbf{w}^k_H[h_t - h_{t'}] + \mathbf{w}^k_L[l_t - l_{t'}] = 2\beta \begin{cases} +1, & \text{if } h_t = h_{t'}, \ l_t - l_{t'} = k, \\ -1, & \text{if } h_t \neq h_{t'}, \ l_t - l_{t'} \neq k, \\ 0, & \text{otherwise,} \end{cases}$$

where $(h_{t'}, l_{t'})$ is the (token, example) position mapping of the input $\boldsymbol{x}_{t'}$. The output $\boldsymbol{u}^k_h = \text{Attn}^k_{\boldsymbol{x}_t \to \boldsymbol{x}_{1:T}}$ of the first attention layer will be calculated as:

$$\begin{aligned} \boldsymbol{u}^k_h &= \sigma(\mathbf{w}^k_H(h, \cdot) + \mathbf{w}^k_L(L+1, \cdot))\boldsymbol{x}^\top_{1:T} \\ &\xrightarrow{\beta \to \infty} \boldsymbol{x}^{l_k}_h{}^\top = (1_{[h_{t'} = h, l_{t'} = L+1-k]})_{t' \in [T]}\boldsymbol{x}^\top_{1:T}, \quad (l_k = L+1-k) \end{aligned} \tag{14}$$

where $k$-th attention head copies the corresponding token from $l_k$-th example for the query $\boldsymbol{x}^{L+1}_h$. By using the disentangled residual, the outputs of the $K$ heads ($K = L$) will be concatenated as:

$$\boldsymbol{v}_h = [\boldsymbol{u}^1_h, \ldots, \boldsymbol{u}^L_h], \text{ with } \boldsymbol{u}^k_h = \boldsymbol{x}^{L+1-k}_h \text{ by Eq. (14).}$$

For the second layer, with the diagonal condition of $\boldsymbol{W}_{KQ}$, the attention weight $\mathcal{A}^{(2)} \in \mathbb{R}^{H \times H}$ is given by:

$$\tilde{\mathcal{A}}^{(2)}_{h \to h'} = \boldsymbol{v}^\top_{h'} \boldsymbol{W}_{KQ} \boldsymbol{v}_h = \sum^L_{l=1} \boldsymbol{x}^{l}_{h'}{}^\top \boldsymbol{W} \boldsymbol{x}^l_h, \quad \mathcal{A}^{(2)} = \sigma(\mathcal{M}(\tilde{\mathcal{A}}^{(2)})) \in \mathbb{R}^{H \times H}, \tag{15}$$

where $\mathcal{M}$ is the causal mask enforcing $\mathcal{A}^{(2)}$ to be strictly lower-triangular. If we define the vector $\hat{\boldsymbol{p}}^h \in \mathbb{R}^{h-1}$ with $\hat{\boldsymbol{p}}^h_{h'} := \tilde{\mathcal{A}}^{(2)}_{h \to h'}$, we have $\forall h' \in [h-1]$:

$$\mathcal{A}^{(2)}_{h \to h'} = \sigma(\mathcal{M}_h(\tilde{\mathcal{A}}_{h \to \cdot}))_{h'} = \sigma(\hat{\boldsymbol{p}}^h)_{h'}, \quad \hat{\boldsymbol{p}}^h(\boldsymbol{W}) = \sum_l \boldsymbol{x}^{l\top}_{1:h-1} \boldsymbol{W} \boldsymbol{x}^l_h, \tag{16}$$

where $\mathcal{M}_h(\cdot)$ is the causal mask applied to row $h$, setting $\mathcal{M}_h(\boldsymbol{v})_{h'} = -\infty$ if $h' \geq h, \forall \boldsymbol{v} \in \mathbb{R}^H$. Then, we set $\boldsymbol{W}$ as $\log \pi$ (with $\log$ applied elementwise), which leads to:

$$\hat{\boldsymbol{p}}^h_{h'}(\log \pi) = \tilde{\mathcal{A}}^{(2)}_{h \to h'} = \sum_l \log \pi(\boldsymbol{x}^l_h | \boldsymbol{x}^l_{h'}). \tag{17}$$

Considering the form in Eq. (17), Lemma 1 shows that the BMA method of Eq. (1) has the same formulation:

$$\mathbb{P}(pa(h) = h' | \boldsymbol{x}^{1:L}_{1:H}) = \sigma(\hat{\boldsymbol{p}}^h(\boldsymbol{W} = \log \pi))_{h'}. \tag{18}$$

Combining Eq. (18) with the limiting behavior of the first layer in Eq. (14), we obtain the convergence result for parent selection as $\beta \to \infty$:

$$\lim_{\beta \to \infty} \mathcal{A}^{(2)}_{h \to h'}(\mathcal{H}; \theta) = \sigma(\hat{\boldsymbol{p}}^h(\boldsymbol{W} = \log \pi))_{h'} = \mathbb{P}(pa(h) = h' | \boldsymbol{x}^{1:L}_{1:H}).$$

We can define $\hat{\boldsymbol{p}}^{h,L}_{\text{BMA}} \in \mathbb{R}^H$ as the corresponding vector form for BMA whose $h'$-th entries are $-\infty$ if $h' \geq h$, otherwise $\hat{\boldsymbol{p}}^h$. Then we have:

$$\lim_{\beta \to \infty} \mathcal{A}^{(2)}_{h \to \cdot}(\mathcal{H}; \theta) = \sigma(\hat{\boldsymbol{p}}^{h,L}_{\text{BMA}}).$$

Furthermore, as guaranteed by the consistency of BMA (Theorem 2), as the sample size $L \to \infty$, the posterior estimation concentrates on the true parent $pa(h)$. Thus, the prediction of the token distribution converges in the limit $\beta, L \to \infty$ as:

$$\lim_{\beta, L \to \infty} \boldsymbol{f}_\theta(\cdot \mid \mathcal{H}) = \sigma\left(\boldsymbol{W}^\top_{OV} \sum_{h'} 1_{[h'=pa(h)]} \boldsymbol{x}^{L+1}_{h'}\right) = \pi(\cdot \mid \boldsymbol{x}^{L+1}_{pa(h)}). \qquad \square$$

In proving the theorem, we rely on Lemma 1 proved below, which illustrates the similarity between BMA and attention weights.

## C.2 PROOF OF LEMMA 1

*Proof.* Here we use $p(s|s')$ to denote $\mathbb{P}(\boldsymbol{x}_h = s | \boldsymbol{x}_{pa(h)} = s')$ for generality beyond discrete Markov chain. Based on Bayesian Theorem, it can be calculated by Eq. (1). Due to the Markovian property $p(\boldsymbol{x}_h | \boldsymbol{x}_{1:h-1}) = p(\boldsymbol{x}_h | \boldsymbol{x}_{pa(h)})$, the joint distribution of this chain $\mathrm{x}_{1:H}$ is:

$$\begin{aligned} p(\boldsymbol{x}_{1:H}) &= p(\boldsymbol{x}_1) \prod_{h=2}^H p(\boldsymbol{x}_h | \boldsymbol{x}_{1:h-1}) = p(\boldsymbol{x}_1) \prod_{h=2}^H p(\boldsymbol{x}_h | \boldsymbol{x}_{pa(h)}) \\ &= p(\boldsymbol{x}_1) \prod_{i \neq h} p(\boldsymbol{x}_i | \boldsymbol{x}_{pa(i)}) \cdot p(\boldsymbol{x}_h | \boldsymbol{x}_{pa(h)}) \end{aligned} \tag{19}$$

Here $pa(h), pa(i)$ in Eq. (19) are random index with prior: $pa(h) \sim \text{Uniform}([h-1])$. Conditioning on $pa(h) = h'$ in Eq. (1), we can substitue $pa(h)$ in Eq. (19) with $h'$. Since $\{pa(i)\}_{i \neq h}$ are random indices out of interests, these terms are eliminated:

$$\frac{\mathbb{P}(\boldsymbol{x}^{1:L}_{1:H} \mid \text{pa}(h) = h')}{\sum_{h'' \in [h-1]} \mathbb{P}(\boldsymbol{x}^{1:L}_{1:H} \mid \text{pa}(h) = h'')} = \frac{\prod_l \left( p(\boldsymbol{x}^l_1) \prod_{i \neq h} p(\boldsymbol{x}^l_i | \boldsymbol{x}^l_{pa(i)}) \cdot p(\boldsymbol{x}^l_h | \boldsymbol{x}^l_{h'}) \right)}{\sum_{h''} \prod_l \left( p(\boldsymbol{x}^l_1) \prod_{i \neq h} p(\boldsymbol{x}^l_i | \boldsymbol{x}^l_{pa(i)}) \cdot p(\boldsymbol{x}^l_h | \boldsymbol{x}^l_{h''}) \right)}, \tag{20}$$

which leads to:

$$\mathbb{P}(pa(h) = h' \mid \boldsymbol{x}^{1:L}_{1:H}) = \frac{\exp\left(\sum_{l \in [L]} \log p(\boldsymbol{x}^l_h | \boldsymbol{x}^l_{h'})\right)}{\sum_{h'' \in [h-1]} \exp\left(\sum_{l \in [L]} \log p(\boldsymbol{x}^l_h | \boldsymbol{x}^l_{h''})\right)} = \sigma\left(\hat{\boldsymbol{p}}^{h,L}(\log \boldsymbol{W}^P)\right)_{h'}, \tag{21}$$

where $\hat{\boldsymbol{p}}^{h,L}(\log \boldsymbol{W}^P) = \sum_l \boldsymbol{x}^{l\top}_{h'} \log \boldsymbol{W}^P \boldsymbol{x}^l_h$ and the matrix $\boldsymbol{W}^P = \pi$ is induced by transition kernel $P(s'|s) = \pi(s'|s)$ in the discrete Markov chain. $\qquad \square$

## C.3 PROOF OF PROPOSITION 1

*Proof.* First, suppose we have $\boldsymbol{W}_{\mathtt{tf}} = \log \pi + \boldsymbol{1}\boldsymbol{a}^{\top}$. The attention scores of the transformer are:

$$\hat{\boldsymbol{p}}_{\mathtt{tf}}^{h} = \sum_{l} \boldsymbol{x}_{1:h-1}^{l\top} \boldsymbol{W}_{\mathtt{tf}} \boldsymbol{x}_{h}^{l} = \sum_{l} \boldsymbol{x}_{1:h-1}^{l\top} \log \pi \boldsymbol{x}_{h}^{l} + \sum_{l} \boldsymbol{x}_{1:h-1}^{l\top} \boldsymbol{1}\boldsymbol{a}^{\top} \boldsymbol{x}_{h}^{l}.$$

For the second term, since $\{\boldsymbol{x}_{h'}\}$ are one-hot, we have:

$$\sum_{l} \boldsymbol{x}_{1:h-1}^{l\top} \boldsymbol{1}\boldsymbol{a}^{\top} \boldsymbol{x}_{h}^{l} = \boldsymbol{1}_{h-1} \boldsymbol{a}^{\top} \left( \sum_{l} \boldsymbol{x}_{h}^{l} \right) = c(\boldsymbol{a}, h) \boldsymbol{1}_{h-1},$$

where $c(\boldsymbol{a}, h) = \boldsymbol{a}^{\top} \left( \sum_{l} \boldsymbol{x}_{h}^{l} \right)$ is a constant with fixed index $h$. And by softmax operation, we have:

$$\sigma(\hat{\boldsymbol{p}}_{\mathtt{tf}}^{h}) = \sigma \left( \sum_{l} \boldsymbol{x}_{1:h-1}^{l\top} \log \pi \boldsymbol{x}_{h}^{l} + c(\boldsymbol{a}, h) \boldsymbol{1}_{h-1} \right)$$

$$= \sigma \left( \sum_{l} \boldsymbol{x}_{1:h-1}^{l\top} \log \pi \boldsymbol{x}_{h}^{l} \right)$$

$$= \sigma(\hat{\boldsymbol{p}}_{\mathtt{BMA}}^{h}(\log \pi)),$$

where the first step shows that the softmax eliminates the constant term, and the last equality is from Lemma 1. This shows that the transformer with $\boldsymbol{W}_{\mathtt{tf}} + \boldsymbol{1}\boldsymbol{a}^{\top}$ gives the same prediction as BMA's.

Further, suppose $\boldsymbol{W}_{\mathtt{tf}} = \log \pi + \boldsymbol{1}\boldsymbol{a}^{\top} + \boldsymbol{b}\boldsymbol{1}^{\top}$. If we have $\boldsymbol{x}_{h} \sim \mu^{\pi}, \forall h \in [H]$, then we can prove the term from $\boldsymbol{b}\boldsymbol{1}^{\top}$ also forms a constant vector asymptotically:

$$\hat{\boldsymbol{p}}_{\mathtt{tf}}^{h} = \sum_{l} \boldsymbol{x}_{1:h-1}^{l\top} \boldsymbol{W}_{\mathtt{tf}} \boldsymbol{x}_{h}^{l}$$

$$= \sum_{l} \boldsymbol{x}_{1:h-1}^{l\top} \log \pi \boldsymbol{x}_{h}^{l} + \sum_{l} \boldsymbol{x}_{1:h-1}^{l\top} \boldsymbol{1}\boldsymbol{a}^{\top} \boldsymbol{x}_{h}^{l} + \sum_{l} \boldsymbol{x}_{1:h-1}^{l\top} \boldsymbol{b}\boldsymbol{1}^{\top} \boldsymbol{x}_{h}^{l}$$

$$= \sum_{l} \boldsymbol{x}_{1:h-1}^{l\top} \log \pi \boldsymbol{x}_{h}^{l} + c(\boldsymbol{a}, h) \boldsymbol{1}_{h-1} + \sum_{l} \boldsymbol{x}_{1:h-1}^{l\top} \boldsymbol{b}.$$

For each term of $\sum_{l} \boldsymbol{x}_{1:h-1}^{l\top} \boldsymbol{b}$, we have:

$$\frac{1}{L} \sum_{l} \boldsymbol{x}_{h'}^{l\top} \boldsymbol{b} = \frac{1}{L} \sum_{l} \sum_{s \in [d]} \boldsymbol{1}_{[\boldsymbol{x}_{h'}^{l} = s]} \boldsymbol{b}_{s}$$

$$\overset{L \to \infty}{\to} \mathbb{E} \left[ \sum_{s \in [d]} \boldsymbol{1}_{[\boldsymbol{x}_{h'} = s]} \boldsymbol{b}_{s} \right] = \sum_{s \in [d]} \mathbb{P}(\boldsymbol{x}_{h'} = s) \boldsymbol{b}_{s}$$

$$= \mu^{\pi\top} \boldsymbol{b},$$

where $\mu^{\pi\top} \boldsymbol{b}$ is a constant $d(\boldsymbol{b})$ w.r.t. $h'$. Using the same technique in Theorem 2 through division to eliminate this term which goes to infinity, we have the desired result:

$$\lim_{L \to \infty} \sigma(\hat{\boldsymbol{p}}_{\mathtt{tf}}^{h,L}) = \lim_{L \to \infty} \sigma(\hat{\boldsymbol{p}}_{\mathtt{BMA}}^{h,L}(\log \pi)).$$

$\square$

## C.4 PROOF OF LEMMA 3

*Proof.* Our target is to show there exists $\alpha < 1$ s.t. $I(\mathrm{x}_{i}; \mathrm{x}_{j}) \leq \alpha \cdot I(\mathrm{x}_{i}; \mathrm{x}_{pa(i)})$ for any $j < i, j \neq pa(i)$. Since $i$ and $j$ are in the same tree of the causal structure $\mathcal{G}$, we use $p(i, j)$ to denote their lowest common ancestor (LCA).

Generally, there are *two cases* for the relation between node $i$ and $j$, as shown in Fig. 7. In the following, we *first* show how we can get the contraction from $I(\mathrm{x}_{i}; \mathrm{x}_{j})$ to $I(\mathrm{x}_{i}; \mathrm{x}_{p(i,j)})$. If $j$ is not an ancestor of $i$, then this contraction is non-trivial and from $I(\mathrm{x}_{i}; \mathrm{x}_{p(i,j)})$ to $I(\mathrm{x}_{i}; \mathrm{x}_{pa(i)})$, we can use the weak version of data processing inequality to bridge them. If $j$ is an ancestor of $i$, we further define the reverse transition kernel and adopt similar technique to derive the contraction.

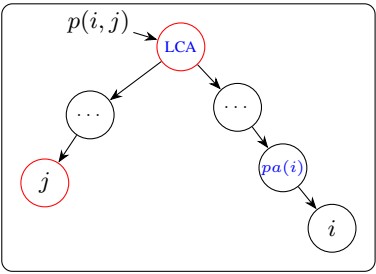
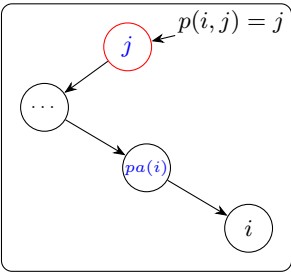

(a) $j$ is *not* an ancestor of $i$ (branching).    (b) $j$ *is* an ancestor of $i$ (ancestor chain).

Figure 7: Two cases of the relation between $i$ and $j$ in a Markov-tree. Directed arrows indicate the dependency direction (ancestor $\to$ descendant).

Recall that the definition of mutual information is

$$I(x_i; x_j) = \sum_{s,s'} \mathbb{P}(x_i = s, x_j = s') \log \frac{\mathbb{P}(x_i = s, x_j = s')}{\mathbb{P}(x_i = s)\mathbb{P}(x_j = s')} \tag{22}$$

$$= \sum_s \mathbb{P}(x_i = s) \cdot \mathrm{KL}(\mathbb{P}(x_j = \cdot | x_i = s) || \mathbb{P}(x_j = \cdot)), \ \ \forall j < i. \tag{23}$$

Let $x$ denote the conditional probability $\mathbb{P}(x_{p(i,j)} = \cdot | x_i = s)$. We can see the distribution $\pi^{d(j,p(i,j))} \circ x$ is:

$$\pi^{d(j,p(i,j))} \circ x(s') = \sum_{s^*} \pi^{d(j,p(i,j))}(s'|s^*) \cdot x(s^*) \tag{24}$$

$$= \sum_{s^*} \mathbb{P}(x_j = s' | x_{p(i,j)} = s^*) \cdot \mathbb{P}(x_{p(i,j)} = s^* | x_i = s) \tag{25}$$

$$= \mathbb{P}(x_j = s' | x_i = s), \tag{26}$$

where the last step comes from Markov property. Let $\mu^i$ denote the marginal distribution $\mathbb{P}(x_i = \cdot)$. By similar calculation, we can obtain:

$$\pi^{d(j,p(i,j))} \circ \mu^{p(i,j)}(s') = \sum_{s^*} \pi^{d(j,p(i,j))}(s'|s^*) \cdot \mu^{p(i,j)}(s^*) \tag{27}$$

$$= \sum_{s^*} \mathbb{P}(x_j = s' | x_{p(i,j)} = s^*) \cdot \mathbb{P}(x_{p(i,j)} = s^*) \tag{28}$$

$$= \mathbb{P}(x_j = s'). \tag{29}$$

Let $k = d(j, p(i,j))$. Hence, we get the following equation for the KL divergence term in Eq. (23):

$$\mathrm{KL}(\mathbb{P}(x_j = \cdot | x_i = s) || \mu^j) = \mathrm{KL}(\pi^k \circ x || \pi^k \circ \mu^{p(i,j)}).$$

For the term above, we apply Lemma 5 which requires $\max_{s \in \mathcal{V}} \pi(s'|s) > 0, \forall s' \in \mathcal{V}$ satisfied by our assumption and gives the contraction for the mutual information with contractive coefficient $\alpha_0 := \frac{1}{2} \max_{i \neq k} \| \pi(\cdot|j) - \pi(\cdot|k) \|_1 \leq 1 - \gamma < 1$, as shown by Lemma 6. This leads to:

$$I(x_i; x_j) \leq \sum_s \mathbb{P}(x_i = s) \Big( \alpha_0^k \cdot \mathrm{KL}(\mathbb{P}(x_{p(i,j)} = \cdot | x_i = s) || \mathbb{P}(x_{p(i,j)} = \cdot)) \Big) \tag{30}$$

$$= \alpha_0^k \cdot \sum_s \mathbb{P}(x_i = s) \cdot \mathrm{KL}(\mathbb{P}(x_{p(i,j)} = \cdot | x_i = s) || \mathbb{P}(x_{p(i,j)} = \cdot)) \tag{31}$$

$$= \alpha_0^k \cdot I(x_i; x_{p(i,j)}), \tag{32}$$

where $k = d(j, p(i,j))$. This shows the contraction from $I(x_i; x_j)$ to $I(x_i; x_{p(i,j)})$. One may notice that the key of the proof is the function of transition $\pi$ on the distribution leading to the contraction in KL divergence. For bridging $I(x_i; x_{p(i,j)})$ to $I(x_i; x_{pa(i)})$, we define the reverse transition matrix to follow the above proof technique. Different from the above, since we don't assume the stationary distribution of $x_h$, the reverse transition is time-inhomogeneous.

For the convenience, we adopt $l$ to denote the distance $d(i, p(i, j))$ between node $i$ and $p(i, j)$. Then since $p(i, j)$ is an ancestor of node $i$, we use $pa^l(i)$ to denote it. Similarly, we use $pa^t(i)$ to denote the $t$-step parent of node $i$. By Bayes' rule, we have:

$$\mathbb{P}(\mathbf{x}_{pa^t(i)} = s | \mathbf{x}_{pa^{t-1}(i)} = s') = \frac{\mathbb{P}(\mathbf{x}_{pa^{t-1}(i)} = s' | \mathbf{x}_{pa^t(i)} = s)\mathbb{P}(\mathbf{x}_{pa^t(i)} = s)}{\mathbb{P}(\mathbf{x}_{pa^{t-1}(i)} = s')}, \tag{33}$$

$$= \frac{\pi(s'|s)\mathbb{P}(\mathbf{x}_{pa^t(i)} = s)}{\mathbb{P}(\mathbf{x}_{pa^{t-1}(i)} = s')}. \tag{34}$$

From the above calculation, we denote the time-$t$ reverse transition kernel $\tilde{\pi}_t \in \mathbb{R}^{d \times d}$ as:

$$\tilde{\pi}_t(s|s') := \mathbb{P}(\mathbf{x}_{pa^t(i)} = s | \mathbf{x}_{pa^{t-1}(i)} = s'). \tag{35}$$

Recall that the mutual information $I(\mathbf{x}_i; \mathbf{x}_{pa^l(i)})$ is:

$$I(\mathbf{x}_i; \mathbf{x}_{pa^l(i)}) = \sum_{s'} \mathbb{P}(\mathbf{x}_i = s') \cdot \mathrm{KL}\Big(\mathbb{P}(\mathbf{x}_{pa^l(i)} = \cdot | \mathbf{x}_i = s') || \mathbb{P}(\mathbf{x}_{pa^l(i)} = \cdot))\Big). \tag{36}$$

Via conditional independence of the Markov chain, we can show:

$$\mathbb{P}(\mathbf{x}_{pa^l(i)} = \cdot | \mathbf{x}_i = s') = \tilde{\pi}_l \circ \cdots \circ \tilde{\pi}_2 \circ \mathbb{P}(\mathbf{x}_{pa(i)} = \cdot | \mathbf{x}_i = s'). \tag{37}$$

Suppose $\tilde{\pi}_{t-1} \circ \cdots \circ \mathbb{P}(\mathbf{x}_{pa(i)} = \cdot | \mathbf{x}_i = s') = \mathbb{P}(\mathbf{x}_{pa^{t-1}(i)} = \cdot | \mathbf{x}_i = s')$ holds. For time $t$:

$$\tilde{\pi}_t \circ \cdots \circ \mathbb{P}(\mathbf{x}_{pa(i)} = \cdot | \mathbf{x}_i = s')_s = \tilde{\pi}_t \circ \mathbb{P}(\mathbf{x}_{pa^{t-1}(i)} = \cdot | \mathbf{x}_i = s')_s, \tag{38}$$

$$= \sum_{s^*} \mathbb{P}(\mathbf{x}_{pa^t(i)} = s | \mathbf{x}_{pa^{t-1}(i)} = s^*)\mathbb{P}(\mathbf{x}_{pa^{t-1}(i)} = s^* | \mathbf{x}_i = s'), \tag{39}$$

$$= \mathbb{P}(\mathbf{x}_{pa^t(i)} = s | \mathbf{x}_i = s'). \tag{40}$$

Similarly, we can prove:

$$\mathbb{P}(\mathbf{x}_{pa^l(i)} = \cdot) = \tilde{\pi}_l \circ \cdots \circ \tilde{\pi}_2 \circ \mathbb{P}(\mathbf{x}_{pa(i)} = \cdot). \tag{41}$$

From the assumption of transition kernel and marginal distribution, $\tilde{\pi}_t$ defined in Eq. (35) is a valid transition kernel and we can lower bound $\tilde{\pi}_t$ by:

$$\tilde{\pi}_t(s|s') = \frac{\pi(s'|s)\mathbb{P}(\mathbf{x}_{pa^t(i)} = s)}{\mathbb{P}(\mathbf{x}_{pa^{t-1}(i)} = s')} \geq \frac{\delta\gamma}{|\mathcal{V}|}. \tag{42}$$

With the above condition, we can apply Lemma 5 and 6 again to get the contractive coefficient $\alpha_1 \leq 1 - \delta\gamma$ such that:

$$\mathrm{KL}(\tilde{\pi}_t \circ x || \tilde{\pi}_t \circ y) \leq \alpha_1 \cdot \mathrm{KL}(x||y), \ \forall t \in [l]. \tag{43}$$

With these ingredients, we obtain:

$$I(\mathbf{x}_i; \mathbf{x}_{pa^l(i)}) \leq \alpha_1^{l-1} \cdot I(\mathbf{x}_i; \mathbf{x}_{pa(i)}) = \alpha_1^{d(p(i,j), pa(i))} \cdot I(\mathbf{x}_i; \mathbf{x}_{pa(i)}). \tag{44}$$

Define $\alpha = \max\{\alpha_0, \alpha_1\}$. Since $\gamma, \delta \in (0, 1)$, $\alpha \leq 1 - \delta\gamma < 1$. And we have:

$$I(\mathbf{x}_i; \mathbf{x}_j) \leq \alpha^{d(j, p(i,j))} \cdot I(\mathbf{x}_i; \mathbf{x}_{p(i,j)}), \tag{45}$$

$$\leq \alpha^{d(j, p(i,j)) + d(p(i,j), pa(i))} \cdot I(\mathbf{x}_i; \mathbf{x}_{pa(i)}), \tag{46}$$

$$= \alpha^{d(j, pa(i))} \cdot I(\mathbf{x}_i; \mathbf{x}_{pa(i)}), \tag{47}$$

$$\leq \alpha \cdot I(\mathbf{x}_i; \mathbf{x}_{pa(i)}), \tag{48}$$

where the last step is due to $j < i$ and $j \neq pa(i)$ leading to the distance of nodes $d(j, pa(i)) \geq 1$. $\quad\square$

In the proof, we adopt the following two lemmas from Nichani et al. (2024):

**Lemma 5** (Lemma 17 in Nichani et al. (2024)). *Suppose the transition kernel $\pi$ of a Markov chain satisfies $\max_{s \in \mathcal{V}} \pi(s'|s) > 0$ for all $s' \in \mathcal{V}$. Then, for any f-divergence $D_f$ and probability vectors $x, y$, we have:*

$$D_f(\pi \circ x || \pi \circ y) \leq \alpha \cdot D_f(x||y),$$

*where the contraction coefficient $\alpha$ is defined as:*

$$\alpha := \max_{j \neq k} \mathrm{TV}(\pi(\cdot|j), \pi(\cdot|k)) = \frac{1}{2} \max_{j \neq k} \|\pi(\cdot|j) - \pi(\cdot|k)\|_1.$$

**Lemma 6** (Lemma 15 in Nichani et al. (2024)). *Suppose $\min_{s, s' \in \mathcal{V}} \pi(s|s') \geq \frac{\gamma}{|\mathcal{V}|}$. Then we have*

$$\frac{1}{2} \max_{j \neq k} \|\pi(\cdot|j) - \pi(\cdot|k)\|_1 \leq 1 - \gamma.$$

## C.5 PROOF OF LEMMA 4

*Proof.* Note that in the target inequality, the LHS equals $-H(\mathrm{x}_h|\mathrm{x}_{pa(h)})$, while the RHS differs from but can be transformed to $H(\mathrm{x}_h|\mathrm{x}_{h'})$. This result holds for both discrete and continuous x.

For $p(\cdot)$ and $q(\cdot)$ which are two distribution, by KL divergence's non-negativity, we have:

$$\int_s p(s) \log q(s) \le \int_s p(s) \log p(s).$$

Hence we can get:

$$\begin{aligned}
\mathrm{RHS} &= \int_{s,s'} \mathbb{P}(\mathrm{x}_h = s, \mathrm{x}_{h'} = s') \log \mathbb{P}(\mathrm{x}_h = s | \mathrm{x}_{pa(h)} = s') \\
&= \int_{s'} \mathbb{P}(\mathrm{x}_{h'} = s') \int_s \mathbb{P}(\mathrm{x}_h = s | \mathrm{x}_{h'} = s') \log \mathbb{P}(\mathrm{x}_h = s | \mathrm{x}_{pa(h)} = s') \qquad (49) \\
&\le - \int_{s,s'} \mathbb{P}(\mathrm{x}_{h'} = s') H(\mathrm{x}_h | \mathrm{x}_{h'} = s') = -H(\mathrm{x}_h | \mathrm{x}_{h'}),
\end{aligned}$$

where $H(\mathrm{x}_h|\mathrm{x}_{h'}) = H(\mathrm{x}_h) - I(\mathrm{x}_h; \mathrm{x}_{h'})$ is defined in Definition 1. By Lemma 3 and $I(\mathrm{x}_h; \mathrm{x}_{pa(h)}) > 0$, we have:

$$\mathrm{RHS} \le I(\mathrm{x}_h; \mathrm{x}_{h'}) - H(\mathrm{x}_h) < I(\mathrm{x}_h; \mathrm{x}_{pa(h)}) - H(\mathrm{x}_h) = \mathrm{LHS}. \qquad (50)$$

□

## C.6 PROOF OF THEOREM 2

*Proof.* Note that the conclusion can extend to the dynamical system setting.[3] For notational generality, we denote the transition kernel by $p(\cdot \mid \cdot)$ in the following proof; this corresponds to $\pi(\cdot \mid \cdot)$ in discrete Markov chain setting.

Recall that the transformer and BMA have the formula in Eq. (2):

$$\mathcal{A}_{h \to h'}^L = \frac{\exp(\sum_l \log p(\boldsymbol{x}_h^l | \boldsymbol{x}_{h'}^l))}{\sum_{h'' \in [h-1]} \exp(\sum_l \log p(\boldsymbol{x}_h^l | \boldsymbol{x}_{h''}^l))} = \frac{1}{\sum_{h''} \boldsymbol{v}_{h'' \to h'}}, \qquad (51)$$

where we define $\boldsymbol{v}_{h'' \to h'} \triangleq \exp(\sum_l \log p(\boldsymbol{x}_h^l | \boldsymbol{x}_{h''}^l) - \sum_l \log p(\boldsymbol{x}_h^l | \boldsymbol{x}_{h'}^l))$.

By the law of large numbers, we have:

$$\lim_{L \to \infty} \frac{1}{L} \sum_l \log p(\boldsymbol{x}_h^l | \boldsymbol{x}_{h'}^l) = \mathbb{E}[\log p(\boldsymbol{x}_h | \boldsymbol{x}_{h'})] < \mathbb{E}[\log p(\boldsymbol{x}_h | \boldsymbol{x}_{pa(h)})] = \lim_{L \to \infty} \frac{1}{L} \sum_l \log p(\boldsymbol{x}_h^l | \boldsymbol{x}_{pa(h)}^l).$$

Let $\hat{g}_{h,h'} \triangleq \frac{1}{L} \sum_l \log p(\boldsymbol{x}_h^l | \boldsymbol{x}_{h'}^l)$. For all $h'' \ne pa(h)$, we have:

$$\boldsymbol{v}_{h'' \to pa(h)} = \exp\left(\sum_l \log p(\boldsymbol{x}_h^l | \boldsymbol{x}_{h''}^l) - \sum_l \log p(\boldsymbol{x}_h^l | \boldsymbol{x}_{pa(h)}^l)\right) = \exp\left(L(\hat{g}_{h,h''} - \hat{g}_{h,pa(h)})\right) \to 0$$

as $L \to \infty$ and $\lim_{L \to \infty}(\hat{g}_{h,h''} - \hat{g}_{h,pa(h)}) < 0$. Hence, we have $\lim_{L \to \infty} \mathcal{A}_{h \to pa(h)} = 1$. □

## C.7 PROOF OF THEOREM 3

*Proof.* First, the transformer as constructed can be simplified as:

$$\boldsymbol{f}_\theta^{(\mathtt{simp})}(\cdot \mid \mathcal{H}) = \pi^\top \boldsymbol{x}_{1:h-1} \sigma\left(\sum_l \boldsymbol{x}_{1:h-1}^{l\top} \boldsymbol{W} \boldsymbol{x}_h^l\right) \in \mathbb{R}^d, \qquad (52)$$

Considering $\hat{\boldsymbol{p}} = \sum_l \boldsymbol{x}_{1:h-1}^{l\top} \boldsymbol{W} \boldsymbol{x}_h^l = \boldsymbol{0}$ when $\boldsymbol{W} = \boldsymbol{0}$, then $\boldsymbol{p} = \sigma(\hat{\boldsymbol{p}}) = \frac{1}{h-1} \boldsymbol{1}_{h-1}$ and:

$$\boldsymbol{f}_{\theta_0}(\cdot \mid \mathcal{H}) = \pi^\top \bar{\mu}(\boldsymbol{x}_{1:h-1}), \quad \text{where } \bar{\mu}(\boldsymbol{x}_{1:h-1}) = \frac{1}{h-1} \sum_{h' \in [h-1]} \boldsymbol{x}_{h'}. \qquad (53)$$

---

[3]In this case, the strong DPI follows from Lemma 2 by verifying that the equality condition does not hold via direct calculation of covariance among Gaussian variables.

Then based on $\frac{\partial \sigma(\hat{\boldsymbol{p}})}{\partial \hat{\boldsymbol{p}}} = \text{diag}(\boldsymbol{p}) - \boldsymbol{p}\,\boldsymbol{p}^\top$, computing the gradient of $\boldsymbol{W}$ w.r.t loss $\ell$ in Eq. (12) yields:

$$
\begin{aligned}
\frac{\partial \ell(\theta; h, \mathcal{G})}{\partial \hat{\boldsymbol{p}}} &= \mathbb{E}_{\boldsymbol{X}}\big[(\frac{\boldsymbol{x}_h}{\boldsymbol{f}_{\theta_0}(\boldsymbol{x}_h) + \epsilon})^\top \frac{\partial \boldsymbol{f}_{\theta_0}}{\partial \hat{\boldsymbol{p}}}\big] \\
&= \mathbb{E}_{\boldsymbol{X}}\big[(\frac{\boldsymbol{x}_h}{\boldsymbol{f}_{\theta_0}(\boldsymbol{x}_h) + \epsilon})^\top \frac{1}{h-1}(\pi^\top \boldsymbol{x}_{1:h-1} - \pi^\top \bar{\mu}(\boldsymbol{x}_{1:h-1})1_{h-1}^\top)\big] \\
&\stackrel{Eq.\ (53)}{=} \frac{1}{h-1}\mathbb{E}_{\boldsymbol{X}}\big[(\frac{\boldsymbol{x}_h}{\boldsymbol{f}_{\theta_0}(\boldsymbol{x}_h) + \epsilon})^\top (\pi^\top \boldsymbol{x}_{1:h-1} - \boldsymbol{f}_{\theta_0}(\boldsymbol{x}_h)1_{h-1}^\top)\big] \\
&= \frac{1}{h-1}\mathbb{E}_{\boldsymbol{X}}\Big[\big[\frac{\pi(\boldsymbol{x}_h|\boldsymbol{x}_1)}{\boldsymbol{f}_{\theta_0}(\boldsymbol{x}_h) + \epsilon}, \cdots, \frac{\pi(\boldsymbol{x}_h|\boldsymbol{x}_{h-1})}{\boldsymbol{f}_{\theta_0}(\boldsymbol{x}_h) + \epsilon}\big] - 1_{h-1}^\top\Big] \\
&= \frac{1}{h-1}\mathbb{E}_{\boldsymbol{X}}\Big[\big[\frac{\pi(\boldsymbol{x}_h|\boldsymbol{x}_1)}{\mu^\pi(\boldsymbol{x}_h)}, \cdots, \frac{\pi(\boldsymbol{x}_h|\boldsymbol{x}_{h-1})}{\mu^\pi(\boldsymbol{x}_h)}\big] - 1_{h-1}^\top\Big] \in \mathbb{R}^{h-1},
\end{aligned}
$$

where $\epsilon = 0$ and $\boldsymbol{f}_{\theta_0}(s) = \mu^\pi(s)$ for all $s \in \mathcal{V}$ by assumption.

Then let $\hat{g}_{h'}^h$ denote $h'$-th entry in $\frac{\partial \ell(\theta_0; h, \mathcal{G})}{\partial \hat{\boldsymbol{p}}} \in \mathbb{R}^{h-1}$ ($h' \in [h-1]$), we have:

$$
\hat{g}_{h'}^h = \frac{1}{h-1}\mathbb{E}_{\boldsymbol{X}}\big[\frac{\pi(\boldsymbol{x}_h|\boldsymbol{x}_{h'})}{\mu^\pi(\boldsymbol{x}_h)} - 1\big] = \frac{1}{h-1}\Big(\sum_{s,s'} \frac{\pi(s|s')\mathbb{P}(\boldsymbol{x}_h = s,\ \boldsymbol{x}_{h'} = s')}{\mu^\pi(s)} - 1\Big). \tag{54}
$$

By Cauchy-Schwartz Inequality and Data Processing Inequality, we have:

$$
\begin{aligned}
&\mathbb{E}_{\boldsymbol{X}}\big[\frac{\pi(\boldsymbol{x}_h|\boldsymbol{x}_{h'})}{\mu^\pi(\boldsymbol{x}_h)} - 1\big] = \sum_{s,s'} \frac{\pi(s|s')\mathbb{P}(\boldsymbol{x}_h = s,\ \boldsymbol{x}_{h'} = s')}{\mu^\pi(s)} - 1 \\
&\leq \frac{1}{2}\big(I_{\chi^2}(\boldsymbol{x}_h;\boldsymbol{x}_{h'}) + I_{\chi^2}(\boldsymbol{x}_h;\boldsymbol{x}_{pa(h)})\big) \leq I_{\chi^2}(\boldsymbol{x}_h;\boldsymbol{x}_{pa(h)}) = \mathbb{E}_{\boldsymbol{X}}\big[\frac{\pi(\boldsymbol{x}_h|\boldsymbol{x}_{pa(h)})}{\mu^\pi(\boldsymbol{x}_h)} - 1\big].
\end{aligned} \tag{55}
$$

Eq. (55) has shown the desired result $\hat{g}_{h'}^h \leq \hat{g}_{pa(h)}^h$.[4] $\qquad\square$

**Assumption 1** (Assumptions on transition kernel (Nichani et al. (2024), Assumption 1)). *Let $1 - \lambda$ denote the spectral gap of $\pi$. We assume there exists $\gamma > 0$ such that the following hold for $\pi$:*

- *(Transition lower bounded):* $\min_{s,s'} \pi(s' \mid s) > \gamma/|\mathcal{V}|$,

- *(Non-degeneracy of chain):* $\sum_{s \in \mathcal{V}} \|\pi(\cdot \mid s) - \mu^\pi(\cdot)\|_2^2 \geq \gamma^2/|\mathcal{V}|$.

## C.8   PROOF OF PROPOSITION 2

*Proof.* In the dynamical system setting, the transition $P(\cdot|\cdot)$ is given by the pdf of $\mathrm{x}_h|\mathrm{x}_{pa(h)}$:

$$
p(\boldsymbol{x} \mid \boldsymbol{y}) = \frac{1}{(2\pi)^{d/2}(1-\rho^2)^{d/2}} \exp\Big(-\frac{1}{2(1-\rho^2)}\,\big\|\boldsymbol{x} - \rho\boldsymbol{A}^\top\boldsymbol{y}\big\|_2^2\Big),\ \boldsymbol{A} \in \mathcal{O}(\mathbb{R}^d).
$$

Then with $\sigma$ eliminating constant terms in $\log p(\boldsymbol{x}_h|\boldsymbol{x}_{h'})$ with respect to the candidate index $h'$, we get the equivalent form in BMA:

$$
\begin{aligned}
\log p(\boldsymbol{x}_h|\boldsymbol{x}_{h'}) &= \frac{\rho}{1-\rho^2}\,\boldsymbol{x}_h^\top \boldsymbol{A}^\top \boldsymbol{x}_{h'} - \frac{\rho^2}{2(1-\rho^2)}\,\boldsymbol{x}_{h'}^\top \boldsymbol{A}\boldsymbol{A}^\top \boldsymbol{x}_{h'} + \text{const}(h), \\
\bar{\boldsymbol{p}}_{h'}^h &:= \sum_{l=1}^L \big(\frac{\rho}{1-\rho^2}\,\boldsymbol{x}_{h'}^{l\top} \boldsymbol{A}\boldsymbol{x}_h^l - \frac{\rho^2}{2(1-\rho^2)}\,\|\boldsymbol{x}_{h'}^l\|^2\big);\quad \mathbb{P}(pa(h)|\boldsymbol{x}_{1:H}^{1:L}) = \sigma(\bar{\boldsymbol{p}}^h).
\end{aligned} \tag{56}
$$

Eq. (56) gives the BMA logits in the DS setting in a softmax form. We now demonstrate that transformers under the observation restriction Eq. (7) cannot represent BMA in this setting.

---

[4]Let $\hat{\mu}_X := \pi^\top \bar{\mu} = \boldsymbol{f}_{\theta_0}(\boldsymbol{x}_h)$. If we remove the assumption $f_\theta = \pi^\top \bar{\mu} = \mu^\pi$, Lemma 24 in Nichani et al. (2024) shows $\big|\mathbb{E}_{\boldsymbol{X}}\big[\frac{\pi(\boldsymbol{x}_h|\boldsymbol{x}_h')}{\hat{\mu}_X(\boldsymbol{x}_h)+\epsilon} - 1\big] - \mathbb{E}_{\boldsymbol{X}}\big[\frac{\pi(\boldsymbol{x}_h|\boldsymbol{x}_h')}{\mu^\pi(\boldsymbol{x}_h)} - 1\big]\big| \lesssim \frac{1}{\sqrt{T_{\text{eff}}}}$, where $T_{\text{eff}}$ is sequence length $h$ divided by numbers of leaves of tree $\boldsymbol{x}_{1:h}$. Under Assumption 1 and strong data processing inequality in Nichani et al. (2024) (Lemma 5), we can obtain the non-asymptotic result $\hat{g}_{pa(h)}^h - \hat{g}_{h'}^h \geq \frac{1}{h-1}\big(\frac{\gamma^3}{2S} - \frac{2C}{\sqrt{T_{\text{eff}}(\lambda)}}\big)$.

Recall that, under Eq. (7), the transformer logits are:

$$\hat{\boldsymbol{p}}_{\mathtt{tf},h'}^h = \sum_{l=1}^{L} \boldsymbol{x}_{h'}^{l\top} \boldsymbol{W}_{\mathtt{tf}}\, \boldsymbol{x}_h^l,$$

while the BMA logits are:

$$\hat{\boldsymbol{p}}_{\mathtt{BMA},h'}^h = c_1 \sum_{l=1}^{L} \boldsymbol{x}_{h'}^{l\top} \boldsymbol{A} \boldsymbol{x}_h^l + d \sum_{l=1}^{L} \|\boldsymbol{x}_{h'}^l\|^2,$$

where $c_1 = \frac{\rho}{1-\rho^2} \neq 0$, $d = -\frac{\rho^2}{2(1-\rho^2)} \neq 0$. Suppose, for contradiction, that the transformer exactly represents BMA, i.e.,

$$\sigma(\hat{\boldsymbol{p}}_{\mathtt{tf}}^h) = \sigma(\hat{\boldsymbol{p}}_{\mathtt{BMA}}^h) \quad \text{for all DS samples and all } h \in [H].$$

Since softmax is invariant under adding a constant independent of $h'$, this means that for each fixed $h$ there exists a scalar $b = b(h)$ such that:

$$\hat{\boldsymbol{p}}_{\mathtt{tf},h'}^h + b = \hat{\boldsymbol{p}}_{\mathtt{BMA},h'}^h \quad \text{for all } h' \in [h-1]. \tag{*}$$

Using the DS model $\boldsymbol{x}_h^l = \rho \boldsymbol{A}^\top \boldsymbol{x}_{pa(h)}^l + \sqrt{1-\rho^2}\, \boldsymbol{\eta}_h^l$, we expand the logits as:

$$\hat{\boldsymbol{p}}_{\mathtt{tf},h'}^h = \rho \sum_{l=1}^{L} \boldsymbol{x}_{h'}^{l\top} \boldsymbol{W}_{\mathtt{tf}} \boldsymbol{A}^\top \boldsymbol{x}_{pa(h)}^l + \sqrt{1-\rho^2} \sum_{l=1}^{L} \boldsymbol{x}_{h'}^{l\top} \boldsymbol{W}_{\mathtt{tf}} \boldsymbol{\eta}_h^l,$$

$$\hat{\boldsymbol{p}}_{\mathtt{BMA},h'}^h = \left( c_1 \rho \sum_{l=1}^{L} \boldsymbol{x}_{h'}^{l\top} \boldsymbol{A}\boldsymbol{A}^\top \boldsymbol{x}_{pa(h)}^l + d \sum_{l=1}^{L} \|\boldsymbol{x}_{h'}^l\|^2 \right) + c_1 \sqrt{1-\rho^2} \sum_{l=1}^{L} \boldsymbol{x}_{h'}^{l\top} \boldsymbol{A}\boldsymbol{\eta}_h^l.$$

Conditioning on all variables except $\{\boldsymbol{\eta}_h^l\}_{l=1}^L$, both sides of $(*)$ become affine functions of the Gaussian noises $\boldsymbol{\eta}_h^l$. Since the DS distribution has full support and $(*)$ is assumed to hold for all DS samples, the coefficients of the linear terms in $\{\boldsymbol{\eta}_h^l\}$ must match for all realizations. Since $\{\boldsymbol{\eta}_h^l\}_l$ are independently sampled, comparing the coefficients of the linear noise term yields:

$$\sqrt{1-\rho^2}\, \boldsymbol{x}_{h'}^{l\top} (\boldsymbol{W}_{\mathtt{tf}} - c_1 \boldsymbol{A})\boldsymbol{\eta}_h^l = 0, \forall \boldsymbol{\eta}_h^l \in \mathbb{R}^d \Rightarrow \boldsymbol{x}_{h'}^{l\top}(\boldsymbol{W}_{\mathtt{tf}} - c_1 \boldsymbol{A}) = 0.$$

Since in the DS model each $\boldsymbol{x}_{h'}^l$ is non-degenerate with full support, this forces:

$$\boldsymbol{W}_{\mathtt{tf}} = c_1 \boldsymbol{A}.$$

Substituting $\boldsymbol{W}_{\mathtt{tf}} = c_1 \boldsymbol{A}$ back into $(*)$, the representation equation simplifies to:

$$b + c_1 \rho \sum_{l=1}^{L} \boldsymbol{x}_{h'}^{l\top} \boldsymbol{A}\boldsymbol{A}^\top \boldsymbol{x}_{pa(h)}^l = c_1 \rho \sum_{l=1}^{L} \boldsymbol{x}_{h'}^{l\top} \boldsymbol{A}\boldsymbol{A}^\top \boldsymbol{x}_{pa(h)}^l + d \sum_{l=1}^{L} \|\boldsymbol{x}_{h'}^l\|^2.$$

Hence,

$$b = d \sum_{l=1}^{L} \|\boldsymbol{x}_{h'}^l\|^2 \quad \text{for all } h' \in [h-1].$$

However, for a DS sample the quantities $\sum_l \|\boldsymbol{x}_{h'}^l\|^2$ vary across $h'$ and across samples, while $b$ is a constant (depending only on $h$). The only way the above equality can hold for all $h'$ and all DS samples is to have $b = d = 0$, which contradicts the assumption $d \neq 0$ in the BMA logits.

We conclude that no $\boldsymbol{W}_{\mathtt{tf}}$ can make the transformer logits represent the BMA logits for all datasets generated from DS. Therefore, under Eq. (7), transformers cannot represent BMA in the DS setting. $\square$

C.9    PROOF OF CORRECTNESS OF CROSS-EXAMPLE PARENT SELECTION

**Lemma 7.** *Let $\mu^\pi$ denote the stationary distribution. For any valid distribution $\bar{\mu}_t \in \Delta^d$, we have:*

$$\sum_{s,s'} \bar{\mu}_t(s')\mu^\pi(s) \log \pi(s \mid s') \leq \sum_{s,s'} \mu^\pi(s')\pi(s \mid s') \log \pi(s \mid s'), \ \forall \bar{\mu}_t \in \Delta^d.$$

*Proof.* Recall that for the transition kernel $\pi(\cdot \mid \cdot)$ and its stationary distribution $\mu^\pi$, it holds

$$\mu^\pi(s) = \sum_{s' \in \mathcal{V}} \mu^\pi(s')\,\pi(s \mid s') \quad \text{for all } s \in \mathcal{V}.$$

Fix an arbitrary $\bar{\mu}_t \in \Delta^d$, and for brevity write $\mu := \mu^\pi$. We first upper bound the left-hand side. For any $s' \in \mathcal{V}$, consider the KL divergence:

$$\mathrm{KL}\big(\mu \,\|\, \pi(\cdot \mid s')\big) = \sum_{s \in \mathcal{V}} \mu^\pi(s) \log \frac{\mu^\pi(s)}{\pi(s \mid s')} \geq 0.$$

Expanding the inequality $\mathrm{KL}(\mu \,\|\, \pi(\cdot \mid s')) \geq 0$ yields:

$$\sum_{s \in \mathcal{V}} \mu^\pi(s) \log \pi(s \mid s') \leq \sum_{s \in \mathcal{V}} \mu^\pi(s) \log \mu^\pi(s) =: C,$$

where the right-hand side $C$ does not depend on $s'$. Multiplying both sides by $\bar{\mu}_t(s')$ and summing over $s'$, we obtain:

$$\sum_{s,s' \in \mathcal{V}} \bar{\mu}_t(s')\mu^\pi(s) \log \pi(s \mid s') \leq \sum_{s' \in \mathcal{V}} \bar{\mu}_t(s')C = C = \sum_{s \in \mathcal{V}} \mu^\pi(s) \log \mu^\pi(s). \tag{57}$$

This bound holds for any choice of $\bar{\mu}_t \in \Delta^d$.

Next, we lower bound the right-hand side. For each $s' \in \mathcal{V}$, consider the reverse KL divergence:

$$\mathrm{KL}\big(\pi(\cdot \mid s') \,\|\, \mu\big) = \sum_{s \in \mathcal{V}} \pi(s \mid s') \log \frac{\pi(s \mid s')}{\mu^\pi(s)} \geq 0.$$

Hence,

$$\sum_{s \in \mathcal{V}} \pi(s \mid s') \log \pi(s \mid s') \geq \sum_{s \in \mathcal{V}} \pi(s \mid s') \log \mu^\pi(s).$$

Multiplying by $\mu^\pi(s')$ and summing over $s'$ yields:

$$\sum_{s,s' \in \mathcal{V}} \mu^\pi(s')\pi(s \mid s') \log \pi(s \mid s') \geq \sum_{s,s' \in \mathcal{V}} \mu^\pi(s')\pi(s \mid s') \log \mu^\pi(s)$$

$$= \sum_{s \in \mathcal{V}} \Big(\sum_{s' \in \mathcal{V}} \mu^\pi(s')\pi(s \mid s')\Big) \log \mu^\pi(s)$$

$$= \sum_{s \in \mathcal{V}} \mu^\pi(s) \log \mu^\pi(s) = C, \tag{58}$$

where we used $\mu^\pi(s) = \sum_{s'} \mu^\pi(s')\pi(s \mid s')$.

Combining (57) and (58), we conclude that, for any $\bar{\mu}_t \in \Delta^d$,

$$\sum_{s,s' \in \mathcal{V}} \bar{\mu}_t(s')\mu^\pi(s) \log \pi(s \mid s') \leq C \leq \sum_{s,s' \in \mathcal{V}} \mu^\pi(s')\pi(s \mid s') \log \pi(s \mid s').$$

$\square$

### C.10 DEFINITION OF MUTUAL INFORMATION

**Definition 1** (Mutual Information and Conditional Entropy). *Consider* x, y *as two random variables in discrete or continuous space* $\Omega$. *Let* $\mathbb{P}_{x,y}$ *denote the joint distribution, and* $\mathbb{P}_x, \mathbb{P}_y$ *represent the marginal distributions. The mutual information* $I(x; y)$*, entropy* $H(x)$*, and the conditional entropy* $H(x|y)$ *are given by:*

$$I(x; y) = \int_x \int_y \mathbb{P}_{x,y}(x, y) \log \frac{\mathbb{P}_{x,y}(x, y)}{\mathbb{P}_x(x)\mathbb{P}_y(y)}, \quad H(x) = -\int_x \mathbb{P}_x(x) \log \mathbb{P}_x(x),$$

$$H(x|y) = -\int_x \int_y \mathbb{P}_{x,y}(x, y) \log \frac{\mathbb{P}_{x,y}(x, y)}{\mathbb{P}_y(y)} = H(x) - I(x; y), \tag{59}$$

*Further,* $\chi^2$*-mutual information is given by:* $I_{\chi^2}(x; y) := \int_x \int_y \frac{\mathbb{P}_{x,y}(x,y)^2}{\mathbb{P}_x(x)\mathbb{P}_y(y)} - 1.$

## D EXPERIMENT DETAILS

All experiments follow the same training setup unless otherwise specified: sequences are generated from a Markov chain with transition kernel $\pi(\cdot \mid s) \sim \text{Dirichlet}(\alpha \cdot \mathbf{1}_d)$ with $\alpha = 0.1$. We use a batch size of 1024 for training and evaluate on 4096 test samples. Parameters are optimized with Adam (Kingma & Ba, 2015), using a learning rate of 0.05 for discrete Markov chains and 0.001 for dynamical systems. For gradient-based analysis, we adopt SGD with learning rate 1. Fresh data are sampled at each iteration, and all implementations are based on JAX.

## E ADDITIONAL EXPERIMENT RESULTS ON MARKOV CHAIN

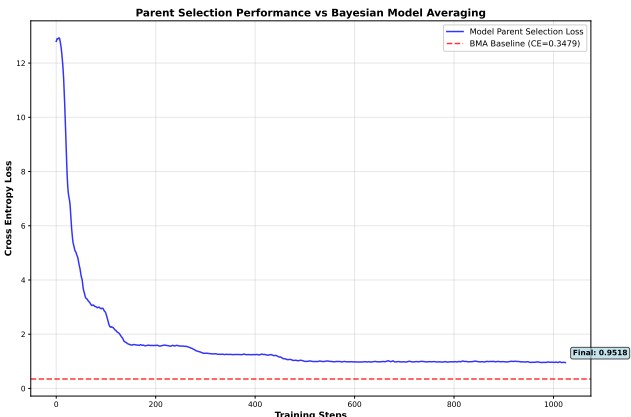

Figure 8: Parent selection $\mathcal{L}_{pa}$ comparison between transformers and BMA during training. The metric is introduced in Eq. (10). The training configuration is the same as in the experiment shown in Fig. 2.

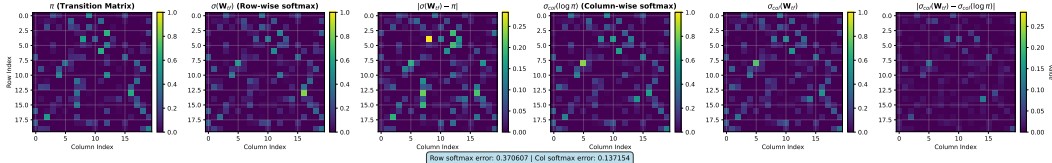

Figure 9: Parameter-level comparison between transformer and BMA ($W = \log \pi$). Trainable $W_{\tt tf}$ and $W_{OV}$. Trained with $d = 20$, $H = 50$, $L = 3$, and 1024 training steps.

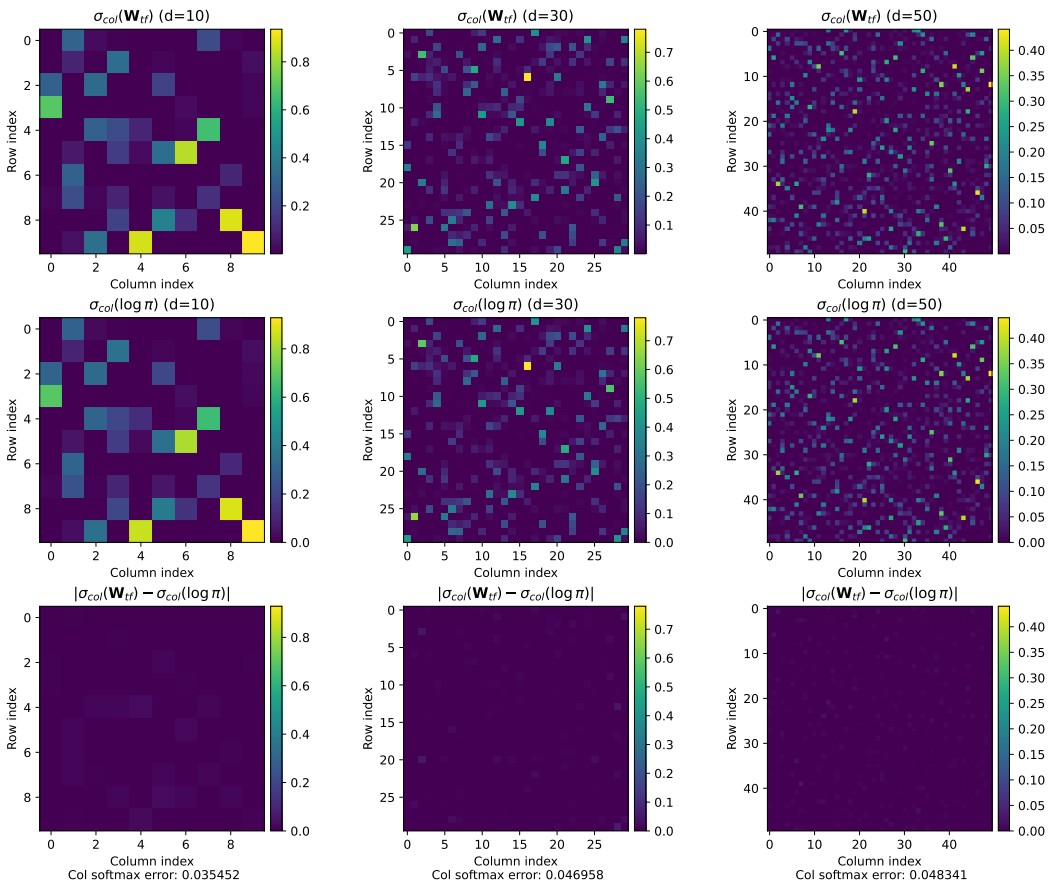

Figure 10: Parameter-level comparison of transformers with different vocabulary sizes where $\boldsymbol{W}_{\mathtt{tf}}$ has size $d \in \{10, 30, 50\}$. We can see $\sigma_{\mathrm{col}}(\boldsymbol{W}_{\mathtt{tf}})$ aligns well with $\sigma_{\mathrm{col}}(\log \pi)$ for all sizes $d$.

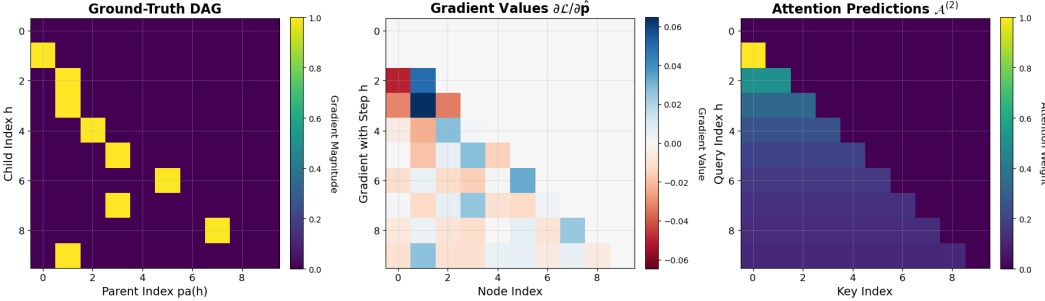

Figure 11: **Gradient Validation of** $\frac{\partial \ell}{\partial \hat{\boldsymbol{p}}}$. From left to right: ground-truth graph $\mathcal{G}$, the gradients of $\frac{\partial \ell}{\partial \hat{\boldsymbol{p}}} \in \mathbb{R}^H$ stacked as row vectors, and attention weights $\mathcal{A}_{h \cdot}^{(2)}$ uniformly distributed since $\mathbf{W} = 0$.

# F    EXPERIMENT RESULTS ON DYNAMICAL SYSTEM

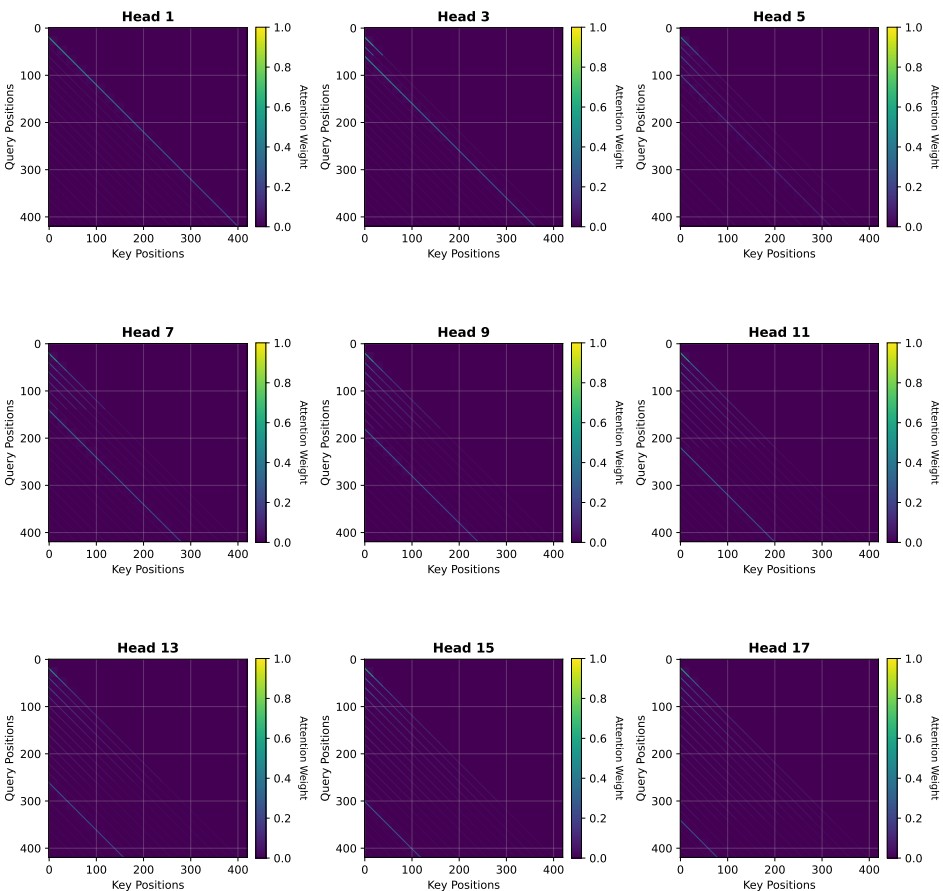

Figure 12: Visualization of 1st-layer attention $\mathcal{A}^{(1)} \in \mathbb{R}^{T \times T}$. For readability, we visualize only nine of the twenty heads, to better highlight the attention patterns on this long sequence of length $400$. The first layer replicates the historical occurrence of the same token. The model was trained with $L = 20$ examples, trajectory length $H = 20$, vocabulary size $d = 10$, 20 heads in the first layer, and $2048$ training steps. The RPE parameters are initialized with a small positive value ($0.5$) along the construction direction, and grow to much larger magnitudes after training.

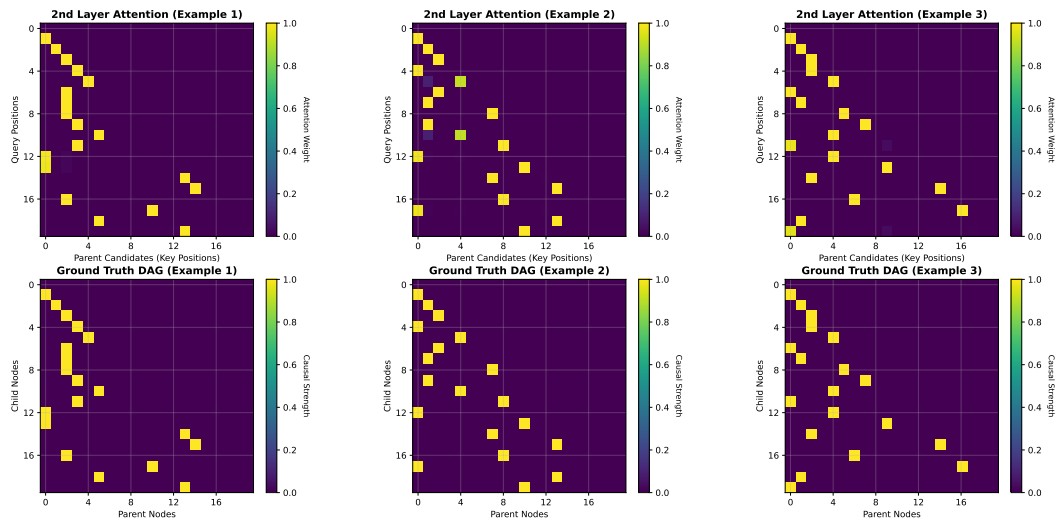

Figure 13: 2nd-layer attention $\mathcal{A}^{(2)} \in \mathbb{R}^{H \times H}$ visualization. The attention patterns match the ground truth causal structure in the dynamical system setting.

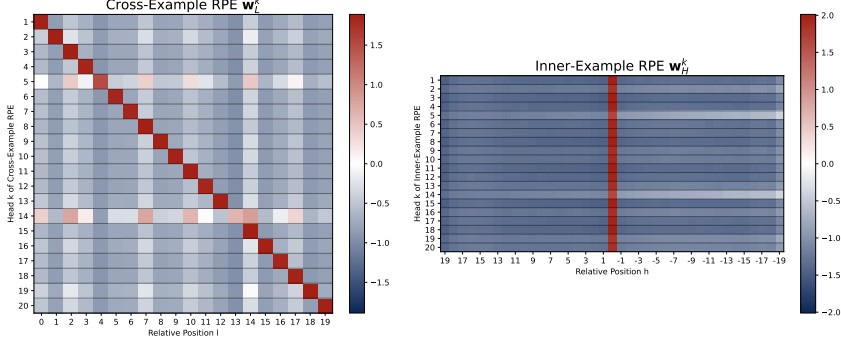

Figure 14: Visualization of the first RPE layer. The parameters are consistent with the construction.

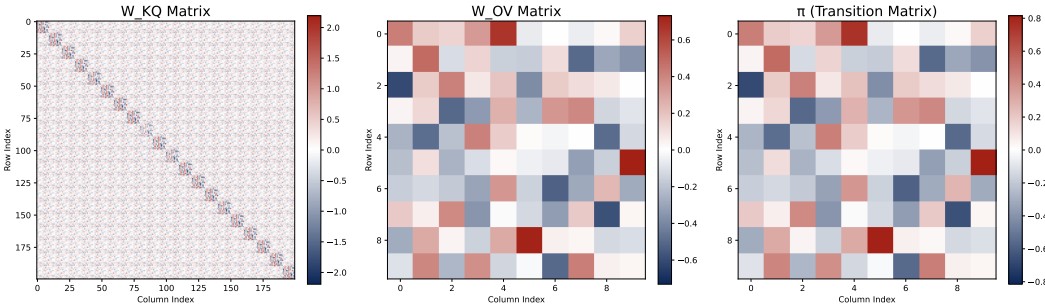

Figure 15: Visualization of the second attention layer. $\boldsymbol{W}_{KQ} \in \mathbb{R}^{dL \times dL}$ shows noticeable non-zero blocks on its diagonal. The occurring block is of size $d \times d$.

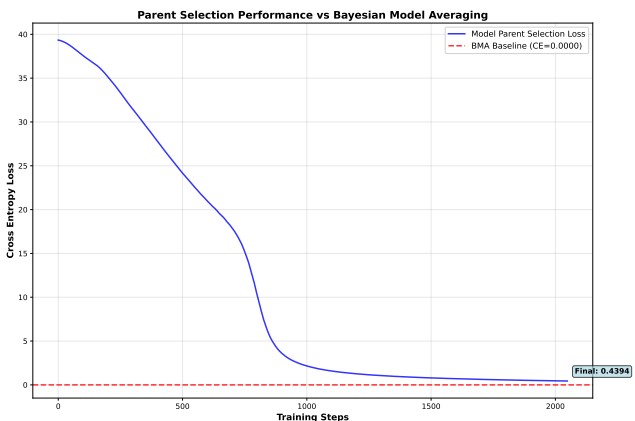

Figure 16: Parent selection loss during training in the dynamical system setting.

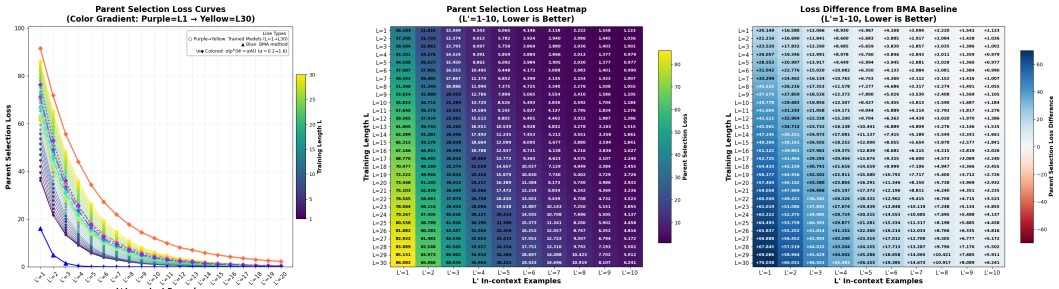

Figure 17: Generalization of parent loss $\{\mathcal{L}_{\mathrm{pa}}^{L'}\}$ for transformers trained with $L \in \{1, \ldots, 30\}$ in the dynamical system setting. Trained with $d = 10$, $H = 15$, and 2048 training steps.

## G  Disentangled Transformer with Absolute Positional Embedding

The disentangled transformer with absolute positional embedding (APE) is formulated as follows:

**Embedding Layer:**[5] $\quad h_t^{(0)} = [E(\boldsymbol{w}_t), \text{Pos}(\boldsymbol{w}_t)] = [\boldsymbol{x}_t, \boldsymbol{e}_t], \; h_t^{(0),q} = [\mathbf{0}, \boldsymbol{e}_t] \qquad \in \mathbb{R}^{d_0},$

**1st Attention (K-head):** $\quad \text{Attn}_t^k(\boldsymbol{H}^{(0)}; \theta) = \sigma\left(h_{1:t-1}^{(0)\top} \boldsymbol{W}_{KQ}^{(1),k} h_t^{(0),q}\right)^\top h_{1:t-1}^{(0)\top} \boldsymbol{W}_{OV}^{(1),k} \qquad \in \mathbb{R}^d,$

**Disentangled Residual:** $\quad h_t^{(1)} = [h_t^{(0)}, \text{Attn}_t^1(\boldsymbol{H}^{(0)}; \theta), \dots, \text{Attn}_t^K(\boldsymbol{H}^{(0)}; \theta)] \qquad \in \mathbb{R}^{d'},$

**2nd Attention (1-head):** $\quad \boldsymbol{f}_{\texttt{tf}}(\cdot \mid \mathcal{H}_t) = \sigma\left(h_{1:t-1}^{(1)\top} \boldsymbol{W}_{KQ}^{(2)} h_t^{(1)}\right)^\top h_{1:t-1}^{(1)\top} \boldsymbol{W}_{OV}^{(2)} \qquad \in \mathbb{R}^d,$

$$\tag{60}$$

First, we can see the model parameter $\boldsymbol{W}_{KQ}^{(1),k} \in \mathbb{R}^{d_0 \times d_0}$ where $d_0 = d + T$ and $T$ is the sequence length. The total number of parameters in the first layer is $O(d^2 + H^2 L^2)$ compared to $O(H + L)$ parameters of the model with RPE in Eq. (5). The redundancy of parameters may lead to difficulties of interpreting the mechanism of transformers. Besides, for the disentangled transformer with APE, the embedding dimension is proportional to the length of input sequence, making it challenging to interpret transformers' mechanism on longer sequence tasks.

As for this transformer, we first provide a theoretical construction which is consistent with our construction for RPE model in Theorem 1. Empirically, we show this transformer can successfully select causal tokens. Besides, we provide results of trainable transformers showing alignments with our construction in attention visualization and parameter verification.

### G.1  Theoretical Construction

In this section, we provide a construction demonstrating how the proposed two-layer architecture possesses the capacity to implement the specific causal selection mechanism derived in our analysis. Let the input embedding dimension be $d_0 = d + T$, where $d$ is the token dimension, $T$ is the sequence length (due to absolute positional embedding) and an input sequence contains $L + 1$ examples of length-$L$ chain $T = H(L + 1)$. Suppose $\mathcal{N}_{L+1}$ denotes the set of nodes from the last example, i.e., we have $\mathcal{N}_{L+1} = \{t \in T \mid \exists h \in [H], t = HL + h\}$.

#### G.1.1  Layer 1: Multi-Head Attention Construction

The first layer consists of $K$ attention heads ($K \le L$). The Query-Key matrix $\boldsymbol{W}_{KQ}^{(1),k}$ attends to specific predecessor tokens based on position. We construct it as a block matrix where the active interaction terms are confined to the positional-embedding subspace:

$$\boldsymbol{W}_{KQ}^{(1),k} = \begin{bmatrix} \mathbf{0}_{d \times d} & \mathbf{0}_{d \times T} \\ \hline \mathbf{0}_{T \times d} & \tilde{\boldsymbol{W}}_{KQ}^{(1),k} \end{bmatrix}, \quad \tilde{\boldsymbol{W}}_{KQ}^{(1),k} = \beta \begin{bmatrix} & \begin{matrix} \mathbf{0}_{H \times H} \\ \vdots \end{matrix} \\ \mathbf{0}_{T \times HL} & \boldsymbol{I}_{H \times H} \\ & \begin{matrix} \vdots \\ \mathbf{0}_{H \times H} \end{matrix} \end{bmatrix} \quad (k\text{-th block active}). \tag{61}$$

From this construction, if $\beta \to \infty$, the attention weight of the first attention layer is given by:

$$\mathcal{A}_{ij}^{(1),k} = \begin{cases} \dfrac{1}{i} \mathbf{1}_{[j < i]}, & \text{if } i \notin \mathcal{N}_{L+1}, \\ \mathbf{1}_{[j = kH + h]}, & \text{if } i \in \mathcal{N}_{L+1},\, i = LH + h. \end{cases} \tag{62}$$

For the value projection, $\boldsymbol{W}_{OV}^{(1),k}$ propagates the semantic content of the attended tokens:

$$\boldsymbol{W}_{OV}^{(1),k} = \begin{bmatrix} \boldsymbol{I}_{d \times d} \\ \hline \mathbf{0}_{T \times d} \end{bmatrix} \in \mathbb{R}^{(d+T) \times d}. \tag{63}$$

---

[5]Following Von Oswald et al. (2023); Zhang et al. (2024), content embeddings $\{E(\boldsymbol{w}_t)\}$ of queries are zeroed to prevent information leakage and self-observation, as they are the targets of prediction. This strict separation of input and target information is consistently applied to the subsequent two Transformer architectures.

And the output of the first attention layer is:

$$\text{Attn}_i^k(\boldsymbol{H}^{(0)}; \theta) = \mathcal{A}_{i\to}^{(1),k} \boldsymbol{h}_{1:T}^{(0)T} \boldsymbol{W}_{OV}^{(1),k} = \begin{cases} \bar{\mu}(\boldsymbol{x}_{1:i-1}), & \text{if } i \notin \mathcal{N}_{L+1}, \\ \boldsymbol{x}_h^k, & \text{if } i \in \mathcal{N}_{L+1}, \, i = LH + h. \end{cases} \tag{64}$$

### G.1.2 DISENTANGLED RESIDUAL STREAM

Unlike standard summation of residuals, the disentangled transformer employs a concatenation strategy. Nichani et al. (2024) proved that this transformer is actually equivalent to a decoder-based attention-only transformer (Theorem 3). The output of the first layer is the concatenation of the original input and the outputs of all $K$ heads:

$$\boldsymbol{h}_t^{(1)} = \left[ \boldsymbol{h}_t^{(0)} \, ; \, \text{Attn}_t^1, \dots, \text{Attn}_t^K \right] \in \mathbb{R}^{d_0 + Kd}. \tag{65}$$

The dimension of the second layer input is $d_1 = d_0 + Kd = d + T + Kd$.

### G.1.3 LAYER 2: SINGLE-HEAD ATTENTION CONSTRUCTION

The second layer employs a single attention head to aggregate the evidence collected by the $K$ heads in the previous layer:

$$\boldsymbol{W}_{KQ}^{(2)} = \begin{bmatrix} 0_{d\times d} & 0_{d\times T} & 0_{d\times Kd} \\ \hline 0_{T\times d} & 0_{T\times T} & 0_{T\times Kd} \\ \hline 0_{Kd\times d} & 0_{Kd\times T} & \tilde{\boldsymbol{W}}_{KQ}^{(2)} \end{bmatrix}, \quad \tilde{\boldsymbol{W}}_{KQ}^{(2)} = \begin{bmatrix} \log\pi & 0_{d\times d} & \cdots & 0_{d\times d} \\ 0_{d\times d} & \log\pi & \cdots & 0_{d\times d} \\ \vdots & \vdots & \ddots & \vdots \\ 0_{d\times d} & 0_{d\times d} & \cdots & \log\pi \end{bmatrix} \tag{66}$$

Finally, the output projection $\boldsymbol{W}_{OV}^{(2)}$ projects the aggregated context back to the semantic space by:

$$\boldsymbol{W}_{OV}^{(2)} = \begin{bmatrix} \log\pi \\ \hline 0_{T\times d} \\ \hline 0_{Kd\times d} \end{bmatrix} \in \mathbb{R}^{d_1 \times d}. \tag{67}$$

From the Eq. (64) and (65), we can see that $\boldsymbol{h}_t^{(1)} = [\boldsymbol{h}_t^{(0)}; \boldsymbol{x}_h^1, \dots, \boldsymbol{x}_h^K]$ if $t \in \mathcal{N}_{L+1}, t = LH + h$, otherwise $\boldsymbol{h}_t^{(1)} = [\boldsymbol{h}_t^{(0)}; \bar{\mu}_t, \dots, \bar{\mu}_t]$. Then, the attention score of the second layer for any $i \in \mathcal{N}_{L+1}$, $i = LH + h$, of our interests, is given by:

$$\tilde{\mathcal{A}}_{ij}^{(2)} = \begin{cases} \sum_{k=1}^K \log\pi(\boldsymbol{x}_h^k | \boldsymbol{x}_{h'}^k), & \text{for } j \in \mathcal{N}_{L+1}, \, j = LH + h', \\ \sum_{k=1}^K \bar{\mu}_j^\top \log\pi \, \boldsymbol{x}_h^k, & \text{for } j \notin \mathcal{N}_{L+1}. \end{cases} \tag{68}$$

So for $i, j \in \mathcal{N}_{L+1}$, we have $\tilde{\mathcal{A}}_{ij}^{(2)} = \sum_{k\in[K]} \log\pi(\boldsymbol{x}_h^k | \boldsymbol{x}_{h'}^k)$ aligned with Theorem 1.

Furthermore, suppose $K = L$, i.e., we use $L$ examples to infer the causal structure, and the Markov chain is stationary $\boldsymbol{x}_h \sim \mu^\pi$. As $L \to \infty$, for any $j \notin \mathcal{N}_{L+1}$, we have $\tilde{\mathcal{A}}_{ij}^{(2)}/L \to \sum_{s,s'} \bar{\mu}_j(s')\mu^\pi(s)\log\pi(s|s') \le \sum_s \mu^\pi(s)\log\mu^\pi(s)$. While for the true parent token $t = HL + pa(h)$, we have $\tilde{\mathcal{A}}_{it}/L \to \sum_{s,s'} \mu^\pi(s')\pi(s|s')\log\pi(s|s')$, which is larger than $\sum_s \mu^\pi(s)\log\mu^\pi(s)$. The above quantity relation is drawn by the non-negativity of KL divergence, whose detailed proof is provided in Appendix C.9. Then, the attention weights of the second layer can select the causal parent token: $pa(h) \in \arg\max_j \lim_{L\to\infty} \mathcal{A}_{ij}$. And $\boldsymbol{W}_{OV}^{(2)}$ predicts the transition.

**Empirical Verification.** We show the parameter visualization of the construction in Fig. 18a and its empirical attention visualization of parent selection in Fig. 18b. In Fig. 19, the constructed model shows precise parent selection accuracy (cross-entropy loss 0.0706), which is very close to the target algorithm BMA's (cross-entropy loss 0.0473).

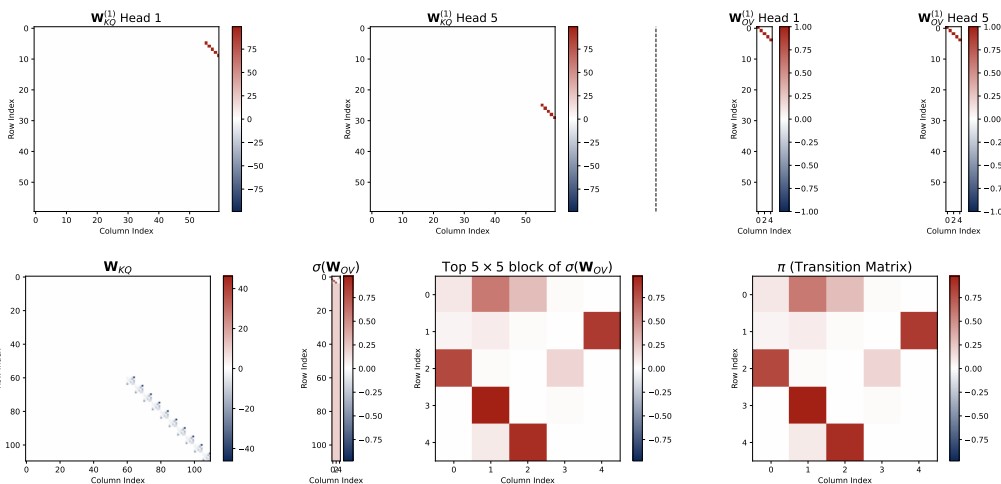

(a) The parameter visualization of theoretical construction.

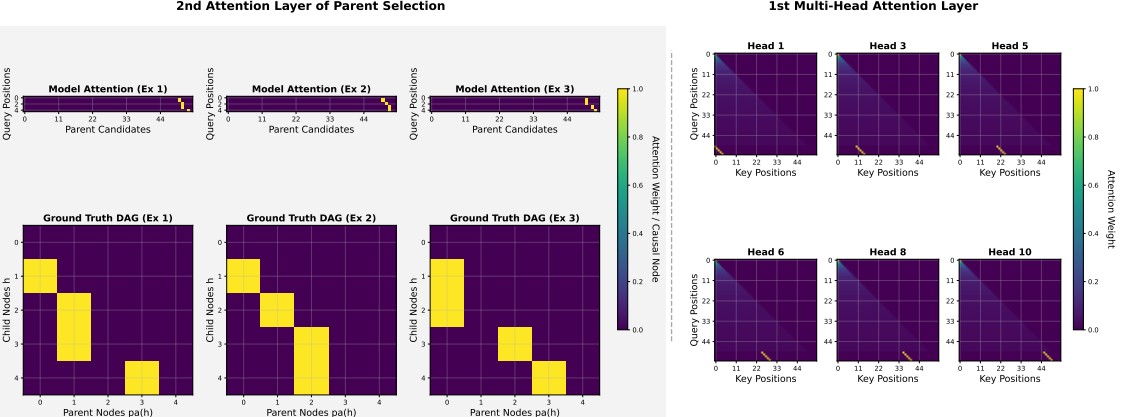

(b) Attention pattern visualization of theoretical construction. In the first layer (right), the previous $L$ examples are copied to the hidden space of the last example, $L + 1$. In the second layer (left), the attention weights attend to the correct causal parents, which are located in the last 5 columns. The queries do not attend to the keys from first $L$ examples empirically.

Figure 18: Parameter visualization and attention pattern visualization of theoretical construction.

## G.2    EXPERIMENTS OF TRAINABLE TRANSFORMERS

In the following, we train the standard disentangled transformer formulated by Eq.(60). To show the alignment with theoretical interpretation, we use three strategies to initialize the network: (a) fully random initialization: all the parameters are initialized randomly with Gaussian distribution; (b) block-amplified random initialization: parameters are initialized randomly (of scale 0.1), while the targeted block of the attention projection matrix is assigned a larger magnitude (of scale 0.5) to introduce an inductive bias; (c) direction-consistent initialization: parameters are initialized such that the dominant blocks point in the analytically derived construction direction, still allowing model learning to refine the magnitudes (initial magnitudes: $0.2\times$ optimal parameters).

We first compare the parent token prediction performance of these models during the training process in Fig. 19. The results show that the 2-layer transformer is fully capable of selecting causal parents in its 2nd-layer attention head.

Then we visualize the attention pattern of the trained model in Fig. 21. For the first attention layer, the figure shows that queries from the last example $L+1$ mostly attend to one example among the $L$ context examples, while some heads demonstrate degeneration with uniform attention to previous tokens. For the second attention layer, the transformers with different initializations all show their noticeable capability of predicting causal parents. Further, we visualize all the parameters of the transformer in Fig. 20. We can see some alignments between the construction in Fig. 18a and the trained parameters. Since the transformer with absolute positional embedding has far more parameters of $(\{\boldsymbol{W}_{KQ}^{(1),k}, \boldsymbol{W}_{OV}^{(1),k}\}_k, \boldsymbol{W}_{KQ}^{(2)}, \boldsymbol{W}_{OV}^{(2)})$ than the one with RPE, the full interpretation of its first layer is difficult. For the second layer, the parameter $\boldsymbol{W}_{KQ}^{(2)}$ also shows the diagonal pattern consistent with construction and $\boldsymbol{W}_{OV}^{(2)}$ shows the $\log \pi$ pattern.

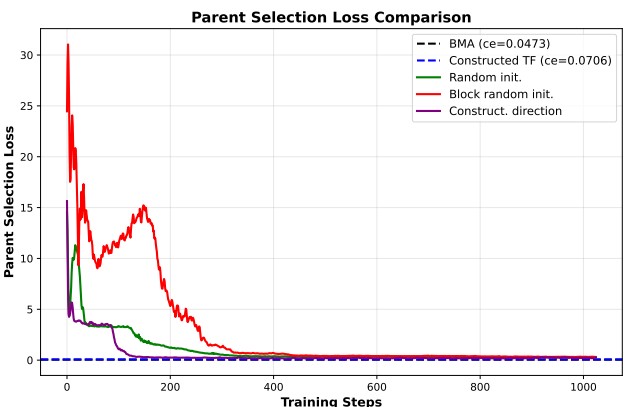

Figure 19: Parent selection loss $\mathcal{L}_{pa}$ of the transformer with absolute positional embedding and different initialization strategies.

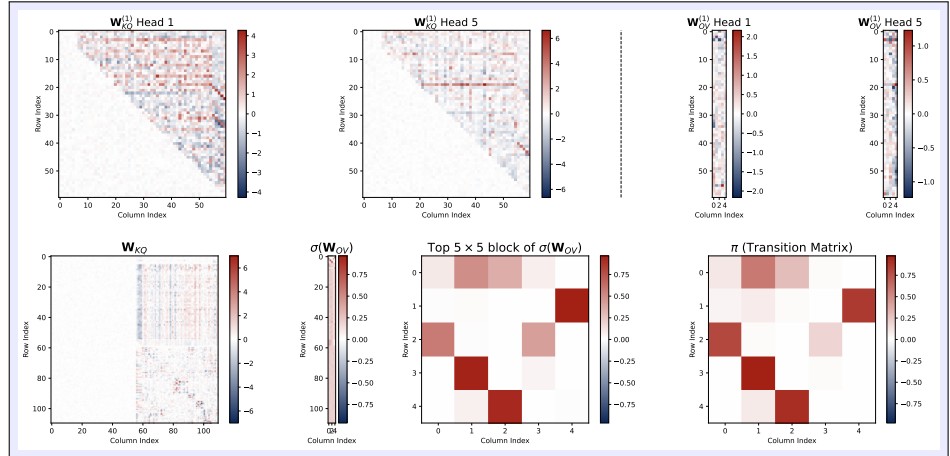

(a) With Fully Random Initialization. Head 1 and 5 of the first layer $\boldsymbol{W}_{KQ}^{(1)}$ exhibit an identity submatrix ($5 \times 5$) at the last column.

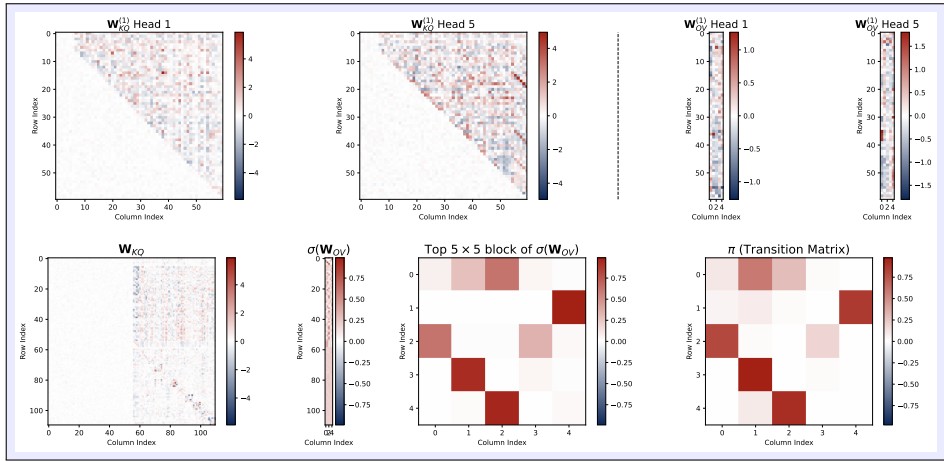

(b) With Block-Amplified Random Initialization. Head 1 of $\boldsymbol{W}_{KQ}^{(1)}$ degenerates which can be verified in attention visualization Fig. 21b (Head 1). Head 5 shows multiple identity submatrices which possibly suggests superposition.

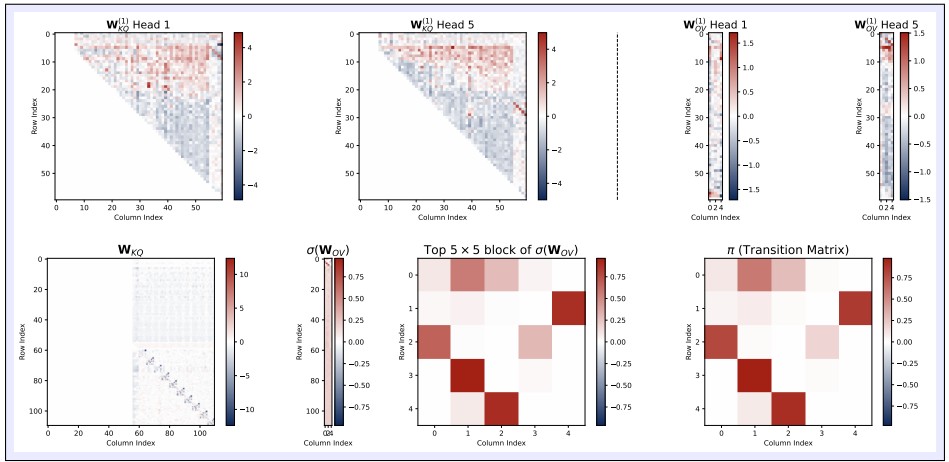

(c) With Direction-Consistent Initialization. Head 1 and 5 of the first layer $\boldsymbol{W}_{KQ}^{(1)}$ exhibit an identity submatrix ($5 \times 5$) at the last column which is aligned with the theoretical construction.

Figure 20: Parameter visualization of trained transformer with absolute positional embedding. The second layer shows strong alignment in diagonal patterns of $\boldsymbol{W}_{KQ}$ and $\log \pi$ pattern of $\boldsymbol{W}_{OV}$.

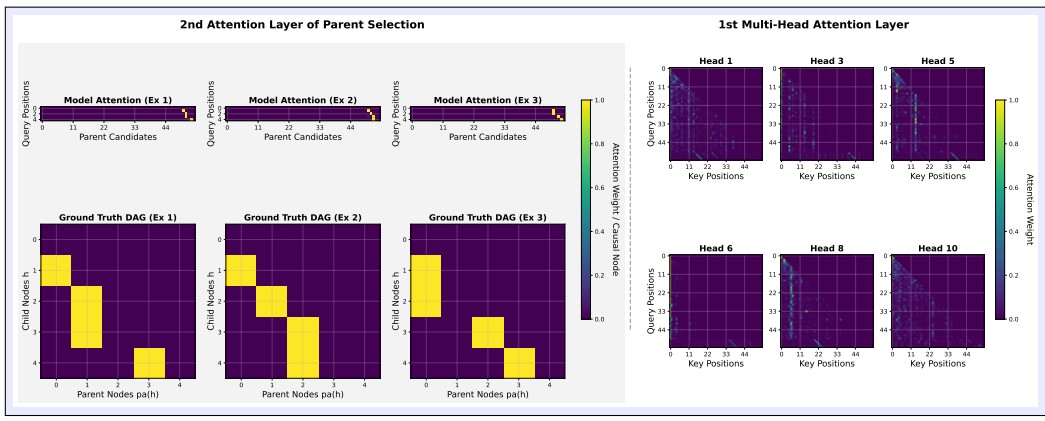

(a) With Fully Random Initialization. In the first layer, Head 1, 3, 5, 6 and 8 of KQ matrices copy tokens from previous examples (to the query token of the last example), while Head 10 degenerates showing uniform attention (uniform features are seen as constants eliminated by 2nd softmax attention layer).

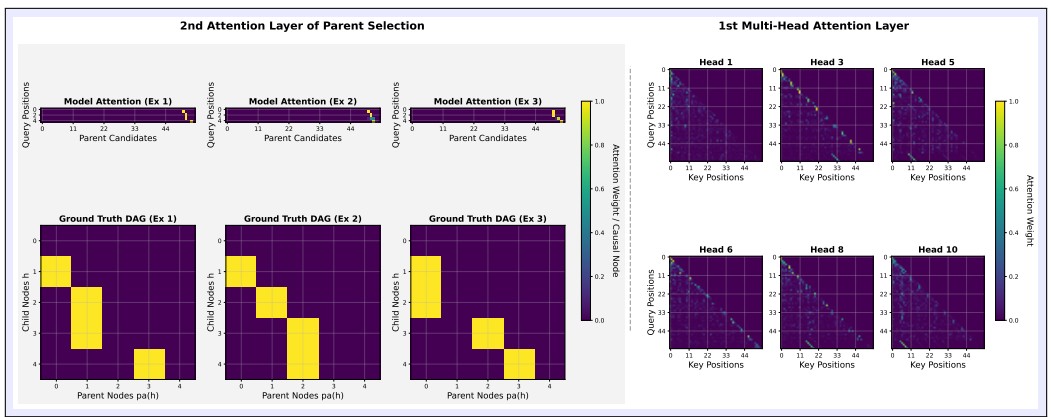

(b) With Block-Amplified Random Initialization. In the first layer, Head 3, 5, 8 and 10 of KQ matrices copy tokens from previous examples, while Head 1 degenerates showing uniform attention.

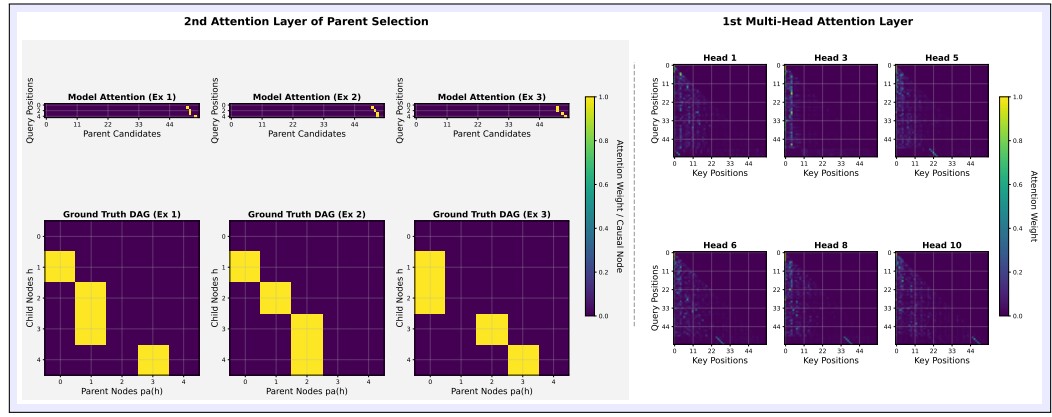

(c) With Direction-Consistent Initialization. In the first layer, Head 1, 5, 6, 8 and 10 of KQ matrices copy tokens from previous examples, while Head 3 degenerates showing uniform attention.

Figure 21: Attention pattern visualization of trained transformer with absolute positional embedding.

# H    DISENTANGLED TRANSFORMER WITH APE VARIANT

## H.1    MODEL ARCHITECTURE

For the variant of two types of APE, we consider the disentangled transformer simplified by eliminating some components added to residual stream. The transformer structure we consider below can be seen as substituing the positional embedding of sturcture Eq. (60) and simplify the model by assuming zero blocks in model weights:

$$
\begin{array}{llll}
\textbf{Embedding Layer:} & \tilde{\boldsymbol{h}}_t^{(0)} = [\mathrm{Pos}_L(\boldsymbol{w}_t), \mathrm{Pos}_H(\boldsymbol{w}_t)] & \in \mathbb{R}^{d_0} \\[2mm]
\textbf{1st Attention (K-head):} & \mathrm{Attn}_t^k = \sigma(\tilde{\boldsymbol{h}}_{1:t-1}^{(0)\top} \boldsymbol{W}_{KQ}^{(1),k} \tilde{\boldsymbol{h}}_t^{(0)})^\top \boldsymbol{x}_{1:t-1}^\top \boldsymbol{W}_{OV}^{(1),k} & \in \mathbb{R}^d, \\[2mm]
\textbf{Disentangled Residual:} & \tilde{\boldsymbol{h}}_t^{(1)} = [\mathrm{Attn}_t^1, \ldots, \mathrm{Attn}_t^K] & \in \mathbb{R}^{Kd}, & (69) \\[2mm]
\textbf{2nd Attention (1-head):} & \boldsymbol{f}_{\mathtt{tf}}(\cdot \mid \mathcal{H}_t) = \sigma\left(\tilde{\boldsymbol{h}}_{1:t-1}^{(1)\top} \boldsymbol{W}_{KQ}^{(2)} \tilde{\boldsymbol{h}}_t^{(1)}\right)^\top \boldsymbol{x}_{1:t-1}^\top \boldsymbol{W}_{OV}^{(2)} & \in \mathbb{R}^d,
\end{array}
$$

where we simply assume $\boldsymbol{W}_{OV}^{(1),k} = I_d$. Since the difference lies in the positional embedding, the construction in Appendix G remains valid which can exhibit capabilities in causal token selection empirically. Besides, we train this transformer under the same Markov chain setup as in the transformer with RPE experiments, obtaining consistent results as shown below.

## H.2    EXPERIMENT RESULTS

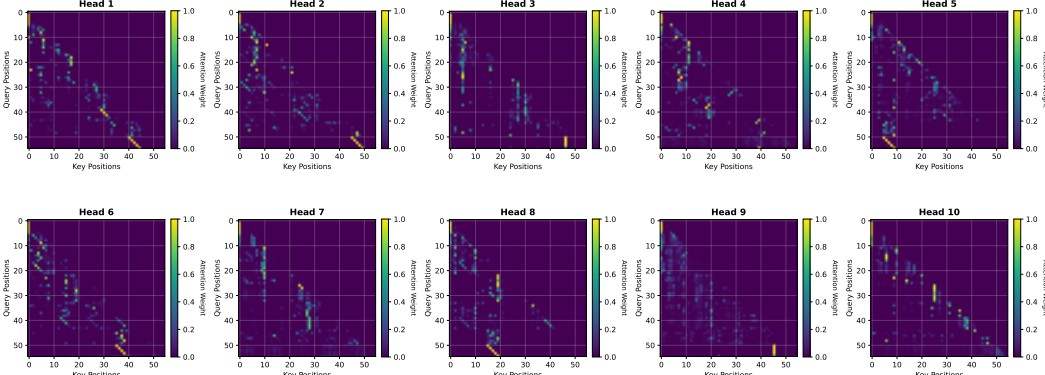

Figure 22: 1st-Layer Attention Visualization of transformers in Eq. (69). Heads 1, 2, 5, 6 and 8 exhibits the diagonal block pattern at the last rows performing the copying mechanism, while Heads 4, 7 and 10 degenerate to uniform attention. Heads 3 and 9 give uniform outputs not influencing the 2nd attention layer (eliminated by softmax attention). Trained with $H = 5, L = 10, d = 5$ and 10000 training steps.

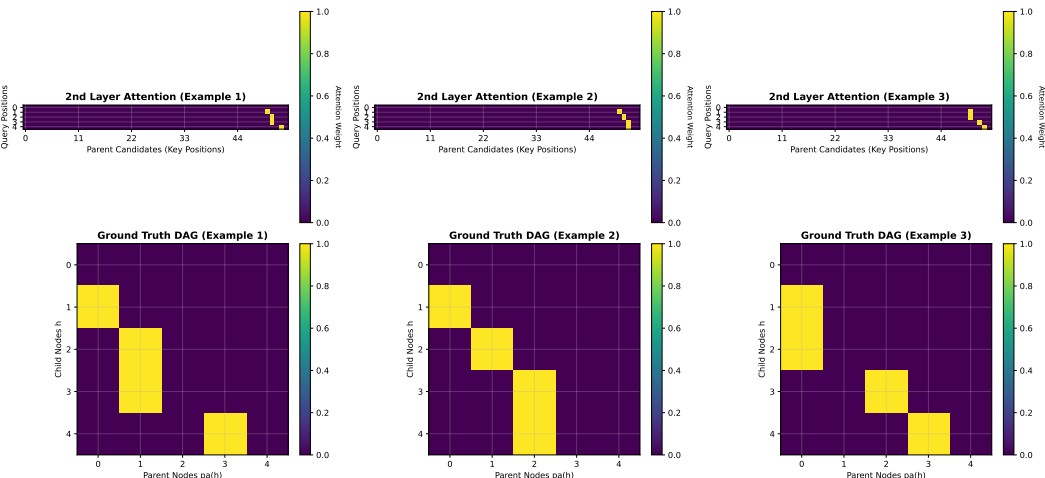

Figure 23: Visualization of 2nd-attention layer. Queries are from the last example $\boldsymbol{x}_{1:H}^{L+1}$. Keys are $\boldsymbol{x}_{1:T} = \boldsymbol{x}_{1:H}^{1:L}$ the whole sequence. Attention layer of disentangled transformer can recognize the causal structure in-context.

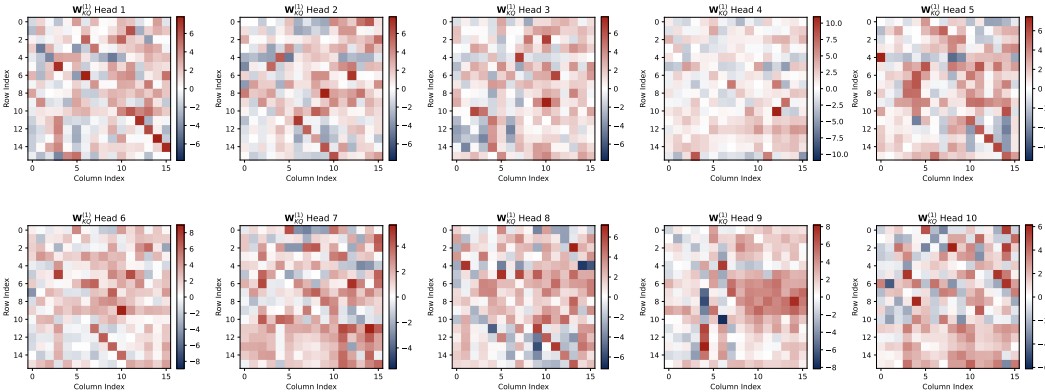

Figure 24: Parameter visualization of the first attention layer $\boldsymbol{W}_{KQ}^{(1),k}$ (10 heads in total). Full interpretation is still challenging for huge parameter space. The attention-level behavior understanding can be referred to Fig. 22.

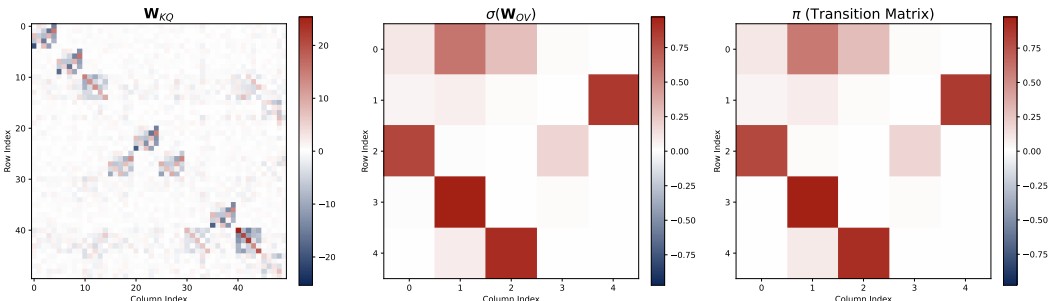

Figure 25: Parameter visualization of the second attention layer $\boldsymbol{W}_{KQ}^{(2)}, \boldsymbol{W}_{OV}^{(2)}$. In the variant of two types of absolute positional embedding, the second layer also shows strong alignment in diagonal patterns of $\boldsymbol{W}_{KQ}$ and $\log \pi$ pattern of $\boldsymbol{W}_{OV}$.

# I  STANDARD TRANSFORMER WITH FEEDFORWARD NEURAL NETWORK

In this section, we consider a standard 2-layer transformer with FFN layers as follows:

$$
\begin{aligned}
\textbf{Learnable Embedding:}^{6} \quad & \boldsymbol{h}_t^{(0)} = \mathbf{Emb_V}(\boldsymbol{w}_t) + \mathbf{Emb_P}(\boldsymbol{w}_t), && \in \mathbb{R}^{d'} \\
\textbf{MHA Layer \& Residual:} \quad & \tilde{\boldsymbol{h}}_t^{(l)} = \boldsymbol{h}_t^{(l)} + \mathrm{MHA}_t(\boldsymbol{H}^{(l)}; \boldsymbol{W}_{KQ}, \boldsymbol{W}_{OV}), && \in \mathbb{R}^{d'}, \\
\textbf{FFN Layer \& Residual:} \quad & \boldsymbol{h}_t^{(l+1)} = \tilde{\boldsymbol{h}}_t^{(l)} + \mathrm{FFN}_t(\tilde{\boldsymbol{H}}^{(l)}; \boldsymbol{W}, \boldsymbol{b}) && \in \mathbb{R}^{d'}, \\
\textbf{Unembedding Layer:} \quad & \boldsymbol{f}_{\mathtt{tf}}(\cdot \mid \mathcal{H}_t) = \boldsymbol{W}_U \boldsymbol{h}_t^{(L)} && \in \mathbb{R}^{d},
\end{aligned}
\tag{70}
$$

where $\boldsymbol{H}^{(l)} = [\boldsymbol{h}_1^{(l)}, \ldots, \boldsymbol{h}_T^{(l)}]$, the multi-head attention (MHA) is formulated by:

$$
\mathrm{MHA}_t(\boldsymbol{H}^{(l)}; \theta) = \sum_k \sigma \left( \boldsymbol{h}_{1:t-1}^{(l)\top} \boldsymbol{W}_{KQ}^{(l),k} \boldsymbol{h}_t^{(l)} \right)^\top \boldsymbol{h}_{1:t-1}^{(l)\top} \boldsymbol{W}_{OV}^{(l),k},
\tag{71}
$$

and the FFN layer is formulated by:

$$
\mathrm{FFN}_t(\tilde{\boldsymbol{H}}^{(l)}; \theta) = \boldsymbol{W}_2 \, \mathrm{ReLU}(\boldsymbol{W}_1 \tilde{\boldsymbol{h}}_t^{(l)} + \boldsymbol{b}_1) + \boldsymbol{b}_2.
\tag{72}
$$

We consider the two-layer transformer $L = 2$ with $K$ heads in the first layer and one head in the second.[7] For the task, the input sequence consists of $M = 10$ in-context examples of Length-$H$ Markov chains with $d = 5$ states and the total length $T = H(M+1)$. We set the hidden dimension as $d' = 128$. For initialization, the parameters $\boldsymbol{W}$ of the transformer are initialized randomly by Gaussian initialization: $\boldsymbol{W}_{ij} \sim \mathcal{N}(0, 1/d_{\boldsymbol{W}})$ where $d_{\boldsymbol{W}}$ is decided by the dimension of $\boldsymbol{W}$. We optimize the model using AdamW with a learning rate of $1 \times 10^{-3}$ and a weight decay of $1 \times 10^{-4}$. Fresh data are sampled at each iteration of training without repetition.

**Experiment Results.** We train two transformers with 5000 steps and $K = 5$ or 10 heads in the first layer. We observe the attention weights of the first layer visualized in Fig. 26a and Fig. 27a implement the copying mechanism where the features of one context example are copied to the position of the last example $M+1$: the heads of the first layer show a diagonal submatrix occurring at the last several rows of example $M+1$. Except for these, the remaining primarily show degenerated attention patterns at the rows of the last example $M+1$. In the visualization of the second layer, we find that the trained standard transformer with MLPs can recognize the causal parents in its attention weights of the second layer. The aligned attention pattern and graph ground truth in Fig. 26b and Fig. 27b support our construction of how transformers can handle in-context causal learning.

**Quantitative Results.** We provide the results regarding how accurately transformers during training can select random parents in their second attention layer in Fig. 28. We use the cross-entropy loss as the evaluation metric for accuracy and compare the trained transformers with BMA. We observe that during the training process, standard transformers gradually acquire the capability of in-context causal learning and approximate the loss of BMA.

---

[6]Similar to the setup in Eq. (60), the query content embeddings are zero-assigned to prevent information leaking.

[7]Our implementation is based on the codebase provided by Nichani et al. (2024).



(a) Visualization of the first multi-head attention layer. Heads 1, 3 and 4 show the diagonal block at the rows of the last example. Information from previous examples is copied to the hidden space of the last example.

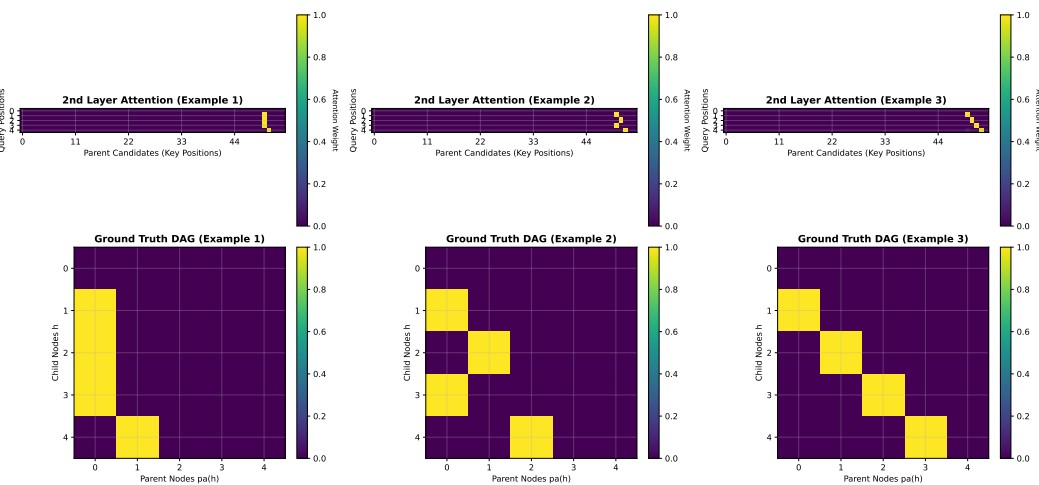

(b) Visualization of the second attention layer. Queries are from the last example $\boldsymbol{x}_{1:H}^{L+1}$. The attention layer of the standard transformer can recognize the causal structure in context.

Figure 26: Attention visualization of the standard transformer with MLPs (5 heads in the first layer).

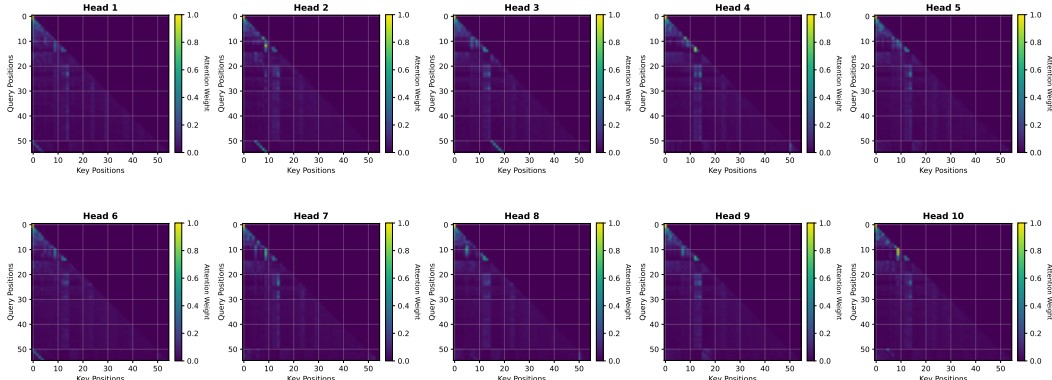

(a) Visualization of the first multi-head attention layer. Heads 1, 2, 3 and 6 show the diagonal block at the rows of the last example. Information from previous examples is copied to the hidden space of the last example.

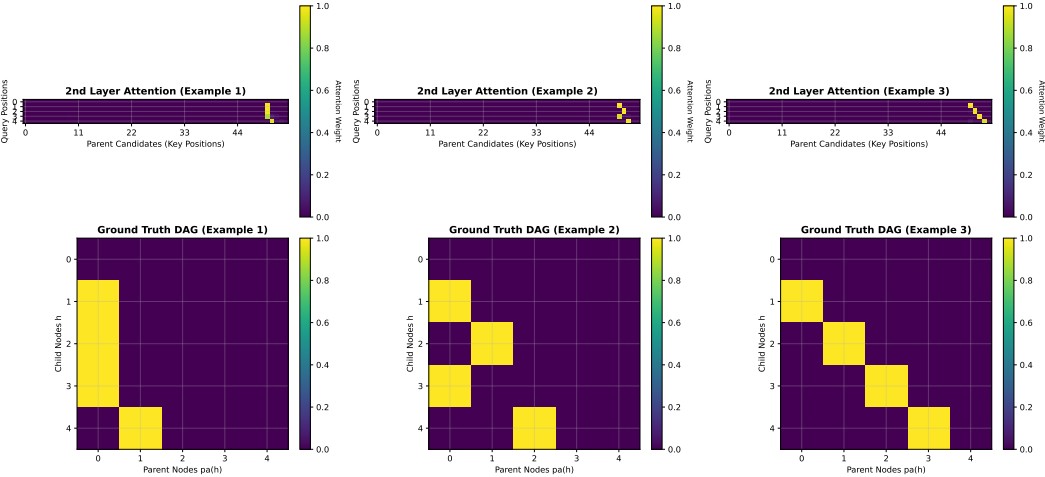

(b) Visualization of the second attention layer. Queries are from the last example $\boldsymbol{x}_{1:H}^{L+1}$. The attention layer of the standard transformer can recognize the causal structure in context.

Figure 27: Attention visualization of the standard transformer with MLPs (10 heads in the first layer).

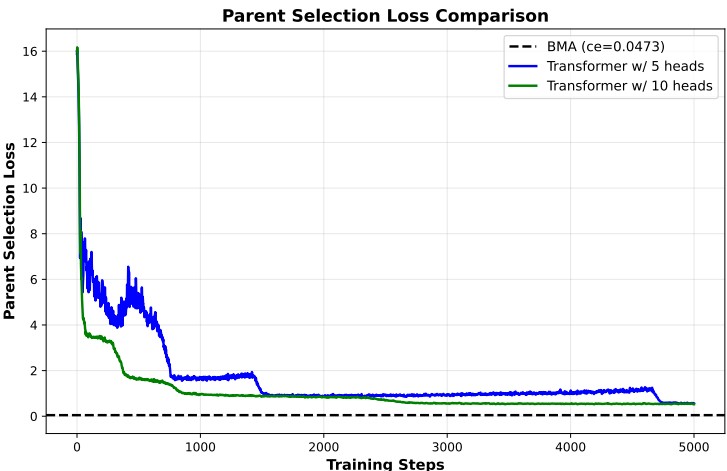

Figure 28: Parent selection loss $\mathcal{L}_{pa}$ of the standard transformer with learnable positional embedding and MLP (5 or 10 heads in the first layer). During training, standard transformers gradually acquire the capability of in-context causal learning and approximate the loss of BMA.

## J PARAMETER-LEVEL VERIFICATION WITHOUT SOFTMAX

In the main experiments, we have compared the discrepancy between $\sigma_{\text{col}}(\boldsymbol{W}_{\text{tf}})$ and $\sigma_{\text{col}}(\log \pi)$. Both of them have scale of $[0, 1]$ after softmax. Here we directly compare $\boldsymbol{W}_{\text{tf}}$ with $\log \pi + \mathbf{1}\boldsymbol{a}^\top$ without softmax. In the following, $(\boldsymbol{W}_{\text{tf}} - \mathbf{1}\boldsymbol{a}^\top)$ is denoted as $\boldsymbol{W}$ for convenience.

To make a reasonable comparison between $\boldsymbol{W} = (\boldsymbol{W}_{\text{tf}} - \mathbf{1}\boldsymbol{a}^\top)$ and $\log \pi$, a reference is needed to assess their difference since $|\boldsymbol{W}_{ij} - \log \pi_{ij}| \in [0, \infty)$. Here, we use $\|\log \pi\|_F$ as the reference and adopt the normalized RMSE (NRMSE):

$$\text{NRMSE}(\boldsymbol{W}, \log \pi) = \frac{\|\boldsymbol{W} - \log \pi\|_1}{\|\log \pi\|_F} = \sum_{ij} \frac{|\boldsymbol{W}_{ij} - \log \pi_{ij}|}{\|\log \pi\|_F}. \tag{73}$$

We trained 3 models with vocabulary dimensions of $\{10, 30, 50\}$. In Fig. 29, we first visualize the element-wise residual $|\boldsymbol{W}_{ij} - \log \pi_{ij}|/\|\log \pi\|_F$ (NRMSE, to preserve scale) and $|\boldsymbol{W}_{ij} - \log \pi_{ij}|^2/\|\log \pi\|_F^2$ (NMSE, to highlight structural deviations) for each model, and compare it with the original $\log \pi$. From the 2nd and 3rd columns of the figure, the residuals are generally within a small region, although some entries have noticeable differences. For these entries, the original $\log \pi$ itself has extremely large values in those place, i.e., $\pi_{ij} \approx 0$ leads to large $\log \pi_{ij}$.

Quantitatively, we characterize the empirical distribution of each element $|\boldsymbol{W}_{ij} - \log \pi_{ij}|/\|\log \pi\|_F$ in the residual matrix. From the last column of Fig. 29, over 90% of the relative error between entries of $\boldsymbol{W}$ and $\log \pi$ lies within the region $(0, 0.01)$.

Combined with heatmap visualization, only at the positions where $\pi_{ij}$ has values close to 0 (e.g., $\pi_{2,6} \sim e^{-40}$, $\log \pi_{2,6} \sim -40$ in Fig. 29, first column), the $\boldsymbol{W}_{\text{tf}} - \mathbf{1}\boldsymbol{a}^\top$ has relatively large approximate error w.r.t. $\log \pi$ (over 0.01 but under 0.13 observed in the last row of Fig. 29). Considering that $\log \pi \in (-\infty, 0]$ has an unbalanced distribution while the trainable $\boldsymbol{W}_{\text{tf}}$ initially has values uniformly around 0, this difficulty in learning small $\pi_{ij}$ may explain the large approximation error.

Further shown in Fig. 30, with continued training (up to $50K$ steps), the trainable $\boldsymbol{W}_{\text{tf}}$ gradually increases its entry magnitudes, moving closer to the boundary values of $\log \pi$. Overall, the results of Fig. 29 and Fig. 30 show that *the trainable transformer parameter $\boldsymbol{W}_{\text{tf}}$ approximates the BMA solution $\log \pi + \mathbf{1}\boldsymbol{a}^\top$ in the parameter space, as stated in Proposition 1.*

Additionally, Fig. 31 shows that most entries of $\sigma_{\text{col}}(\log \pi)$ are close to zero. And after softmax operation eliminates the numerical differences from small values augmented by log (e.g., $e^{-5}$ vs. $e^{-40}$), $\sigma_{\text{col}}(\boldsymbol{W}_{\text{tf}})$ aligns with $\sigma_{\text{col}}(\log \pi)$ across different steps.

**Remark.** The $\boldsymbol{a} \in \mathbb{R}^d$ in the experiments shown in Fig. 29 and 30 is solved by the optimization problem $\arg\min_{\boldsymbol{a}} \|\boldsymbol{W}_{\text{tf}} - \mathbf{1}\boldsymbol{a}^\top - \log \pi\|_F$. Since it has a closed-form solution, the column-wise mean of the residual matrix, we compute it directly.

## K THE USE OF LARGE LANGUAGE MODELS (LLMS)

We acknowledge the use of LLMs like ChatGPT primarily to refine the grammar and improve the presentation of the paper. Besides, they are employed to polish the math proofs where the results are validated by the authors. LLMs also assisted in writing portions of the experimental code, particularly for data visualization, which were reviewed and verified by the authors.

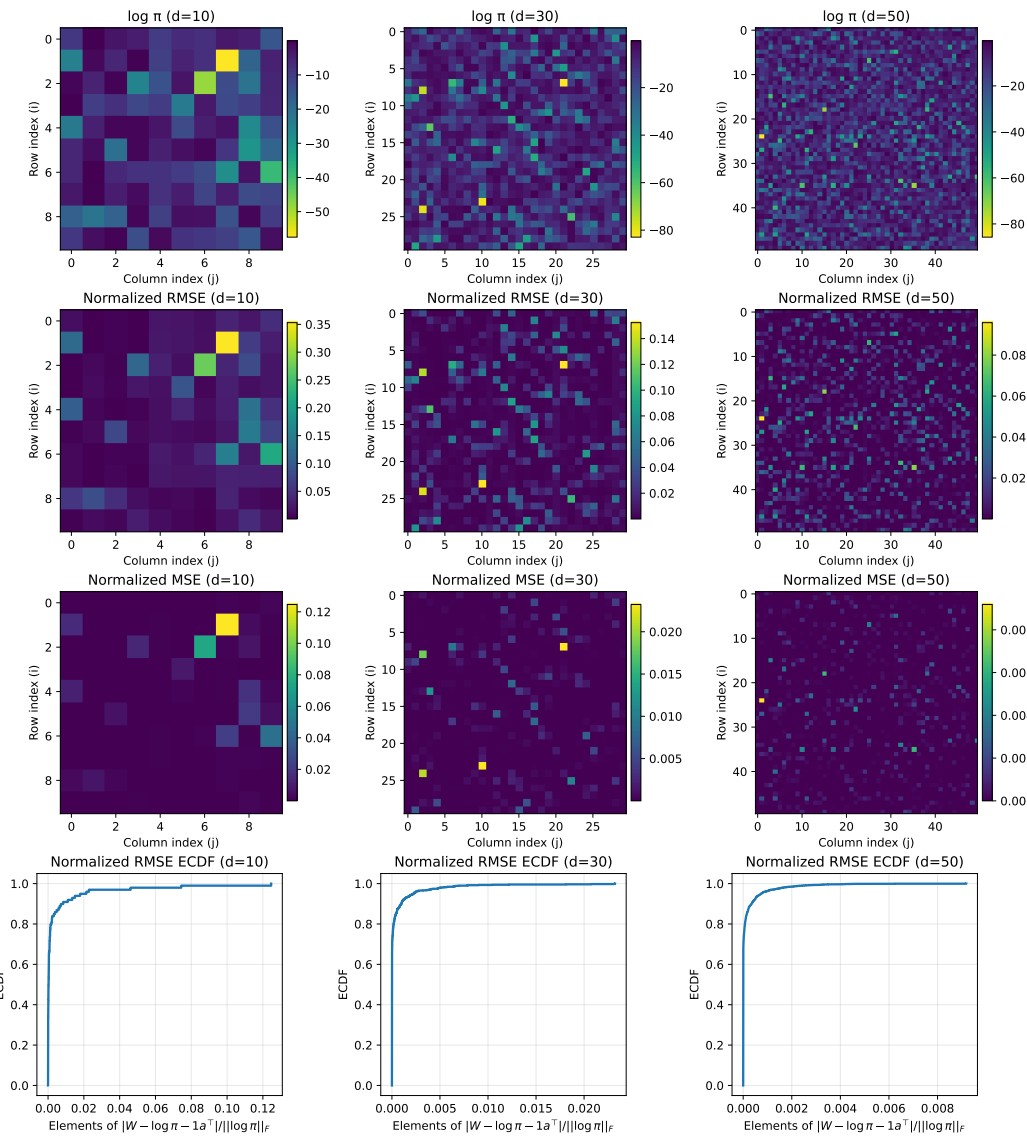

Figure 29: Visualization of $\log \pi$, element-wise NRMSE in Eq. (73), NMSE and the Empirical Cumulative Distribution Function (ECDF) of NRMSE. Larger approximation errors for $\log \pi$ occur at entries where $\pi_{ij}$ is numerically close to zero. The ECDF plots show that a large proportion of the entries approximate $\log \pi$ with a small error, with over 90% falling within a relative error of 0.01. Trained with $H = 50, L = 3, d \in \{10, 30, 50\}$, and 2048 steps.

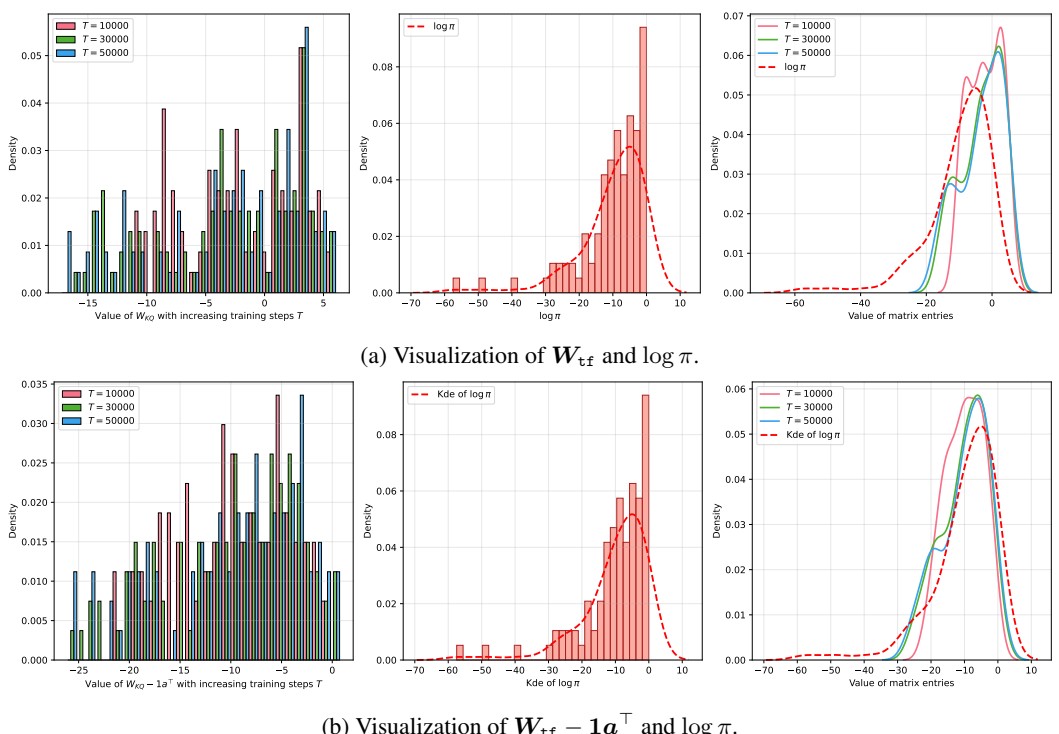

(a) Visualization of $\boldsymbol{W}_{\texttt{tf}}$ and $\log \pi$.

(b) Visualization of $\boldsymbol{W}_{\texttt{tf}} - \boldsymbol{1}\boldsymbol{a}^\top$ and $\log \pi$.

Figure 30: Histogram and density plot of $\boldsymbol{W}_{\texttt{tf}}$ and $\boldsymbol{W}_{\texttt{tf}} - \boldsymbol{1}\boldsymbol{a}^\top$ with increasing training steps $T \in \{10K, 30K, 50K\}$. The entry magnitudes of $\boldsymbol{W}_{\texttt{tf}}$ gradually decrease towards $-\infty$. The density is estimated by kernel density estimation (KDE). Trained with $H = 50, L = 3, d = 10$, and $50K$ steps.

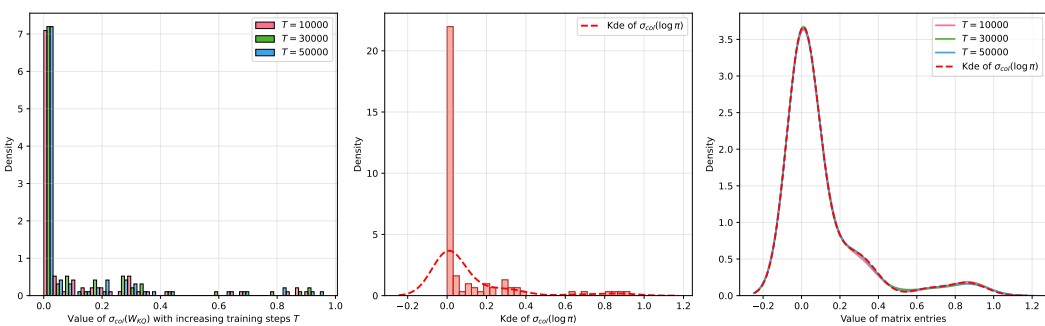

Figure 31: Histogram and density plot of $\sigma_{\text{col}}(\boldsymbol{W}_{\texttt{tf}})$ with increasing training steps $T \in \{10K, 30K, 50K\}$. Density is estimated by kernel density estimate (KDE). $\sigma_{\text{col}}(\boldsymbol{W}_{\texttt{tf}})$ shows an aligned distribution with that of the task parameter $\sigma_{\text{col}}(\log \pi)$. Training setup is the same as above.

