# OpenReview forum: "How Transformers Learn Causal Structures In-Context: Explainable Mechanism Meets Theoretical Guarantee"
_ICLR.cc/2026/Conference — ICLR 2026 Poster_

### Official Review · Reviewer_BaHK · 2025-10-22

**Soundness:** 3
**Presentation:** 3
**Contribution:** 3
**Rating:** 6
**Confidence:** 2

**Summary:**

This paper investigates whether transformers can learn causal structures underlying sequential data. Using a framework based on Markov chains with randomly sampled causal dependencies, it shows that a two-layer transformer with relative positional embeddings (RPE) can exactly implement Bayesian Model Averaging (BMA). It further provides information-theoretic guarantees explaining how transformers recover causal structures.

**Strengths:**

This paper provides a rigorous theoretical construction showing that a two-layer transformer with RPE can provably implement BMA for causal structure inference. The information-theoretic analysis strengthens the understanding by proving that attention weights converge to the true parent structure under increasing in-context examples $L$, using mutual information arguments. So I think this paper make meaningful contribution.

**Weaknesses:**

While this paper is theoretically solid, some assumptions, such as linearity of the dynamics and  specific design of RPE, limit its generality. However, I think that these limitations are acceptable for the scope of this paper. Out of interest, I would like to ask the questions below.

**Questions:**

1. Extension to nonlinear dynamical system:
The current analysis focuses on linear dynamical systems, while transformer is a nonlinear mapping.
Do you expect that the this framework could be extended to the nonlinear dynamical system ?

2. Essential role of RPE: The theoretical construction appears to rely on the use of RPE. Do you think other PE, such as absolute PE or learned PE, could, in principle, reproduce the same behaviors, or is RPE essential for causal structure inference?

---

> ### Author Response · Authors · 2025-11-25
> **Response to Reviewer BaHK**
>
> We thank the reviewer for the positive assessment and for recognizing the rigor of our theoretical construction and the value of our information-theoretic analysis. We appreciate your insightful questions regarding the scope of our framework. We have revised the manuscript to include further discussions and experiments addressing your points.
>
> > **1. Extension to Nonlinear Dynamical Systems**
>
> Thank you for the thoughtful question. We agree that extending the understanding beyond linear dynamical systems is an important next step.
> - The core mechanism we study is that transformers can copy in-context demonstrations through attention and accumulate evidence across demonstrations. In our setting, this leads to an implementation of the BMA estimator, which depends directly on the conditional density $p(x_h|x_{pa(h)})$ of tasks (e.g., Markov chain or dyanmical system). From this conceptual standpoint, the mechanism is not restricted to linear dynamics: if the conditional density of a **nonlinear dynamical system** can be parameterized in a form compatible with the attention architecture, then the transformer can in principle estimate $p(x_h|x_{pa(h)})$ from demonstrations and use it for dependency recovery.
> - However, for the nonlinear case, since the exact BMA score for estimating $p(x_h|x_{pa(h)})$ generally does not admit a bilinear structure as in Markov chain, transformers might need to **approximate** the key BMA term using nonlinear MLP modules rather than implementing BMA in closed form.
>
> We fully acknowledge that this is a potential direction of our future work and discussed it in the Conclusion section **(Appendix C)**.
>
> > **2. Essential Role of RPE (vs. Absolute PE)**
>
> We are glad to report that **RPE is not essential** for the mechanism to emerge. As noted in the **General Response**, we verified the mechanism in the new **Appendix G** on a standard disentangled transformer architecture (equivalent to a decoder-only transformer [1]) with **absolute position embedding**. The results support that our conclusions hold under a more generalized architecture with absolute PE. Besides, we conducted experiments on the standard transformer with MLPs in **Appendix I**, whose **position embedding is fully learnable**. The attention visualization result of its second layer also supports our arguments of the parent selection mechanism. These results show that the mechanism is not tied to RPE, but rather emerges in more general transformer architectures.
>
> ### References:
> [1] Eshaan Nichani, Alex Damian, and Jason D. Lee. How transformers learn causal structure with gradient descent. In Proceedings of the 41st International Conference on Machine Learning, 2024.

---

> > ### Comment · Reviewer_BaHK · 2025-11-26
> >
> > Thank you very much for the clarification. I would like to keep my score.

---

### Official Review · Reviewer_9XhA · 2025-10-28

**Soundness:** 3
**Presentation:** 2
**Contribution:** 2
**Rating:** 4
**Confidence:** 3

**Summary:**

This paper introduces a framework based on Markov Chains to explain how transformers create a causal dependency graph between tokens using ICL. In the Markov chain, each token depends on exactly one prior parent.
1. The paper shows that a disentangled transformer can be trained to do Bayesian Model Averaging.
2. Provides empirical evidence
3. Extends the framework to continuous case (Linear dynamical systems)
4. Provides a theoretical guarantee based on information theory for the transformer causal structure selection.

**Strengths:**

1. The paper makes a good connection between BMA and attention (for the disentangled transformer).
2. Shows there exists a model which, by construction, implements BMA. Also supports the claim empirically.
The core strength of the paper is that it helps in formalising how ICL works through a probabilistic framework.

**Weaknesses:**

1. As mentioned in the strengths, the formalisation of ICL that the paper brings is useful; however, the main weakness is that both the proofs and the experimental work are limited to a special form of the transformer. It is unclear how the results can be applied to the standard architecture.

2. Additionally, the paper claims that the proofs and experiments were conducted on a standard transformer. As far as I understand, they work on the disentangled transformer, which is not the same. Claims should be addressed to reflect the paper.

3. It seems that the paper is missing a conclusion.

**Questions:**

Do the authors have any insight into how their work can be extended to transformers with MLPs and multiple layers?
If not, maybe an experimental section on standard transformers to back the claims empirically?

---

> ### Author Response · Authors · 2025-11-25
> **Response to Reviewer 9XhA**
>
> We thank the reviewer for recognizing the value of our probabilistic framework and the connection between BMA and attention mechanisms. We appreciate your constructive feedback regarding the generality of our architecture and the clarity of our claims. We have significantly revised the manuscript to address these points.
>
> > **1. Applicability to Standard Architectures (APE & Generality)**
>
> We agree that demonstrating generality is crucial. As noted in the **General Response**, we verified the mechanism in the new **Appendix G** on a standard disentangled transformer architecture (equivalent to a decoder-only transformer [1]). The results support that our conclusions hold under a more general architecture. Besides, we also conducted experiments on the **standard transformer with MLPs** in **Appendix I**, whose attention visualization result of its second layer supports our arguments of the parent selection mechanism.
>
> > **2. Clarification on "Standard Transformer" Claims**
>
> We appreciate the reviewer for raising this point. The term **"standard transformer"** was used to refer to disentangled transformer with abosulte position embedding. To reduce ambiguity, we have revised the text to explicitly state where we use the Disentangled architecture rather than standard transformer (**lines 213-214** and **lines 268-269**).
>
> > **3. Missing Conclusion**
>
> We thank the reviewer for the advice. We have added a dedicated **Conclusion** section in **Appendix C** to summarize the key contributions of our work.
>
> > **4. Insights on MLPs and Multiple Layers**
>
> We appreciate the reviewer for raising this question. **Our core insight is providing a mechanistic explanation for how transformers perform in-context causal learning**. Specifically, this is conducted on the minimal architecture sufficient to implement the statistical algorithm (BMA).
>
> * **Theoretical Sufficiency (2-Layer Model):** We prove that a **2-layer** attention-only architecture is theoretically **sufficient** to exactly implement **Bayesian Model Averaging (BMA)** (Theorem 1). We also provide the theoretical guarantee of this causal structure recovering mechanism (Theorem 2). In this work, our goal is to provide a mechanistic understanding of how transformers learn this in-context task, rather than exploring the efficiency of various architectures. Given that the 2-layer model is theoretically sufficient for optimality, extensive experiments on deeper models were not the primary focus.
> * **Experiment with MLPs:** We thank the reviewer for this advice. As discussed above, we conducted experiments on **the standard transformer with MLPs** in **Appendix I**, whose attention visualization result of its second layer supports our arguments of the parent selection mechanism. Theoretically, the existence of MLPs does not prevent the BMA mechanism from emerging: In a standard model with MLPs, ReLU activation layers can approximate identity mappings within the positive quadrant. Since the hidden outputs of our constructed transformer remain non-negative, the MLP layers allow information to pass through without distortion. In the revised manuscript, we have added **Appendix G**, which demonstrates that a constructed disentangled transformer is capable of selecting causal tokens in-context and is verified by empirical results. This structure is equivalent to a **standard decoder-based transformer** proved in [1], thereby validating the existence of the interpreted mechanism within a more generalized model architecture.
>
> ### References:
> [1] Eshaan Nichani, Alex Damian, and Jason D. Lee. How transformers learn causal structure with gradient descent. In Proceedings of the 41st International Conference on Machine Learning, 2024.

---

> > ### Comment · Reviewer_9XhA · 2025-11-28
> >
> > My concerns have been addressed. I will raise my score to 6.

---

### Official Review · Reviewer_h4T5 · 2025-10-31

**Soundness:** 3
**Presentation:** 3
**Contribution:** 3
**Rating:** 6
**Confidence:** 4

**Summary:**

This paper extends prior work on analyzing how transformers learn causal structure by proposing and analyzing a task which requires estimating and performing inference on an unknown causal graph from a number of in-context examples. The main theoretical result is a construction of a two layer disentangled transformer which can compute the posterior distribution over causal graphs (Theorem 1). It then uses experiments on disentangled transformers to support the claim that transformers learn a similar construction.

**Strengths:**

- The toy task is interesting and captures the idea of adapting to an unknown causal structure
- The paper is generally well written
- The authors show consistency of the construction (Theorem 2)

**Weaknesses:**

- Theorems 1 and 3 aren't mathematical theorems due to the use of $\approx$. For example for Theorem 1 I believe the intended theorem is something like there exists a sequence of weights $\theta_\beta$ such that $\lim_{\beta \to \infty} f_{\theta_\beta}(\ldots) = \pi(\ldots)$? Similarly, the mathematical statement of Proposition 2 is also unclear to me.
- As far as I can tell, the paper doesn't actually demonstrate weight-space agreement between their construction and the one learned by gradient descent (only the attention maps are verified). Is the challenge in matching the behaviors of the different heads? It's not clear at all to me from Figure 16 that the construction matches the one in Theorem 1.
- It would be good to run some experiments on a standard transformer with MLPs to check whether you learn a similar construction or get similar performance.

Minor points:
- There is a redundant prove in the abstract: "We **prove** that a two-layer transformer with relative position embeddings can **provably** implement"
- The figures in the appendix are very hard to read. For example, Figure 10 makes it look like every weight is identically 0 since the diagonal entries are barely visible. Perhaps putting the attention weights in log scale or reducing the size would help?
- Typo on line 268: "We set transformers has $K$..."
- I'm not sure what footnote 3 is saying?
- Proposition 2 is cited at Lemma 2 in Appendix B.5.

**Questions:**

- It seems that in addition to reducing the number of parameters, factoring the positional embeddings into L and H also simplifies the construction since it gives you parameter sharing between the in-context examples and the test-example for free. How do the experiments change if you use a more standard architecture?
- Is it possible to interpret the learned attention heads and show that it really does implement the same construction as Theorem 1?

---

> ### Author Response · Authors · 2025-11-25
> **Response to Reviewer h4T5 (Part 1)**
>
> We thank the reviewer for the positive assessment and the constructive feedback regarding mathematical rigor and experimental validation. We are glad you found the task interesting and the paper well-written. We have revised the manuscript to address your concerns.
>
> > **1. Mathematical Rigor (Theorems 1, 3 and Proposition 2)**
>
> We thank the reviewer for raising this point about mathematical rigor. In the revision, we have made these limits and assumptions explicit and stated the results in a fully rigorous form:
>
> * **Theorem 1:** In the initial version, the approximation symbol was used to briefly illustrate that transformers can approximate the formulation of BMA and predict the transition probability. In the revision, we replaced the approximation with limits as $\beta \to \infty$ (RPE inverse temperature) and $L \to \infty$ (number of examples). The revised theorem now states:
>     1.  The causal token selection is consistent with BMA: $\lim_{\beta\to\infty}\mathcal{A}^{(2)}(\mathcal{H};\theta)[h,:]=\lim_{\beta\to\infty}\sigma(p_{\theta}^h)=\sigma(p_{BMA}^h)$.
>     2.  The final prediction converges to the true conditional distribution: $\lim_{\beta, L \to \infty} f_\theta(\cdot|\mathcal{H}) = \pi(\cdot|x_{pa(h)}^{L+1})$ (via Theorem 2).
> * **Theorem 3:** We have refined the statement to explicitly specify the expectations and the stationarity assumption ($x_h \sim \mu^\pi$) required for the gradient dynamics analysis.
> * **Proposition 2:** We have clarified the statement regarding the representational limitation in the dynamical system setting. The proof expands the term $x_h=\frac{1}{c}(A^\top x_{pa(h)}+\epsilon_h)$ and utilizes the independence of the noise $\epsilon_h$. If transformers can represent BMA, it builds an equation which forces $W_{KQ}=\log\pi$ while this weight cannot model the quadratic term in BMA of dynamical system. Detailed description and proof is provided in Proposition 2 and Appendix C.7, respectively.
>
> > **2. Weight-Space Agreement (Construction vs. Learned Weights)**
>
> The reviewer noted that Figure 16 (in the original draft) did not clearly demonstrate the match between learned weights and the construction. We appreciate the reviewer for raising this concern. The construction in Theorem 1 is based on the transformer with relative position embedding (RPE) architecture described by Eq. (5), while Figure 16 in the Appendix (now Fig. 20 in revision version, Appendix H) illustrated the model with absolute position embedding (APE) variants.
>
> We listed the empirical verification conducted in the following and we added them to paper takeaways for better summarizing our work.
>
> * **Attention Visualization:** As mentioned by the reviewer, in Figure 2, we visualize the attention patterns of the trained transformer with RPE. The results show that the first layer facilitates copying context examples to the corresponding query positions. Furthermore, the visualization reveals that the second attention layer possesses the capability to learn causal structures in-context.
>
> * **Parent Selection Evaluation:** In Section 3.3, we conducted experiments to evaluate the parent selection accuracy of the transformer across varying numbers of in-context examples. Figure 5 demonstrates that the transformer’s performance is closely comparable to that of Bayesian Model Averaging (BMA).
>
> * **Parameter-Level Match:** We visualize the parameters of the trained transformer, revealing noticable patterns that align with our theoretical construction. Specifically, we investigate the core weights responsible for parent token selection, $W_{KQ}$. After accounting for equivalence factors, we show that the transformer's learned weight $W_{KQ}$ aligns strongly with the BMA formulation (Figure 6). We further verify that across different training dimensions, the MSE between the theoretical construction $W=\log\pi$ (which implements BMA) and the trained $W_{tf}$ remains within a controllable numerical range (Eq. (11)). These results suggest that the trainable model agrees well with our theoretical construction and interpretation in the weight space.

---

> > ### Author Response · Authors · 2025-11-25
> > **Response to Reviewer h4T5 (Part 2)**
> >
> > > **3. Experiments on Standard Architectures**
> >
> > We agree that results on standard sturctures can make our conclusions more persuasive. As noted in the **General Response**, we added a new **Appendix G** where we constructed a standard disentangled transformer architecture (equivalent to a decoder-only transformer [1]). This transformer is theoretically consistent with the BMA construction in Theorem 1, and empirically, it is verified by experiment results. These findings support that our conclusions hold under a more general architecture.
> >
> > Regarding **MLPs**, thanks for the reviwer's advice. We have conducted the experiment on the **standard transformer with MLP** in **Appendix I**, whose results support our interpretation in attention visualization level. The attention weight of its second layer **can recognize causal parents** from the in-context samples. Theoretically, the existence of MLPs does not prevent the BMA mechanism from emerging: In a standard model with MLPs, ReLU activation layers can approximate identity mappings within the positive quadrant. Since the hidden outputs of our constructed transformer remain non-negative, the MLP layers allow information to pass through without distortion.
> >
> > > **4. Minor Points & Presentation**
> >
> > We have corrected the typos and improved the figures as suggested:
> > * **In Abstract:** We have removed the redundant "provably" phrasing.
> > * **Figure Readability:** We have selected a subset of heads to visualize and enlarged the heatmap figures (Figure 10 in Appendix F) to make the diagonal entries and values legible.
> > * **Typos:** We have fixed the typos pointed out by the reviewer.
> >
> > > **5. Questions: Factoring PE & Interpreting Heads**
> >
> > **Factoring PE:** While factoring RPE into $L$ and $H$ components simplifies the theoretical construction and parameter sharing, it is not strictly necessary. As shown in our new **Appendix G**, the standard architecture with **absolute position embeddings** can implement and learn the same mechanism as the one with RPE introduced in the main text.
> >
> > **Interpreting Heads:** It is indeed possible to interpret the learned heads. As discussed above (in Point 4), we interpret what the trained model implemented via experiment results (Fig. 2, 3, 5 and 6). We have added a summary of key takeaways after each section and refined the figure captions to better illustrate these findings.
> >
> > ### References:
> > [1] Eshaan Nichani, Alex Damian, and Jason D. Lee. How transformers learn causal structure with gradient descent. In Forty-first International Conference on Machine Learning, 2024.

---

### Official Review · Reviewer_M4B8 · 2025-11-01

**Soundness:** 3
**Presentation:** 1
**Contribution:** 3
**Rating:** 4
**Confidence:** 3

**Summary:**

The paper studies how transformers can learn causal structure in context. It designs a task where for each input sequence a causal graph is sampled and used to generate Markov chains (with fixed kernel). Given $L$ examples in context, the model has to learn to predict the continuation of the $L+1$-th sample. The paper shows that there exist a construction of a 2-layer transformer with $L$ attention heads in the first layer which can implement BMA to solve the task, and this is (approximately) learnt by the trained models. Finally, it studies the extension of this task and model to continuous data, i.e. a dynamical system.

**Strengths:**

- The paper studies a novel problem, i.e. how transformers can learn causal structures in-context, which continues an established line of work on understanding the internal functioning of transformers in simplified setups.

- The paper studies the proposed task both theoretically and empirically, showing that transformers implement BMA, and even provides an extension to the continuous case.

**Weaknesses:**

- The paper contains a lot of different parts, which makes it cluttered and none of them is presented too clearly. For examples, the construction of the transformer to implement BMA is not actually given, the figures are too small to be readable, there's no conclusion section. Moreover, the notation is in several places imprecise, e.g. in L292 $x$ is discussed but doesn't appear in the previous equation, in Eq. (6) $k'$ is not defined, etc., and the writing at times unclear. Overall, this makes the paper hard to follow.

- The construction of the transformer which implements BMA needs the number of heads to scale with the length input sequence, which I think it's a limitation and makes the construction even more distant from real-world models.

- While the paper provides several results, it's not clear what the takeaway message is. In fact, the paper shows that transformers can implement BMA to learn a causal structure in context, but this is shown by tailoring the architecture to a specific task (see point above). This construction doesn't seem to reveal some general mechanism applicable to other tasks, as it was for example with induction heads, which may limit the impact of this result on future work.

**Questions:**

See above.

---

> ### Author Response · Authors · 2025-11-25
> **Response to Reviewer M4B8 (Part 1)**
>
> We thank the reviewer for recognizing the novelty of our problem setting and the theoretical significance of connecting Transformers to Bayesian Model Averaging (BMA). We also appreciate the constructive feedback on both the presentation and the generality of our construction. In the revised manuscript, we have substantially improved readability and strengthened the empirical and theoretical evidence to address these concerns.
>
>
> > **1. Presentation: Structure, Notation, and Clarity**
>
> We thank the reviewer for this high-level assessment. We understand that the initial submission contained several presentation issues that made the paper harder to follow. In the revision, we performed a thorough restructuring to improve clarity and readability:
>
> * **Unified Notation:** We have formalized the notations in the **main body** to ensure equations are explicit and unambiguous (e.g., line 299-300, 360 & 367). Additionally, we updated **Appendix A (Notation and Related Work)** to strictly define terms and symbols used.
> * **Smoother Transitions & Takeaways:** To better guide the reader through the theoretical and empirical sections, we have improved the transitions between sections (e.g., line 426-428). We also added explicit **"Takeaway" blocks** throughout the main text (Takeaways 1-5) to summarize key findings at each stage.
> * **Construction of Theorem 1:** We appreciate the reviewer pointing out that the description of the transformer construction was not explicit in the original submission. The revised **Theorem 1** now contains a detailed explanation of the construction that implements $W = \log \pi$ under the restriction in Eq. (7) and followed by a clearer high-level explanation of our construction (line 346-349).
> * **Conclusion Section:** We have added a dedicated **Conclusion** section (and expanded discussion in Appendix C) to clearly summarize the key contributions of our work.
> * **Figures:** We have enlarged key figures (e.g., Figures 2, 3, 5 and 6) and improved the layout of appendix figures to ensure readability.
>
> > **2. Scaling of Attention Heads ($K=L$)**
>
> The reviewer noted that the construction requires the number of heads $K$ to scale with the number of examples $L$, which may be a limitation. We clarify that $K=L$ is a sufficient condition for *exact* theoretical recovery, but not a strict requirement for both theoretical construction and practical performance:
>
> * **Theoretical Construction:** Our construction remains valid for fewer heads with more in-context examples $K < L$. In this case, the first layer copies the (most recent) $K$ examples out of $L$ examples in total, and the second layer performs inference based on this subset with $K$ demonstrations.
> * **Empirical Performance:** Empirically, we can also train transformers with fewer heads $K<L$ for $L$ examples. Since our focus is to illustrate **how transformers can recognize causal structures in-context**, we set $K=L$ in the main experiments for simplicity. Regarding the reviewer’s question, Fig. 5 reports the model’s performance under **varying numbers of demonstrations**. The BMA baseline (implemented by our constructed transformer) shows that the parent-selection loss approaches zero once $L \approx 10$ and plateaus thereafter. This indicates that **a fixed number of heads** (e.g., $K=10$) is sufficient to effectively utilize $K$ demonstrations ($K\geq 10$) for strong in in-context causal structure learning. For sequences with **more than $10$ demonstrations**, one may restrict the number of heads to $K=10$ and use only these $10$ demonstrations to infer the dependency structure.

---

> > ### Author Response · Authors · 2025-11-25
> > **Response to Reviewer M4B8 (Part 2)**
> >
> > > **3.1 Generality of the Model Structure (Tailored vs. Standard Architecture)**
> >
> > Thanks for the comment on the architectural generality concern. As noted in the **General Response**, we added a new **Appendix G** where we constructed a standard disentangled transformer architecture (equivalent to a decoder-only transformer [1]). This transformer is theoretically consistent with the BMA construction in Theorem 1, and empirically, it is verified by experiment results. These findings support that our conclusions hold under a more general architecture.
> >
> > Besides, we also conducted experiments on the **standard transformer with MLPs** in **Appendix I**, whose attention visualization result of its second layer supports our arguments of the parent selection mechanism.
> >
> > > **3.2 Takeaway Message from our work & Broader Implications**
> >
> > We appreciate the reviewer's feedback regarding the clarity of our takeaway and its general applicability. As discussed abbove, new experiments and theoretical construction on disentangled transformers and standard transformers show our interpreted mechanism is general enough on more empirical-aligned architectures. We **acknowledge** that our analysis focuses on specific tasks (Markov chains and linear dynamical systems). However, this choice was deliberate to prioritize **mechanistic interpretability**—enabling us to derive exact proofs and perform parameter-level verification that the model implements Bayesian Model Averaging (BMA). This level of white-box transparency is often intractable in more complex settings but serves as a necessary foundation for understanding the mechanism [1].
> >
> > Despite the specific setting, the **mechanism** we identified may have the following implications for the community of foundation models and LLMs.
> >
> > * **From Copying to Inference:** We identify a higher-order mechanism where transformers perform statistical inference over latent structural uncertainty. This is distinct from "Induction Heads" [1, 2] which typically focus on copying fixed position-dependent patterns. We demonstrate a probabilistic setting where the model must infer a latent dependency structure that varies per example. Our result that trained transformers can implement BMA provides support evidence for a line of work that analyzes transformer-based in-context learning from a statistical inference view [3].
> > * **Mechanistic Grounding:** In a highly structured setting, our work provides a mechanistic understanding of how transformers can adapt to flexible, context-dependent rules rather than relying on the fixed structures [1]. While our construction is intentionally simplified, the mechanism can potentially help explain how transformers can recognize more complex symbolic structures such as context-free grammar [4]. Moreover, empirical ICL observations on natural-language tasks shows that LLMs demonstrate ICL capabilities on seemingly "unstructured" language data [5]. Our results may help understand the pheonomeons due to setting similarity where the transition mappings between words are fixed while the structural positions of word couples vary.
> >
> > While we focus on mechanistic interpretation within a well-formulated in-context causal learning task, extending this work to more complex tasks remains a promising direction for future research. We have explicitly added a discussion of these implications to the Conclusion **(Appendix C)** of our revision.
> >
> > ### References:
> > [1] Eshaan Nichani, Alex Damian, and Jason D. Lee. How transformers learn causal structure with gradient descent. In Proceedings of the 41st International Conference on Machine Learning, 2024.
> >
> > [2] Catherine Olsson, et al. In-context learning and induction heads. arXiv preprint arXiv:2209.11895, 2022.
> >
> > [3] Sang Michael Xie, et al. An explanation of in-context learning as implicit bayesian inference. In International Conference on Learning Representations, 2022.
> >
> > [4] Zeyuan Allen-Zhu and Yuanzhi Li. Physics of Language Models: Part 1. arXiv:2305.13673, 2023.
> >
> > [5] Kevin Christian Wibisono and Yixin Wang. From unstructured data to in-context learning. In The Thirty-eighth Annual Conference on Neural Information Processing Systems, 2024.

---

### Official Review · Reviewer_EPSS · 2025-11-03

**Soundness:** 3
**Presentation:** 2
**Contribution:** 1
**Rating:** 2
**Confidence:** 4

**Summary:**

This paper investigates how transformers can learn and adapt to different causal structures in-context. The authors propose a framework using Markov chains with randomly sampled parent dependencies and prove that a two-layer transformer with some version of Relative Positional Embeddings (RPE) can implement Bayesian Model Averaging (BMA), the optimal algorithm for this inference task. The work provides a detailed theoretical construction, information-theoretic guarantees, and empirical validation showing that trained transformers learn to approximate this mechanism.

The paper's contribution lies in the connection to BMA and its 2-layer RPE construction. However, the manuscript suffers from a significant weakness in the way in which it acknowledges and contextualizes these contributions with respect to relevant prior works. Additionally, the theoretical construction relies on several specific assumptions that tailor the architecture to the specific task.

**Strengths:**

1.  **Principled Connection to Bayesian Inference:** The paper provides a link between the transformer's attention mechanism and Bayesian Model Averaging (BMA).
2.  **2-Layer RPE Construction:** The explicit construction of a 2-layer transformer using a two-axis RPE to implement BMA reinforces the idea that architectural components can be mapped to algorithmic steps.
3.  **Information-Theoretic Guarantees:** The paper provides additional theoretical guarantees for causal structure recovery based on mutual information, strengthening the formal understanding of the learning dynamics.

**Weaknesses:**

1.  **Insufficient Acknowledgment of Prior Work:** The paper lacks a proper discussion of the related work  [1]. It does not properly contextualize its contributions, preventing a clear understanding of the paper's own contribution over prior works.
    - **Framing of Novelty:** The introduction frames the manuscript’s contribution as a departure from prior work limited to "fixed dependency structures" yet [1] seems to have already moved beyond this by introducing the same setup considered here based on in-context causal structure selection in Markov chains. The section "Our Approach", which conceptually outlines a setup closely related to [1] (each token depends exactly on one of the past tokens, with this dependence inferred in-context), omits any mention of this direct precedent work.
    - **Appendix Citation:** The citation in the appendix appears to report incorrect information. It claims that the verification in [1] is "limited to attention visualizations" however, [1] also provides quantitative validation using KL Divergence. It also criticizes [1] for "task-specific constructions," a trait shared by this work (see points below).
    - **Unacknowledged Theoretical Precedent:** the work presents in its Lemma 2, the central statistical argument proving the identifiability of the true causal parent. However, the result seems to closely parallel Lemma 2 in [1].  The present version is slightly stronger (the strict inequality) and presents a different proof, but leads to the same conclusion. This fact should be correctly acknowledged.
    *   **Setups overlap** It would be helpful if the authors could acknowledge the conceptual overlap and similarities with [1]. The key difference seems to me to be that [1] infers a single global structure from within one sequence, while this work infers position-specific structures from across multiple sequences. This makes this work a valuable generalization and a genuine advance in the expressivity of the underlying graphs, but also requires the number of heads as well as the embedding to scale with the length of the examples (H). The two frameworks address the same underlying estimation problem: at a given position, identifying the correct parent accumulating log-likelihoods in a one-parent Markov process,  but differ in how the evidence is presented in the context (within-sequence vs. across-sequences). Acknowledging this would strengthen the paper by highlighting its novelty while giving proper credit.

2. **Dependence of the BMA result on architectural tailoring:** While the paper presents interesting ideas regarding linking in-context causal selection to Bayesian Model Averaging (BMA) in transformers. It remains unclear whether the demonstrated BMA implementation genuinely reflects what standard transformers learn and can implement or rather a consequence of the architectural choices, aligned with the task structure, made in the construction.

3.  **Different attention domains across layers (T×T then H×H)** Layer 1 attends over the full concatenated sequence (shape (T\times T) across examples×positions) while Layer 2 attends only across positions inside an example (shape (H\times H)). This effectively changes the logical index set the model operates on between layers, i.e. the model is built so different layers operate on different axes, tailored to the task they need to solve. It remains unclear whether a standard attention mechanism can discover a similar pattern.

4.  **Two-axis positional encoding (separate lookups for example-index and position-within-example)**
  The construction uses distinct positional biases for the example axis and the within-example position axis. This is a RPE that is tailored to the task and explicitly factors the task axes; it is not the standard single-axis absolute or relative positional scheme used in many transformer models. It remains unclear if replacing the two-axis RPE with a single standard positional encoding (absolute or standard RPE) the same BMA mechanism emerges or if it can even be represented.

5.  **First layer omits W_QK and W_OV.** The first attention layer is constructed to behave as a direct copier (fetching particular past tokens), which is commonly employed in the literature. However, the matrices acting over the semantics W_QK and W_OV seem to be omitted in the construction. It is acceptable for the purpose of the theoretical construction to fix these components (W_{QK} and W_{OV}​) to zero, but this design choice should be stated explicitly. Moreover, the paper should provide empirical evidence showing that allowing these parameters to be trainable does not substantially alter the main BMA mechanism or invalidate the proposed construction.

6.  **No positional encoding  for the second attention layer**. Similarly, the construction does not handle positional information consistently across layers: the second attention layer either omits or fixes to zero positional encodings compared to the first. This asymmetry effectively hardcodes aspects of the intended computation and departs from the symmetry typical of standard multi-layer transformers. As in the previous point, it would be important to assess whether including full positional biases in the second layer would materially change the mechanisms the model implements.


7. **Theorem 3** Theorem 3 leverages the same underlying statistical mechanism used in [2] (as acknowledged by the authors): it expresses the gradient signal in terms of a χ²–mutual information measure between each candidate parent and the child token, and then applies the Data Processing Inequality to show that this measure is maximized for the true parent. The contribution of Theorem 3 is, in my opinion, incremental: it adapts the same statistical idea to the specific transformer construction that exactly implements Bayesian Model Averaging. Moreover, the paper states that its proof “eliminates the stationary assumption of the data distribution and doesn’t require the Markov chain to be mixed,” but  Theorem 3 still relies on expectations such as $(\mathbb{E}[\pi(x_h \mid x_{h'}) / \mu(x_h)])$, which implicitly assume access to a stationary marginal distribution or i.i.d. sampling across examples. Without some ergodicity or mixing condition, it is unclear to me how these expectations are defined or estimated.

[1] D'Angelo Francesco, Francesco Croce, and Nicolas Flammarion. "Selective induction heads: How transformers select causal structures in context." In The Thirteenth International Conference on Learning Representations. 2025.

[2] Nichani Eshaan, Alex Damian, and Jason D. Lee. "How Transformers Learn Causal Structure with Gradient Descent." In Forty-first International Conference on Machine Learning. 2024.

**Questions:**

1.  **Unused Heads (Lines 294–295):** The authors state that "some heads... didn’t learn meaningful features." According to the construction, each head in the first layer should learn a specific copying mechanism. Does this observation imply the model was trained with more heads than theoretically necessary, or that some heads failed to specialize as expected?
2.  **Attention Notation (Lines 246–252):** The equations appear to use both a generic activation `σ(⋅)` and `softmax(⋅)` in the context of attention. Could the authors clarify the distinction or correct the notation if they are intended to be the same?
3.  **BMA Formulation (Lines 138–140):** For clarity, it would be helpful to explicitly write out the analytical expression for the likelihood `P(x | pa(h))` and the resulting posterior-predictive (BMA) distribution. This would make the target algorithm that the transformer is shown to implement immediately clear to the reader.
4.  **Theorem 3** Can the authors clarify under what distribution the expectations in Theorem 3 are taken, and how the result holds when the underlying Markov chain is non-stationary or non-mixing?
5. Could the authors clarify whether a transformer with conventional architecture and positional encodings (without separating L position and H position) would still approximate BMA under the same task formulation, or if these design constraints are essential for the claimed behavior?

---

> ### Author Response · Authors · 2025-11-25
> **Response to Reviewer EPSS (Part 1)**
>
> We thank Reviewer EPSS for the detailed and insightful evaluation. We deeply appreciate the reviewer’s careful engagement with the paper’s theoretical construction, architectural generality, and connections to prior work. We have substantially revised the manuscript, particularly the **Related Work (Appendix A)** and **Experiments (Appendix G, H & I)**, to address the reviewer's concerns.
>
> > **1.1 & 1.4 Insufficient Acknowledgment of Prior Work (D'Angelo et al. [1]), Framing of Novelty & Setup Overlaps.**
>
> we appreciate the reviewer for pointing out the lack of proper discussion about [1] in our initial submission. We acknowledge that [1] is a seminal contribution to in-context causal learning. We have carefully revised the Related Work section and added a dedicated discussion in **Appendix A** to reflect the connection more clearly.
> While both works address in-context causal learning, we respectfully emphasize that the two contributions are complementary rather than overlapping. For clarity, we summarize the key differences as follows:
>
> * **Local vs. Global Structure:** As noted in our revised Related Work, [1] focuses on inferring a single **global** structural parameter (the lag $k$) shared across the sequence. In contrast, our work generalizes this to **local** structure inference, where dependencies $pa(h)$ are sampled uniformly and vary for each position. This requires the model to infer a latent tree structure rather than a fixed-order chain.
> * **Algorithmic Construction:** [1] constructs a 3-layer transformer to implement Maximum Likelihood estimation. We derive a **2-layer** architecture that explicitly implements **Bayesian Model Averaging (BMA)**.
> * **Additional Contributions:** Furthermore, we provide parameter-level verification, the information-theoretic proof to understand in-context causal token selection, and results on a dynamical system extension.
>
> > **1.2 Appendix Citation (Report Incorrect Information of [1])**
>
> We thank the reviewer for pointing this out. We have revised **Appendix A (Related Work)** to include the *quantitative validation using KL divergence* reported in [1], which was previously omitted in our summary. The revised text now (i) accurately summarizes the contributions of [1], including its empirical validation, and (ii) highlights the complementary relationship between the two works.
>
> > **1.3 Unacknowledged Theoretical Precedent (Lemma 2)**
>
> The reviewer noted that our Lemma 2 (now **Lemma 3** in the revised manuscript) "closely parallels Lemma 2 in [1]". While the conclusion (identifying the parent) is directionally similar, the mathematical quantities and proof techniques are fundamentally different, as discussed below:
>
> * **Our Proof (Information-Theoretic):** We prove the inequality based on **Cross Entropy (Log-Likelihood)**:
>     $\mathbb{E} [x_{pa(h)}^\top \log\pi x_h] \geq \mathbb{E} [x_{h'}^\top \log\pi x_h]$
>     Our proof uses the **Data Processing Inequality (DPI)** on Mutual Information ($I(x_h; x_{pa(h)}) \geq I(x_h; x_{h'})$). **We do not assume the Markov chain is stationary** for this inference bound.
> * **Precedent [1] Proof:** In contrast, Lemma 2 in [1] bounds the **Linear Probability**: $\mathbb{E} [x_{pa(h)}^\top \pi x_h]\geq\mathbb{E}[x_{h'}^\top\pi x_h]$. Its proof explicitly relies on the existence of a **stationary distribution** and their analysis cannot be directly applied to the $\log\pi$ case.
>
> > **2. Dependence on Architectural Tailoring (Standard vs. Disentangled)**
>
> We appreciate the concern about architectural dependence.  As noted in the **General Response**, we added a new **Appendix G** where we constructed a standard disentangled transformer architecture (equivalent to a decoder-only transformer [1]). This transformer is theoretically consistent with the BMA construction in Theorem 1, and empirically, it is verified by experiment results. These findings support that our conclusions hold under a more general architecture.
>
> Besides, we also conducted experiments on the **standard transformer with MLPs** in **Appendix I**, whose attention visualization result of its second layer supports our arguments of the parent selection mechanism.

---

> > ### Author Response · Authors · 2025-11-25
> > **Response to Reviewer EPSS (Part 2)**
> >
> > > **3-6. Architecture Concerns: Different attention domains, Two-axis PE, First layer omits $W_{QK}$/$W_{OV}$, No PE for 2nd layer**
> >
> > The reviewer raised concerns regarding the generality of our simplified model. To address this, we emphasize that in the **revision version**, the **standard disentangled transformer** (formulation given by Eq. (34) in the revision, Appendix G) and **standard transformer with MLP** (Appendix I) **incorporates all the components** mentioned in points 3-6 as discussed by the reviewer. Take the disentangled transformer in Appendix G as the example: it uses the full attention which can attend to each position of the input sequence **(for Point 3)**; it adopts the standard absolute positional embedding **(for Point 4)**; in the first layer, the model adopts the $W_{QK}$/$W_{OV}$ matrices for attention mechanism **(for point 5)**; since in the residual stream, it incorporate the information from the embeded feature, it considers the information from absolute position embedding **(for Point 6)**. From the concern of generality, this structure is proved to be mathematically equivalent to the **standard transformer** ([2], Theorem 3). And we theoretically constructed a disentangled transformer whose parent token selection is consistent with our Theorem 1 for the simplified model.
> >
> > As discussed above, the new **experiments** (Appendix G & I) demonstrate that this standard architecture learns the same mechanism as our simplified model. These consistent results fully **support** the validity of our simplification, where the simplification will allow for a more comprehensive interpretation of the mechanism and ease the training process. We have added a discussion on this in the **Architecture section** (lines 213-215) and **Appendix G** (lines 1474-1477).
> >
> > > **7. Theorem 3 and Stationarity Assumptions**
> >
> > We thank the reviewer for raising this point. Theorem 3 relies on the stationarity assumption while Theorem 2 of Section 3.4 does not.
> > * **Clarification in the text.:** We have refined the statement of our proof assumption clearly showing it is for **Theorem 2** and clarified in **Theorem 3** that our *gradient dynamics* analysis (Thm. 3) indeed assumes stationarity ($x_h \sim \mu^\pi$).
> > * **Contribution Compared to [2]:** While [2] shows how model weights learn a fixed causal structure, our gradient-based analysis focuses on showing that the graph structures *from in-context examples* can be recovered in the gradients w.r.t. intermediate features *not w.r.t. parameters*, as these structures are not encoded in static parameters. This theorem intends to illustrate how the optimization process can utilize the groundtruth graph information for in-context causal learning.
> >
> > > **8. Specific Questions**
> >
> > * **Unused Heads:**
> >     - In experiments, we train transformers for $L$ in-context examples with $K$ heads where we set $K=L$. The degeneracy of some heads to uniform attention is likely a feature of the optimization, as we use fully random initialization. If we use construction-consistent initialization (with large magnitude), the heads do not degenerate.
> >     - From theoretical perspective, our construction of the transformer doesn't require learning with exactly $K=L$ heads. It remains valid with $K < L$ active heads. In this case, the constructed second layer will use $K$ context examples to implement BMA since in the first layer, $K$ heads copy $K$ examples.
> > * **Attention Notation:** The notations $\sigma(\cdot)$ and $\text{softmax}(\cdot)$ are intended to be the same. We have standardized the notation in the revision.
> > * **BMA Formulation:** We have added **Lemma 1** in the main body to explicitly write out the analytical BMA formulation as requested.
> > * **Theorem 3 Expectations:** We have revised the theorem to explicitly state the stationarity assumption and clarify the distribution under which the expectations are taken.
> > * **Conventional Architecture:** This question is similar to Q2-Q6. We have provided results for the disentangled transformer with Absolute Positional Embedding in **Appendix G** and results of standard transformer with MLP in **Appendix I**. As discussed above **(in Answer to Q2 & Q3-Q6 and our General Response)**, these results support our arguments in the main body.
> >
> > ### References:
> > [1] Francesco D’Angelo, Francesco Croce, and Nicolas Flammarion. Selective induction heads: How transformers select causal structures in context. In The Thirteenth International Conference on Learning Representations, 2025.
> >
> > [2] Eshaan Nichani, Alex Damian, and Jason D. Lee. How transformers learn causal structure with gradient descent. In Forty-first International Conference on Machine Learning, 2024.

---

### Author Response · Authors · 2025-11-25
**General Response to All Reviewers (Part 1)**

We thank all reviewers for their insightful and constructive feedback. We are encouraged that reviewers recognized the novelty of our task (M4B8), the principled nature of our theoretical connection to BMA (EPSS, 9XhA, BaHK), and the sound execution of our experiments (h4T5).

We have uploaded a revised manuscript that incorporates substantial updates, with the revisions highlighted in orange to facilitate the review process. Below, we address the shared concerns regarding architectural generality and provide a concise summary of the other major revisions.

### **Common Response 1: Generalization to Standard Architectures (New Appendix G & I)**
**(Relevant to Reviewers: EPSS, M4B8, h4T5, 9XhA, BaHK)**

A primary concern shared by multiple reviewers was whether our findings and mechanisms are artifacts of the specific simplified architecture used in our initial construction (two-axis Relative Position Embedding & simplified weights with zero blocks). We have added **Appendix G** in the revision to address the concern about architectural generality:
* **Standard Structure (Appendix G):** We implemented and trained a **standard** Disentangled Transformer using standard **Absolute Position Embeddings** (APE). As proven in Nichani et al. [1] (Theorem 3), this structure is mathematically equivalent to a **standard decoder-based transformer**.
* **Theoretical Construction (Appendix G.1):** We construct a 2-layer disentangled transformer based on the design principles of our RPE model. We demonstrate that the correctness of its parent selection is guaranteed via our information-theoretic proof. Empirically, we verify that its parent selection loss (0.0706) is comparable to that of BMA (0.0473) (Fig. 17).
* **Results of Trained Transformer (Appendix G.2):** We conducted experiment on training standard disentangled transformers: Fig. 17 shows the trained model achieves higher accuracy in parent selection during training approximating the construction. Fig. 19 of attention visualization shows that the disentangled transformers learn the same behavior in attention mechanism as the model considered in the main body: first layer copies previous examples while second layer recognizes causal parents. Fig. 18 shows the trained weights align with the construction.
* **Conclusion:** The BMA mechanism is robust to the choice of architectures (e.g., positional encoding) and not limited to our theoretical construction for the reduced model.

To further assess alignment with real-world application scenarios, we also ran experiments with a **standard transformer including MLP layers** (Appendix I). In experiments, the second-layer attention of standard transformer also exhibits the same parent-selection mechanism as in the main results, indicating that the learned causal-inference behavior is robust to more empirical architectural variations and not tied to the simplified analytical construction.


### **Other Major Revisions**

**2. Clarification of Prior Work & Novelty (Appendix A & C)**

To address concerns from Reviewer EPSS, we have explicitly discussed D’Angelo et al. [2] in **Appendix A**. We clarified that their work infers a **global** structural parameter (i.e., a fixed lag), whereas our method infers **local**, position-specific dependency trees. We further elaborated on how our construction differs in its formulation of the inference algorithm.

In addition, we highlighted several contributions that go beyond this distinction: 1) **parameter-level mechanism verification**; 2) **the information-theoretic understanding of the mechanism**, and 3) **the extension to dynamical systems**.

To help readers better navigate our contributions, we added a conclusion section summarizing the above in **Appendix C**, following the suggestions from Reviewers M4B8 and 9XhA.

---

> ### Author Response · Authors · 2025-11-25
> **General Response to All Reviewers (Part 2)**
>
> **3. Mathematical Rigor**
>
> Following Reviewer h4T5's suggestions, we have refined multiple formal statements:
>
> * **Theorem 1:** We replaced informal approximations ($\approx$) with precise limits ($\beta \to \infty, L \to \infty$). In the initial version, the approximation symbol was used to briefly illustrate that transformers can approximate the formulation of BMA and predict the transition probability; in the revision, we have rigorously formalized the statement using the language of limits.
>
> * **Theorem 3:** The first version did not specify the distribution under which the expectations were taken and omitted the required chain condition. We now explicitly state the expectations and stationarity assumptions ($x_h \sim \mu^\pi$) necessary for the gradient dynamics analysis.
>
> * **Proposition 2:** This result aims to characterize the limitation of transformers in representing the BMA inference method in the dynamical system setting. Unlike the discrete Markov chain case, an additional quadratic term arises in BMA, which transformers cannot represent. In the revision, we formalize the condition under which this discrepancy occurs and consequently leads to the failure of transformers to represent BMA.
>
>
> **4. Presentation Improvements**
>
> Several reviewers noted the density of the submission and suggested clarifying the presentation. To improve the flow and ensure key messages stand out, we have substantially reorganized the manuscript:
>
> * **Enhanced Figure Readability:** We have enlarged key figures (e.g. **Figures 2, 3 and 10**) for better readability.
> * **Standardized Notation & Narrative**: We formalized notation in the main text, refined transitions for smoother flow, and revised the proof overviews to better clarify the intuition behind our theoretical results.
> * **Terminology Disambiguation:** We now explicitly distinguish between the *standard disentangled transformer* and the *standard transformer*. Besides, newly introduced symbols are defined in-place within the related equations.
> * **"Takeaway" Summaries:** We added **"Takeaway"** blocks after key sections (i.e., Takeaways 1–5) to concisely highlight our primary theoretical and empirical findings.
> * **New Conclusion Section:** We added a **Conclusion** section (Appendix C) to summarize the key contributions of our work.
>
> ### References:
> [1] Eshaan Nichani, Alex Damian, and Jason D. Lee. How transformers learn causal structure with gradient descent. In Forty-first International Conference on Machine Learning, 2024.
>
> [2] Francesco D’Angelo, Francesco Croce, and Nicolas Flammarion. Selective induction heads: How transformers select causal structures in context. In The Thirteenth International Conference on Learning Representations, 2025.

---

### Meta-Review · Area_Chair_7vdJ · 2025-12-13

**Summary:**

There is a mix of strong theoretical contributions and initially significant concerns about novelty positioning, generality, and presentation. Several reviewers acknowledged the principled approach, the soundness, and the novelty. However, early reviews raised substantial concerns regarding insufficient acknowledgment of prior work, lack of generality, lack of clarity in theoretical statements, and missing or unclear conclusions. Through the rebuttal and revisions, the authors addressed most of these issues by strengthening the related-work discussion, formalizing theorems, adding experiments on more standard architectures, and improving presentation and structure.

**Reviewer Concerns:**

Addressed

- Insufficient acknowledgment and contextualization of prior work
- Generality concerns were mitigated by new theoretical constructions and experiments
- Mathematical rigor issues in Theorems 1 and 3 and Proposition 2 were resolved by replacing informal approximations with precise limits and explicitly stating assumptions.
- Missing conclusion, unclear takeaways, and presentation issues (notation, figure readability, structure) were substantially improved through reorganization, added takeaway blocks, and a new conclusion section.
- Questions were addressed conceptually and empirically.

Unresolved

- Broader generality and real-world impact
- Scaling limitation

**Reviewer Scores:**

- Reviewer EPSS: Likely to increase modestly but remain cautious. While concerns about prior work, rigor, and architectural tailoring were addressed in detail, the reviewer originally viewed the contribution as limited; the score would likely move upward but not to a strong accept.
- Reviewer M4B8: Likely unchanged or slightly improved. The reviewer already leaned positive and explicitly stated they would not mind acceptance; improved clarity, added conclusions, and generality experiments address most concerns.
- Reviewer h4T5: Likely unchanged or slightly more positive. The reviewer was already marginally above acceptance and explicitly requested the exact changes that were implemented (rigor, standard architectures), which strengthens confidence in the work.
- Reviewer 9XhA: Increased score to 6 after the rebuttal, indicating that their concerns were fully addressed.
- Reviewer BaHK: Likely unchanged. The reviewer was already positive and indicated satisfaction with the clarifications while keeping their score.

---

### Decision · Program_Chairs · 2026-01-26

Accept (Poster)